# Rising atmospheric CO$_2$ reduces nitrogen availability in boreal forests

Kelley R. Bassett[1✉], Stefan F. Hupperts[1], Sandra Jämtgård[1], Lars Östlund[1], Jonas Fridman[2], Steven S. Perakis[3] & Michael J. Gundale[1]

Anthropogenic nitrogen (N) pollution is a cause of eutrophication globally[1]. However, recent datasets indicate that some ecosystems may be experiencing widespread oligotrophication—declining N availability—which is suggested to be a response to elevated atmospheric carbon dioxide (CO$_2$)[2]. Plant N isotope ($\delta^{15}$N) chronologies have served as primary evidence for oligotrophication, but there is wide disagreement whether rising CO$_2$ or temporal changes in N deposition explain these patterns[3–6]. Here we construct $\delta^{15}$N tree-ring chronologies using archived samples from Sweden's 23.5-million-hectare forest area from 1961 to 2018. The study area spans a 1,500-km latitudinal distance where N deposition varies fourfold, but where rising CO$_2$ is spatially uniform. Our data show declining $\delta^{15}$N chronologies throughout Sweden, including forests in the far north where atmospheric N deposition rates are very low. Linear mixed-effects models showed that rising CO$_2$ is the strongest predictor of $\delta^{15}$N values, whereas N deposition variables, temperature and forest basal area had lower explanatory power. Our findings suggest that elevated atmospheric CO$_2$ is causing oligotrophication in boreal forests, which has implications for predicting their future role as sinks in the global carbon cycle[7–9].

Human-induced environmental change has approached or exceeded several planetary Earth-system process boundaries[10], in part owing to substantial alterations to both the global nitrogen (N) and carbon (C) cycles. Since 1960, humans have increased the rate of reactive nitrogen (Nr) creation by a factor of 10 owing to the industrial Haber–Bosch process, expansion of leguminous crops and fossil fuel combustion; and Nr creation is expected to increase by a factor of 18 by 2050 (ref. 11). In addition, considerable atmospheric Nr emissions and subsequent deposition of NH$_x$ and NO$_y$ to Earth's surface occur as a consequence of global energy and food production[12]. These changes have enriched terrestrial and aquatic ecosystems with N, causing well-documented eutrophication and acidification problems in many regions with high-Nr inputs[13]. At the same time, human activities have increased atmospheric carbon dioxide (CO$_2$) concentrations by more than 50% since the start of the industrial age, which has been shown to enhance terrestrial net primary productivity (NPP)[14,15]. Furthermore, enhanced NPP from rising atmospheric CO$_2$ has been proposed to interact with the N cycle, potentially reducing N availability and intensifying N limitation of terrestrial NPP (referred to as progressive N limitation (PNL)[16] or ecosystem oligotrophication[2,17]). PNL may occur if rising CO$_2$ stimulates plant growth and N uptake that reduces soil N availability for further plant growth, or if CO$_2$ increases the plant C:N ratio that stimulates microbial N immobilization during detrital decomposition, thus also reducing soil N availability. Consequently, although it is well known that Nr deposition causes eutrophication in some regions, it is plausible that rising CO$_2$ may have an opposing effect, leaving substantial uncertainty in the trajectory of terrestrial N limitation,

which has implications for Earth-system model predictions of the future terrestrial C sink[18].

Although eutrophication owing to excess N has been an active focus of research over many decades in natural ecosystems, several recent datasets instead indicate that widespread oligotrophication may be occurring. Chronologies (that is, time series) of N stable isotope ratios ($\delta^{15}$N) in plant tissues that span decades to centuries have served as key evidence for oligotrophication because they integrate multiple N-cycling processes in ecosystems, yet the interpretation of these chronology datasets has been the subject of ongoing debate[2–6]. The $\delta^{15}$N values of plants broadly reflect sources of plant N as well as soil and ecosystem N availability, with higher N availability yielding elevated plant $\delta^{15}$N values[19]. This pattern arises from $^{15}$N/$^{14}$N isotopic fractionation that occurs at both the ecosystem level and during plant N uptake[20–22]. At the ecosystem level, high N availability relative to plant N demand promotes losses of soil inorganic N with low $\delta^{15}$N values (for example, ammonia volatilization, nitrification, nitrate leaching and denitrification), which increases the $\delta^{15}$N of remaining soil N pools and of the ecosystem as a whole, thus increasing plant $\delta^{15}$N (ref. 22). During plant N uptake in forests, the primary fractionation mechanism is through the production of $^{15}$N-enriched mycorrhizal tissue and preferential transfer of $^{14}$N to host plants, with the greatest fractionation occurring at low N availability and for ectomycorrhizal associations[20]. In contrast to the above, only small N-isotope discrimination occurs by direct root N uptake at levels present in natural ecosystems, and N-isotope ratios are generally well correlated among plant tissues (leaves, stems and roots)[22]. Thus, when N limitation intensifies, reduced N losses and

[1]Department of Forest Ecology and Management, Swedish University of Agricultural Sciences, Umeå, Sweden. [2]Department of Forest Resource Management, Swedish University of Agricultural Sciences, Umeå, Sweden. [3]US Geological Survey, Forest and Rangeland Ecosystem Science Center, Corvallis, OR, USA. ✉e-mail: kelley.bassett@slu.se

greater plant reliance on mycorrhizal fungi can steer plant $\delta^{15}N$ values more negative.

The spatial relationship between plant $\delta^{15}N$ values, foliar N content and N availability is now well established and accepted[2,23,24], and as a result, $\delta^{15}N$ values have more recently been used as a proxy to indicate changes in N limitation through time[21], where $\delta^{15}N$ plant chronologies with negative slopes have been interpreted as evidence for oligotrophication[2,4,25]. Notably, wood samples from forests across the continental USA showed a slight temporal decline of $\delta^{15}N$ (about $-0.009 \pm 0.002‰$ yr$^{-1}$, 1850 to 2010)[25]; furthermore, a meta-regression of published foliar $\delta^{15}N$ data found a temporal decline of $\delta^{15}N$ globally ($-0.043 \pm 0.014‰$ yr$^{-1}$, 1980 to 2017)[2]. Most recently, ref. 4 compiled a range of evidence for widespread ecosystem oligotrophication, including temporal $\delta^{15}N$ declines for several sample types, namely, foliar, wood, lake sediment and herbarium samples, as well as other N-cycling response variables, for example, direct measurement of watershed N exports and soil N mineralization. Corresponding datasets showing temporal declines of foliar N content provide further support for interpretations of reported $\delta^{15}N$ chronologies[2,26].

In response to these findings, alternative interpretations of declining $\delta^{15}N$ trends have been proposed. First, it is proposed that increasing N limitation over the past three decades is instead the result of declining Nr deposition rates (that is, declining eutrophication), which reached peak levels in North America and Europe in the 1980s[3,6]. Furthermore, it is suggested that the decline in N deposition rates during recent decades has been driven by policy-regulated reductions in the oxidized forms of Nr deposition ($NO_y$), which are derived from fossil organic matter combustion and typically have higher $\delta^{15}N$ values[6], whereas reduced forms ($NH_x$) have increased[3]. Shifting $\delta^{15}N$ values of Nr deposition owing to a change in the $NH_x{:}NO_y$ ratio is therefore an additional alternative explanation for the observed declining $\delta^{15}N$ values of plants, instead of $CO_2$-induced oligotrophication. Furthermore, a study[27] applying an ecosystem $^{15}N$ fractionation model concluded that several other factors in addition to increasing $CO_2$ can plausibly explain declining $\delta^{15}N$ values, including the amount and signature of Nr deposition inputs, and changes in temperature. Consequently, there remains wide disagreement regarding the drivers of reported $\delta^{15}N$ chronologies, and thus uncertainty in their interpretation.

## Constructing $\delta^{15}N$ chronologies

To address this debate, we analysed $\delta^{15}N$ values from 1,609 independent, archived tree cores collected between 1961 and 2018 (Fig. 1a); decadal increments from tree cores representing the time span 1950 to 2017 across Sweden's 23.5-million-hectare forest area. We focused on $\delta^{15}N$ rather than %N, given that %N of wood is not considered a good indicator of plant N limitation owing to its very low concentration and subsequent sensitivity to concentration variability of more abundant wood elements (for example, C and oxygen (O))[22,28–31]. Swedish forests span a 1,500-km latitudinal gradient of Nr deposition, with southern Sweden experiencing around fourfold higher Nr deposition during the industrial age, as well as stronger reductions in total and $NO_y$ deposition in response to emissions-reduction policy[32,33]. In contrast, Nr deposition rates in northern Sweden have been consistently low and stable[33] (Fig. 1a,b). Compared with the strong gradients in Nr deposition across Sweden, rising atmospheric concentrations of $CO_2$ gas are generally well mixed and show relatively negligible spatial variation (about 0.5% variation regionally)[34].

Owing to the independent spatio-temporal relationship between atmospheric Nr deposition and rising $CO_2$ across Sweden (Extended Data Fig. 1), the Swedish forest landscape is an ideal large-scale study system to separate the debated drivers of $\delta^{15}N$ chronologies and thus reconcile their interpretation. Furthermore, we use a sampling approach that yields more robust $\delta^{15}N$ chronologies than previous efforts. First, we acquired wood samples for two tree species, *Picea abies* (L.) H. Karst

and *Pinus sylvestris* L., from the Swedish National Forest Inventory (NFI) tree-core archive for which sample collection occurred between 1961 and 2018, using a stratified random sampling approach (Methods). This systematic approach has several advantages. First, although using wood cores can be problematic for establishing $\delta^{15}N$ chronologies owing to tree ageing and N translocation across xylem[35–37], our use of archived samples allows us to sample trees of the same age class across space and through time, eliminating these inherent problems[38]. Second, this systematic approach allows us to control for spatial or temporal variation in the species sampled, as well as tree and site characteristics, factors that have not been held constant in previous analyses[2,25,39]. Finally, our focus on wood $\delta^{15}N$ avoids concerns raised regarding the interpretation of foliar %N chronologies as an indicator of PNL, which have alternatively been suggested to result from photosynthetic down-regulation instead of PNL, as this alternative explanation would not directly affect wood N-isotope ratios[3].

We first analysed our data by dividing Sweden into four regions—north, central, southeast and southwest—which represent a large gradient of atmospheric Nr deposition levels ranging from very low in the north (≤3 N kg ha$^{-1}$ yr$^{-1}$) to their maximum rates in the southwest (regional mean approximately 12 kg N ha$^{-1}$ yr$^{-1}$; Fig. 1b) when at their peak in the late 1990s[40]. The $\delta^{15}N$ chronologies for each of these regions were negative for both *P. sylvestris* and *P. abies* (between $-0.01‰$ yr$^{-1}$ and $-0.07‰$ yr$^{-1}$; Fig. 1c–j). These $\delta^{15}N$ chronologies were markedly more negative and less variable than some previously published $\delta^{15}N$ chronologies[25,32,41,42], because our methodology was not confounded by tree age or N translocation across tree rings, thus making our measurements more sensitive to detect temporal changes in ecosystem N-cycle processes[38]. Our data also showed that pairwise comparisons of the $\delta^{15}N$ chronology slopes never differed between the low- and high-Nr-deposition northern and southern regions, respectively, for both *P. sylvestris* and *P. abies*, indicating that N deposition was not strongly influential on $\delta^{15}N$.

## Drivers of forest $\delta^{15}N$ variation

We constructed linear mixed-effects models to directly compare the explanatory power of $CO_2$ versus atmospheric Nr deposition variables on the spatial and temporal variation of tree-ring $\delta^{15}N$ values, while controlling for additional factors of mean annual temperature, temperature change and total basal area. Given that all Nr deposition variables (total N deposition, $NH_x$, $NO_y$ and $NH_x{:}NO_y$) were strongly co-linear (Extended Data Table 1), we selected four separate models, substituting each Nr deposition variable in successive model runs, as well as successive model variants that included either latitude and longitude, and absolute temperature or temperature change (Extended Data Tables 2 and 3).

The model results showed that $CO_2$ was consistently the strongest predictor of $\delta^{15}N$ values (see partial marginal coefficient of determination ($R^2$m) values in Extended Data Tables 1–3). This conclusion persisted among model variants that held space constant by including latitude and longitude as fixed factors (Extended Data Table 2), as well as when absolute temperature or temporal temperature change was considered (Extended Data Table 3). The relationship between $\delta^{15}N$ and $CO_2$ was strongly negative for both *P. sylvestris* ($-0.04 \pm 0.005‰$ $\delta^{15}N$ ppm$^{-1}$ $CO_2$) and *P. abies* ($-0.028 \pm 0.005‰$ $\delta^{15}N$ ppm$^{-1}$ $CO_2$; Fig. 2a). The emergence of $CO_2$ as the strongest predictor of declining $\delta^{15}N$ values for both tree species is consistent with ecosystem oligotrophication in response to increasing atmospheric $CO_2$ (Fig. 2). Furthermore, *P. sylvestris* showed a small but significantly more negative relationship with $CO_2$ than *P. abies* (Fig. 2a). It is plausible that this small difference between the two species is related to their N-use efficiencies. Specifically, *P. sylvestris* has a higher rate of needle turnover and thus lower N-use efficiency than *P. abies*[43], making *P. sylvestris* more reliant on annual soil N uptake for growth, and which may reveal stronger effects of $CO_2$ on $\delta^{15}N$ chronologies in this species.

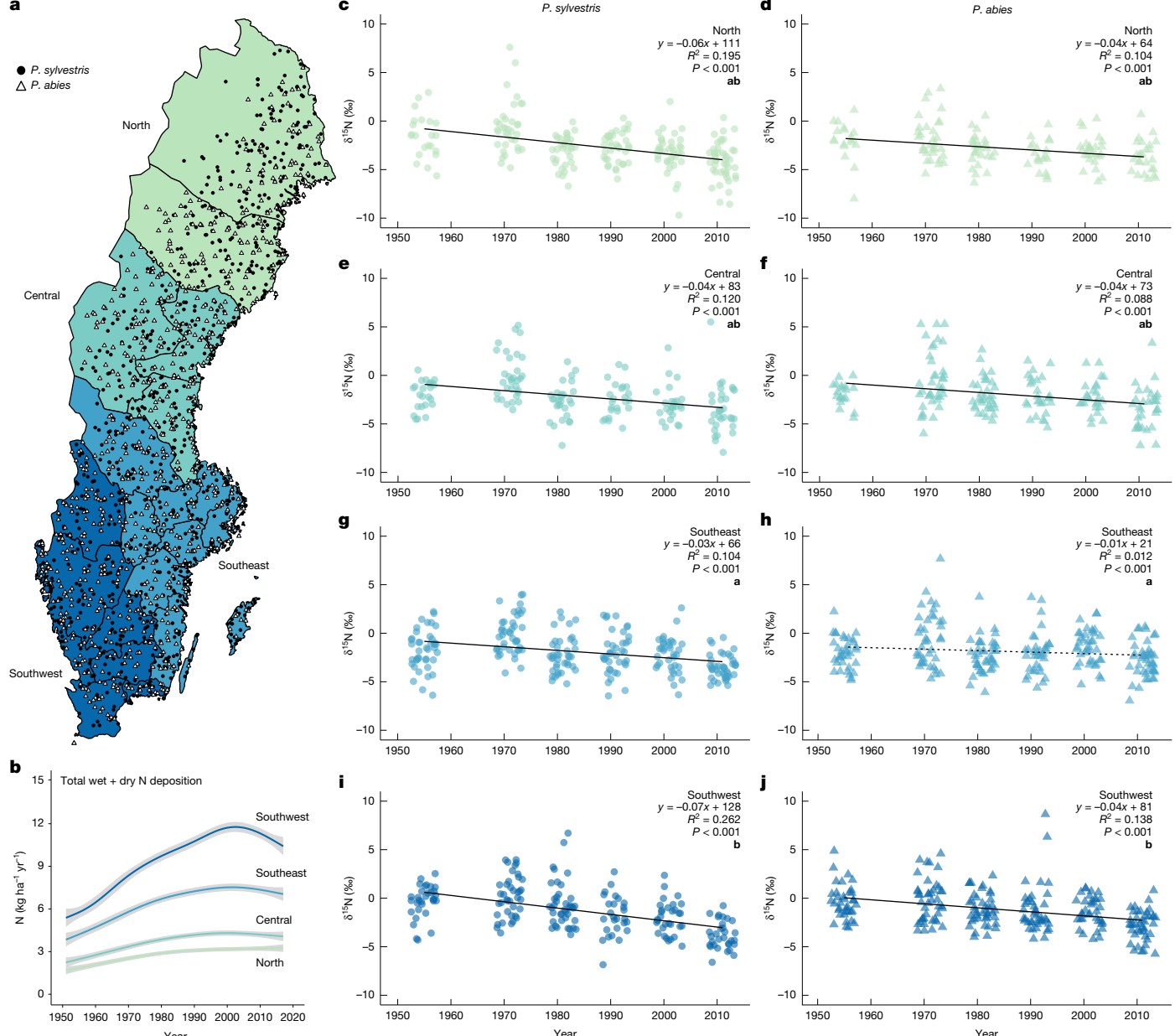

**Fig. 1 | Spatial and temporal patterns of wood δ[15]N and N deposition across Sweden. a**, Spatial distribution of independent samples ($n = 1609$) across 4 Nr deposition regions spanning Sweden's north–south gradient, representing increasing deposition rates from lowest (north) to highest (southwest). **b**, Temporal change in total wet plus dry N deposition (kg ha[−1] yr[−1]) for samples across the regions. **c–j**, Temporal linear regression trends in stemwood δ[15]N (‰) for *P. sylvestris* (**c**,**e**,**g**,**i**) and *P. abies* (**d**,**f**,**h**,**j**) from 1951 to 2017 in each of the Nr deposition regions: north (**c**,**d**), central (**e**,**f**), southeast (**g**,**h**) and southwest (**i**,**j**). All relationships are significant ($P < 0.001$) except for *P. abies* in the southeast region (**h**), where the non-significant relationship is shown as a dashed line. Lowercase bold letters in **c–j** denote significant differences in regression slopes among regions, within each species ($\alpha = 0.05$). Shading in **b** indicates the 95% confidence interval. A small horizontal offset was applied to time plots **c–j** to increase the visibility of points (see 'Statistical methods'). Data sources are cited in Methods. Sweden boundary map in **a** from Lantmäteriet under a Creative Commons licence CC0 1.0.

Regarding Nr deposition variables, the NH$_x$:NO$_y$ ratio had no significant relationship with δ[15]N. However, the remaining three Nr deposition variables (total N deposition, NH$_x$ and NO$_y$) showed weak positive relationships with δ[15]N, which could indicate either that higher Nr deposition increases N availability or that changing δ[15]N values of Nr deposition directly alters the δ[15]N values of trees. Regardless of the mechanism, the positive relationship we identified between these Nr deposition variables and δ[15]N provides some support that declining Nr deposition can steer δ[15]N chronologies in the negative direction, given that Nr deposition has been declining since the late 1990s in southern Sweden[6] (Fig. 1b). However, we highlight that the explanatory power of these Nr deposition variables is substantially lower than CO$_2$ (partial $R^2$m values ≤0.005 versus 0.176, respectively; Extended Data Table 1), and further, the suggested importance of NH$_x$:NO$_y$ dynamics was not supported by our data.

Our analysis also showed that δ[15]N showed a positive relationship with 10-year averages of both annual temperature and total basal area (Fig. 2f,g). The positive relationship between temperature and δ[15]N persisted regardless of whether absolute temperature or temperature change were considered in the model. Positive relationships of δ[15]N with temperature or temperature change[23,44] are attributed to faster N cycling and greater fractionating N loss in warmer and warming climates, which increases the δ[15]N value of soil inorganic N for plant uptake. Basal area was the only predictor variable derived from

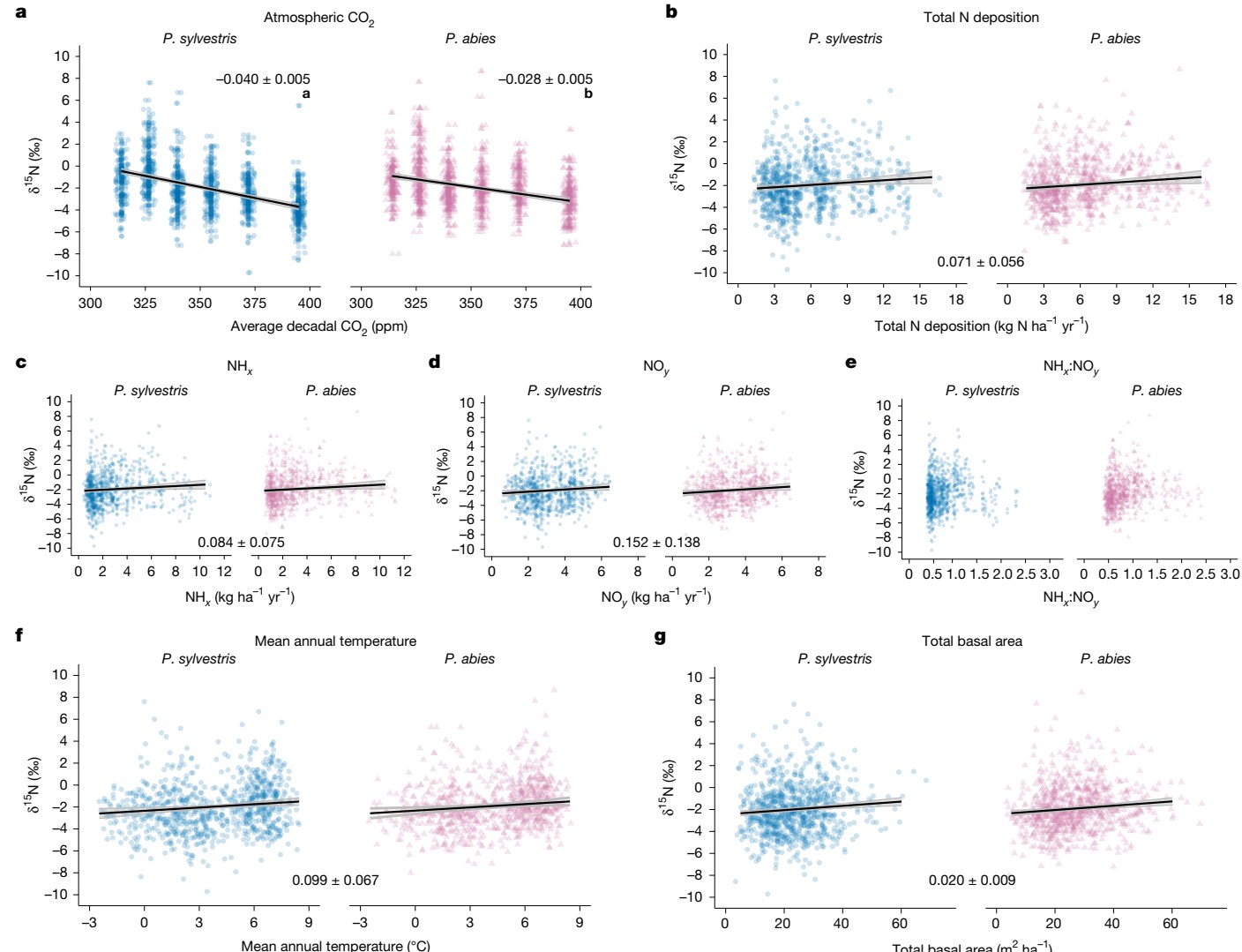

**Fig. 2 | Modelled relationship between wood δ¹⁵N and environmental drivers across Sweden. a–g**, Wood $\delta^{15}N$ as a function of: the main predictors in the final model, atmospheric $CO_2$ (**a**) and total N deposition (**b**); alternative N deposition variables, $NH_x$ (**c**), $NO_y$ (**d**) and $NH_x$:$NO_y$ (**e**) (Extended Data Table 1b–d); mean annual temperature (**f**); and total basal area (**g**). Regression lines show modelled $\delta^{15}N$ values accounting for fixed and random effects. Numbers within each panel indicate modelled slopes ± 95% confidence intervals. Shading represents 95% confidence intervals. Lowercase bold letters in **a** denote significant differences in regression slopes between species ($\alpha = 0.05$). The relationship between $\delta^{15}N$ and $NH_x$:$NO_y$ in **e** was not significant; therefore, no regression line is shown. A small horizontal offset was applied to plots **a**–**e** to increase visibility of points (see 'Statistical methods'). Data sources are cited in Methods.

individual sample plots and thus accounted for plot-to-plot variation in stand productivity within grid cells, while also representing forest biomass variation with latitude and time[45,46] (Extended Data Fig. 2i,j). The weak positive relationship may reflect that higher-fertility sampling plots within a given grid cell have higher basal area, after accounting for some degree of broad spatial or temporal co-linearity with temperature and $CO_2$, which were environmental factors with greater explanatory power in our model[47]. Despite the additional significant effects of temperature and basal area in our models, $CO_2$ consistently emerged as the dominant factor explaining forest $\delta^{15}N$ variation, indicated by its much larger partial $R^2$m in all model variants (Extended Data Tables 1–3).

## Implications

Our results have multiple implications for understanding the response of boreal forest N availability to environmental change factors, particularly rising atmospheric $CO_2$ and anthropogenic Nr deposition. Foremost, the highly significant effect of atmospheric $CO_2$ on tree-ring $\delta^{15}N$ that we observed, using a well-constrained sampling methodology

that minimizes isotopic effects associated with tree ageing and N translocation across tree rings[38], provides robust evidence that negative $\delta^{15}N$ chronologies are an indicator of ecosystem oligotrophication in response to rising $CO_2$ (refs. 2,4,16,25), rather than a symptom of changing Nr deposition patterns[3,6]. As our sampling area spanned a large gradient in latitude, forest biomass, mean annual temperature and Nr deposition, these findings are broadly relevant for understanding the trajectory of N availability in northern forests where declining $\delta^{15}N$ chronologies have previously been identified[48,49]. Furthermore, global increases in $CO_2$ may indirectly influence $\delta^{15}N$ chronologies owing to the influence that rising $CO_2$ has on climate warming. Our analysis showed that temperature and temperature change had opposite and weaker relationships than $CO_2$ with $\delta^{15}N$ (positive versus negative, respectively), reinforcing that the direct influence of rising $CO_2$ was a more dominant driver than any potential indirect effect that rising $CO_2$ has on warming. These findings are further consistent with meta-analysis of elevated $CO_2$ (e$CO_2$) experiments showing a predominance of declines in foliar $\delta^{15}N$ across multiple experiments[50], and datasets indicating that $CO_2$ may be tightening the N cycle in many environments, including

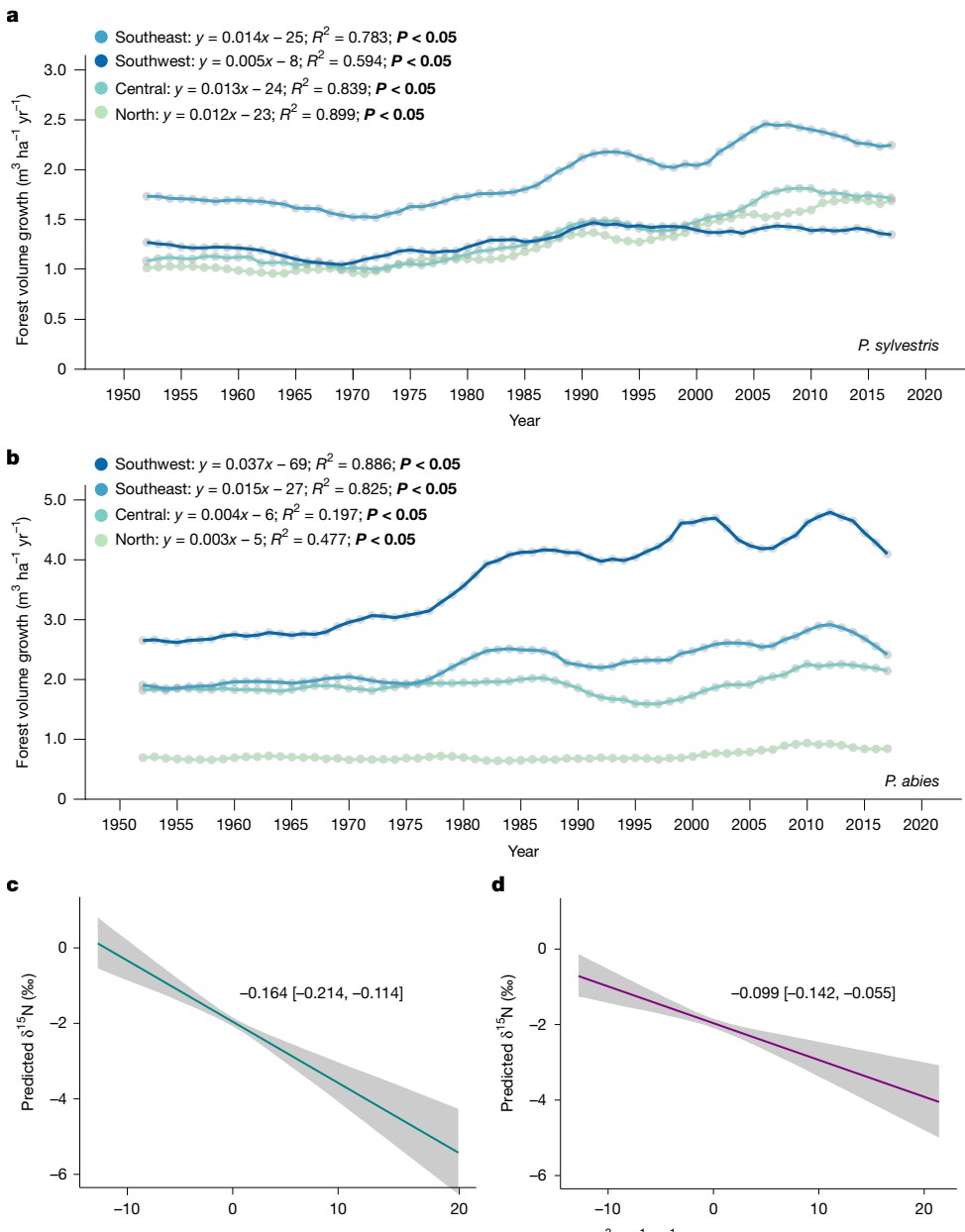

**Fig. 3 | Forest growth dynamics and their relationship with δ¹⁵N across Swedish regions. a**,**b**, Annual forest volume growth (m³ ha⁻¹ yr⁻¹) for *P. sylvestris* (**a**) and *P. abies* (**b**) by region—north, central, southeast and southwest Sweden—over 1950–2017. Significant linear regression slopes ($P < 0.05$) are in bold; however, piecewise linear interpolation is shown for enhanced visualization. **c**, Predicted δ¹⁵N (‰) values as a function of relative forest volume growth change (m³ ha⁻¹ yr⁻¹) accounting for random effects (grid) and fixed effect of absolute forest volume growth (m³ ha⁻¹ yr⁻¹). **d**, Predicted δ¹⁵N (‰) values as a function of relative forest volume growth change (m³ ha⁻¹ yr⁻¹) accounting for random effects (grid) and fixed effects of total basal area (m² ha⁻¹). Regression lines show modelled δ¹⁵N values accounting for all fixed and random effects. Shading in **c** and **d** indicates 95% confidence intervals. Data sources are cited in Methods.

declining foliar N content in Europe and North American forests[2,26], declining riverine N exports in central Europe and North America[4,51], and declining stream water N exports in northernmost Sweden, where Nr deposition is very low[52–55].

Boreal forests similar to those we studied across Sweden have an important role in the global C cycle by accumulating and storing a considerable amount of terrestrial C (about 17% of the terrestrial land area, but accounting for about 32% of terrestrial C)[46,56]. Our findings have further implications for understanding how rising $CO_2$ may impact their future C sink strength. Two non-mutually exclusive mechanisms provide a link between rising $CO_2$ and declining N availability and δ¹⁵N chronologies. First, rising $CO_2$ can increase plant photosynthesis, growth and N demand[57] in accumulating biomass, thus depleting available

N from soil and reducing fractionation associated with ecosystem N losses[16,22]. At the same time, rising $CO_2$ can increase plant C:N ratios and the production of high C:N litter, which increases the N demand for detritus decomposition, and stimulates soil microbial N immobilization[9,58,59]. Both of these responses to rising $CO_2$ would be expected to reduce fractionating N loss and the δ¹⁵N values of inorganic N in soil, and thus steer δ¹⁵N chronologies more negative through time[4].

In support that increased forest growth may contribute to reducing N availability, data from the Swedish NFI show that growth of mesic pine and spruce forests, similar to those from which we established δ¹⁵N chronologies, shows a long-term growth increase since the 1950s in all regions (Fig. 3a,b). Our analysis further indicates a strong negative relationship between the temporal change in forest growth and

wood $\delta^{15}N$ (Fig. 3c,d), which is consistent with the prediction of PNL that increasing plant N demand promotes a negative feedback on N availability (Fig. 3 and Extended Data Table 4). It has alternatively been suggested that photosynthetic downregulation of foliar N is a dominant process that reduces plant N demand in response to rising $CO_2$ (refs. 3,60). Although not mutually exclusive, our analysis indicates that increasing forest N demand at the stand level, owing to increasing biomass and growth, appears to outweigh any potential physiological reduction in N demand at the leaf level[60]. Furthermore, given that the photosynthetic downregulation hypothesis predicts reduced plant N demand in response to rising $CO_2$, this alternative hypothesis does not provide a mechanism to explain the declining $\delta^{15}N$ chronologies we observe, nor the observed patterns of declining aquatic N exports from Swedish forest landscapes[53,55]. In recent decades, forest growth has levelled off in many parts of Sweden (Fig. 3a,b), which is also consistent with predictions by some Earth-system models that PNL feedbacks eventually constrain the response of terrestrial NPP to rising $CO_2$ (refs. 18,61–63), possibly interacting with other factors (for example, drought and disease). Two previous experiments in northern Sweden that applied experimental $CO_2$ treatments, that is, $eCO_2$, also reached the conclusion that forest growth response to $eCO_2$ is constrained by N availability[64]. These findings support our interpretation that reduced N availability, as indicated by our $\delta^{15}N$ chronologies, may reduce the sensitivity of boreal forest growth to future increases in $CO_2$ (refs. 7,16).

In addition to increased growth and plant C:N ratios of plant biomass and detritus, mycorrhizal fungi provide the other non-mutually exclusive mechanism that links rising $CO_2$ to declining $\delta^{15}N$ chronologies. As forest N demand increases, trees may enhance their C investment in mycorrhizal fungi to enhance soil N acquisition, particularly as soil inorganic N availability declines[65–67]. Many tree species worldwide, and especially those in high-latitude regions such as boreal forests, associate with ectomycorrhizal fungi that mobilize N from the soil[65,68]. During N transfer from fungus to host trees, mycorrhizal fungi disproportionately retain $^{15}N$ in their hyphal mass, and pass $^{14}N$ to their tree hosts, with the degree of such $^{15}N$ fractionation increasing as N availability declines[20]. Thus, the declining $\delta^{15}N$ chronologies we observed may reflect that an increasing proportion of plant N is obtained via ectomycorrhizal fungi, possibly owing to an increase in fungal biomass that sequesters a larger fraction of the mobilized N pool. Furthermore, studies from boreal forest fertility gradients indicate that some specific ectomycorrhizal taxa are especially responsive to increasing N limitation intensity, including some taxa that acquire N by degrading soil organic matter (that is, 'ectomycorrhizal decomposers')[65,66,68,69]. A recent meta-analysis of 108 $eCO_2$ experiments showed that although $eCO_2$ leads to increased soil C in most ecosystem types, ecosystems dominated by ectomycorrhizal tree species showed no such increase, despite positive NPP responses[70]. This suggests that acquisition of soil organic N by ectomycorrhizal decomposers may allow sustained forest growth in response to rising $CO_2$ (ref. 71), but this may subsequently constrain soil C accumulation in these forest mycorrhizal types.

Resolving the dominant mechanisms of the boreal forest ecosystem response to rising $CO_2$ has important implications for C balances, and especially the distribution of above- versus belowground C (ref. 67). Carbon stocks of boreal soils are about fivefold greater than C stored in aboveground biomass[66] and may respond differently to $eCO_2$ owing to increases in tree growth versus upregulation of ectomycorrhizal fungal partners for N acquisition[70]. Both mechanisms probably operate simultaneously and contribute to our finding of declining $\delta^{15}N$. Determining the relative importance of these two mechanisms as controls on $\delta^{15}N$ values across different boreal forest contexts, and the resulting sensitivity of above- versus belowground C compartments in response to $CO_2$-induced PNL may be addressed through a combination of approaches, such as intercomparisons of coupled C–N Earth-system models, and deployment of $eCO_2$ experiments, which

are poorly represented in boreal ecosystems. Although it is clear that humans have already surpassed several of Earth's planetary boundaries[10], in part owing to alterations of both the N and C cycles, our data show that these cycles are not changing independently. It further suggests that declining N availability in response to rising $CO_2$ will be a more dominant driver of boreal forest C exchange in the future, compared with N enrichment from atmospheric Nr deposition.

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

## Methods

### Sample collection and preparation

We used tree cores archived by the Swedish NFI between 1961 and 2018 (ref. 72). The Swedish NFI samples approximately 10,000 plots per year on productive, evergreen-dominated forestlands (range 55–69° N, 11–24° E), from which tree cores are collected and archived. To systematically sample independent tree cores from the archive, we applied a grid of 250 square cells (50 × 50 km) over Sweden. We then identified samples in the archive by filtering their associated database, for cores originating from dominant or co-dominant Norway spruce (*P. abies* (L.) H. Karst) and Scots pine (*P. sylvestris* L.) samples from inventory plots categorized as mesic site types, with slopes of 20% slope or less, and from trees in the 41–60-year-old age class. This age class was selected to minimize the likelihood of N fertilization application; forest fertilization treatment in Sweden is applied to a relatively small area and late in stand rotation[73]. Samples were then randomly selected for each tree species from each grid cell, and for each of 6 sampling decades (1960s–2010s). Specific sampling years were chosen for each decade based on the available number of archived samples: 1961, 1977, 1986–1988, 1996–1998, 2006–2008 and 2016–2018. Three-year periods were sampled for the 1980s, 1990s, 2000s and 2010s because the NFI's sampling intensity was reduced from this period onwards, requiring an expanded selection period to ensure a comparable number of samples to those available in 1961 and 1977. Some grid cells did not contain trees, and occasionally samples were not available for a given species, grid cell and decade combination, resulting in a total of $n = 1,609$ samples collected from the archive for analysis. Data for two counties in Sweden (Jämtland and Västernorrland) were previously reported[38] and were combined with new data for all other counties in Sweden. For each sample, collected at breast height (1.3 m), we removed the outer bark and cambial layers, and the most recent annual ring to exclude incomplete ring growth during the collection year. Then, to obtain sufficient material for chemical analysis, the subsequent 10-year annual growth segment was separated from the remainder of the core[38]. Dissections were performed using a No. 11 stainless steel surgical blade under a stereo microscope with ×20 magnification, with an accuracy of 0.01 mm. As the samples represent a 10-year growth segment, we designated each sample by its corresponding intermediate 10-year growth increment year (for example, samples collected in 1961 were representative of the period 1951 to 1960 and were consequently referred to as 1955).

### Measurement and analysis of N isotopes

Nitrogen isotope ratios ($\delta^{15}N$) were analysed at the Central Appalachians Stable Isotope Facility (CASIF), University of Maryland Center for Environmental Science (UMCES) Appalachian Laboratory (Frostburg, MD, USA) with a Carlo Erba NC2500 elemental analyser (CE Instruments) interfaced with a Thermo Finnigan Delta V+ isotope-ratio mass spectrometer (IRMS). The Carlo Erba NC2500 Elemental Analyser with Costech zero-blank autosampler modifications permits for analysis of N isotopes in solid organic samples with content less than 0.5% N, such as wood[47]. From each tree core, a radial slice was precisely sectioned to represent each 10-year segment and chopped with a steel razor. Approximately 10 mg of wood from the radial core slice was then weighed, placed in a tin and analysed for the $\delta^{15}N$ value. Samples were analysed with a Carbosorb trap to remove $CO_2$ in advance of removing water vapour with magnesium perchlorate ($MgClO_4$). The $\delta^{15}N$ data were normalized to the Ambient Inhalable Reservoir (AIR) scale using a two-point normalization curve with internal standards, including ground corn, cocoa and caffeine powder calibrated against international standards, USGS40 and USGS41. Analytical precision ($1\sigma$) of an internal wood standard (ground pine powder) analysed alongside samples was 0.3‰ for $\delta^{15}N$; atropine powder was used for determining N-content values.

The ratio of heavy ($^{15}N$) to light ($^{14}N$) isotopes of samples is expressed in standard delta ($\delta$) notation with reference to a standard of known isotopic ratio.

$$\delta^{15}N = (R_{sample}/R_{standard} - 1) \times 1,000$$

where $R_{sample}$ and $R_{standard}$ are the ratios of the heavy to light isotopes in the sample and standard, respectively, and are expressed in units of parts per thousand or per mil (‰). The chosen standard for N is AIR. Corresponding wood %N data are reported in Extended Data Fig. 3.

### Climate, Nr deposition and forest data

We leveraged external databases for use in the linear mixed-effects model, including the climate parameters mean annual temperature (°C) and atmospheric $CO_2$ concentrations (ppm); the N deposition parameters $NH_x$ and $NO_y$ (g m$^{-2}$); and the forest stand parameters total basal area and stand age. Monthly temperature data were extracted from CRU TS v4.07 (ref. 74) (resolution 0.5°) from which we calculated a 10-year mean annual temperature specific to each wood core sample, using its unique geographic coordinates. In addition to absolute temperature, we also calculated a relative temperature change variable, using 1961 as the reference temperature value for each grid cell, to minimize the large latitudinal gradient in temperature across our study area. We acquired $CO_2$ data from previously compiled measurements for the period 1951–2004 (ref. 75) and atmospheric $CO_2$ dry air mole fraction data for the period 2005–2017 from the National Oceanic and Atmospheric Administration (NOAA) Global Monitoring Laboratory recorded at PAL: Pallas-Sammaltunturi, GAW Station, Finland (67.9733° N, 24.1157° E; 565 m above sea level)[76]. We obtained monthly wet plus dry deposition as $NH_x$ and $NO_y$ between 1951 and 2017 from the global gridded Inter-Sectoral Impact Model Intercomparison Project (ISIMIP3a; resolution 0.5 × 0.5°)[77] dataset to compile 4 variables describing spatially explicit atmospheric Nr deposition: $NH_x$, $NO_y$, $NH_x$:$NO_y$ and the 10-year average total N deposition for each sample based on specific geographic location. The forest stand parameter total basal area was used as a proxy for forest biomass variation, collected at the time of each tree-core collection in the field and archived in the Swedish NFI database. The observed increase in total basal area over time can be attributed to the combined influence of forest management and environmental change factors[78]. Notably, commercial forestry in the north is relatively recent compared with southern Sweden, which has undergone multiple forest rotations[79].

### Swedish NFI growth data

Data from the NFI are acquired through annual systematic sampling of plots covering Swedish forests. The NFI sampling protocol is based on probability sampling, using a stratified systematic cluster design with a combination of permanent and temporary circular plots. Temporary plots, from where cores are extracted from sample trees, have a 7-m radius, whereas permanent plots have a 10-m radius. Permanent plots are resampled every 5 years and temporary plots are measured once[80]. The survey includes over 100,000 individual tree measurements each year (that is, obtained from about 10,000 survey plots). The growth rate is estimated using different approaches for both permanent and temporary plots. For permanent plots, growth is estimated as the difference in volume estimates (using tree species, diameter at breast height, tree height and crown height) between two consecutive inventories. Growth from temporary plots is estimated via the collection of tree cores, from which radial increment is measured on a microscope, upon which regression models are used to estimate volume growth. The estimated growth rate for permanent and temporary plots is then combined into a weighted mean, with the uncertainty of the estimate ≤1% (ref. 72). For the current study, the total volume growth (that is, the total growth for all of Sweden) for mesic *P. sylvestris*- and *P. abies*-dominated forests, respectively, was acquired from the NFI database.

Within each year and each species, we converted total volume growth to stand-level growth data by dividing the total volume growth by the forest area of the respective species.

## Statistical methods

We analysed data using R Statistical Software[81]. First, we conducted linear regressions of the response variable, $\delta^{15}N$, as a function of time (predictor variable) for each of the two tree species across the geographical regions: north, central, southeast and southwest (Fig. 1c–j); a modified delineation (Fig. 1a) based on regions commonly used to evaluate Swedish environmental quality in relation to acidification and eutrophication[32]. This resulted in sample counts ranging from 18 to 45 samples per region and decade combination by species. In addition, we performed separate follow-up two-way analysis of variance with Tukey's post hoc pairwise comparisons ($\alpha = 0.05$) to identify differences in regression slopes between the four regions for each species (Fig. 1c–j). We checked for normal distribution and homoscedasticity of all model residuals. Next, we constructed a linear mixed-effects model (lme function; nlme package in R[82,83]) using stepwise forwards and backwards selection combined with Akaike information criteria using maximum likelihood to choose the model with the best combination of predictors to explain the observed patterns on the response variable, $\delta^{15}N$. The initial model included fixed effects: atmospheric $CO_2$, mean annual temperature, total Nr deposition, total basal area, stand age and tree species, and the interactions of tree species with $CO_2$ and total Nr deposition with $CO_2$. Numerical fixed effects were centred and scaled using the scale function in R. Grid cell was included as a random effect (intercept) to account for spatial autocorrelation among plots within the same grid. The variance inflation factor (VIF), a check for multi-collinearity of continuous numerical main predictors in the model, was calculated and a VIF limit of five was used to indicate an acceptable level of collinearity (Extended Data Table 1). The final model with the lowest Akaike information criterion retained all variables with the exception of stand age and the interaction of $CO_2$ with total Nr deposition. Finally, we repeated the linear mixed-effects model and replaced total Nr deposition with each of the remaining N deposition variables: $NH_x$, $NO_y$ and $NH_x:NO_y$. The final model was run using restricted maximum likelihood (REML). We calculated the marginal (fixed effects), conditional (fixed and random effects) and partial $R^2$m values for each model (Extended Data Table 1) using the MuMIn[84] package (MuMIn::r.squaredGLMM function) in R. We also ran the final model with two variations: (1) all fixed and random effects of the final model with the addition of latitude and longitude (Extended Data Table 2); and (2) all fixed and random effects from the final model with mean annual temperature replaced by temperature change (Extended Data Table 3). We modelled $\delta^{15}N$ as a function of the significant and non-significant scaled variables (emmeans function) and estimated the slopes of significant model variables (emtrends function) using the emmeans package[85]. Finally, scaled explanatory variables were backtransformed for plotting significant and non-significant main effects (Fig. 2).

To investigate the potential role of forest volume growth change on wood $\delta^{15}N$ values, we calculated an additional forest growth variable describing 'relative forest volume growth change' in each grid cell. This was done by calculating a mean forest volume growth value for all sampling times for each species in each grid cell for which we had $\delta^{15}N$ values and then subtracting this mean value from each of the absolute forest volume growth values for each individual sample plot. We note that forest volume growth change and $\delta^{15}N$ values were based on 5-year and 10-year increments before the sampling date, respectively. We then constructed two linear mixed-effects models where relative forest volume growth change and either absolute forest volume growth or basal area were included as an additional fixed factor, and grid cell was included as a random factor (Extended Data Table 4). Both models were run using REML. We calculated the marginal (fixed effects)

and conditional (fixed and random effects) $R^2$ values for each model using the MuMIn[84] package (MuMIn::r.squaredGLMM function) in R (Extended Data Table 4).

We modelled $\delta^{15}N$ as a function of relative forest volume growth change, including either absolute forest volume growth (Fig. 3c) or absolute basal area (Fig. 3d) as additional fixed effects, while also accounting for the random effect of grid. Slopes of the significant fixed effects were estimated from the models. We conducted a principal component analysis with normalized climate, N, and forest stand parameters that were significant in the final model, as well as the remaining three Nr deposition parameters to provide a visual representation of the impact of the specific parameters (Extended Data Fig. 1). All variables were scaled using the scale function in R. We conducted linear regression of all predictor variables in the final model: atmospheric $CO_2$, mean annual temperature, total N deposition and total basal area, and temperature difference as a function of predictor variables time (average year; Extended Data Fig. 2a,c,e,g,i) and space (latitude; Extended Data Fig. 2b,d,f,h,j). Finally, we conducted linear regressions of response variables: annual volume growth ($m^3 ha^{-1} yr^{-1}$) and %N as a function of time (predictor variable) to determine whether a change through time occurred for each of the two species (*P. sylvestris* and *P. abies*) across the geographical regions: north, central, southeast and southwest (Fig. 3a,b and Extended Data Fig. 3, respectively). To enhance the visual differentiation of overlapping data points, the geom jitter function (ggplot2; ref. 86) was applied in the horizontal plane to Fig. 1c–j, Extended Data Fig. 2a,c,e,g,i, and Extended Data Fig. 3. Graphical representations were made using ggplot2 and FactoMineR[87] in R.

## Data availability

The data supporting the findings of this study are accessible in the figshare repository, available at https://doi.org/10.6084/m9.figshare.30675002 (ref. 88). Atmospheric reactive nitrogen (Nr) deposition data from the Inter-Sectoral Impact Model Intercomparison Project phase 3a (ISIMIP3a) datasets are available at https://www.isimip.org. Temperature data from the CRU TS v4.07 dataset are available at https://crudata.uea.ac.uk/cru/data/hrg/, and atmospheric $CO_2$ data are available from the National Oceanic and Atmospheric Administration (NOAA) Global Monitoring Laboratory.

## Code availability

All R scripts used for data processing and analysis are available in the figshare repository, available at https://doi.org/10.6084/m9.figshare.30675002 (ref. 88).

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

**Acknowledgements** We thank the Swedish NFI field crews (1961–2018) for sample and data collection; and F. Johansson, M. Karlsson, T. Sundvall, P. Edlund and K. Spies for archive assistance. We thank the Central Appalachians Stable Isotope Facility, Maryland Center for Environmental Science Appalachian Laboratory, for laboratory analysis. We also thank F. Maillard for manuscript review. The research was funded by Stiftelsen Gunnar och Birgitta Nordins fond (KSLA) to K.R.B.; Brattås Stiftelsen, Troëdssons Stiftelsen, VR (2020-09308) to M.J.G.; T4F, and the Wallenberg Foundation (2018.0259). Any use of trade, firm, or product names is for descriptive purposes only and does not imply endorsement by the US Government.

**Author contributions** M.J.G. formulated the original concept presented in the paper. K.R.B., M.J.G., S.S.P., S.J., J.F. and L.Ö. contributed to discussions regarding the study design. K.R.B. managed all archival procedures and prepared samples for analysis. K.R.B. conducted all statistical analyses with assistance from S.F.H. in data management and R programming. K.R.B., S.F.H., M.J.G., S.J., L.Ö. and S.S.P. analysed and interpreted the results. K.R.B. authored the initial draft of the paper. All authors engaged in the discussion and revision processes of the paper.

**Funding** Open access funding provided by Swedish University of Agricultural Sciences.

**Competing interests** The authors declare no competing interests.

**Additional information**
**Correspondence and requests for materials** should be addressed to Kelley R. Bassett.

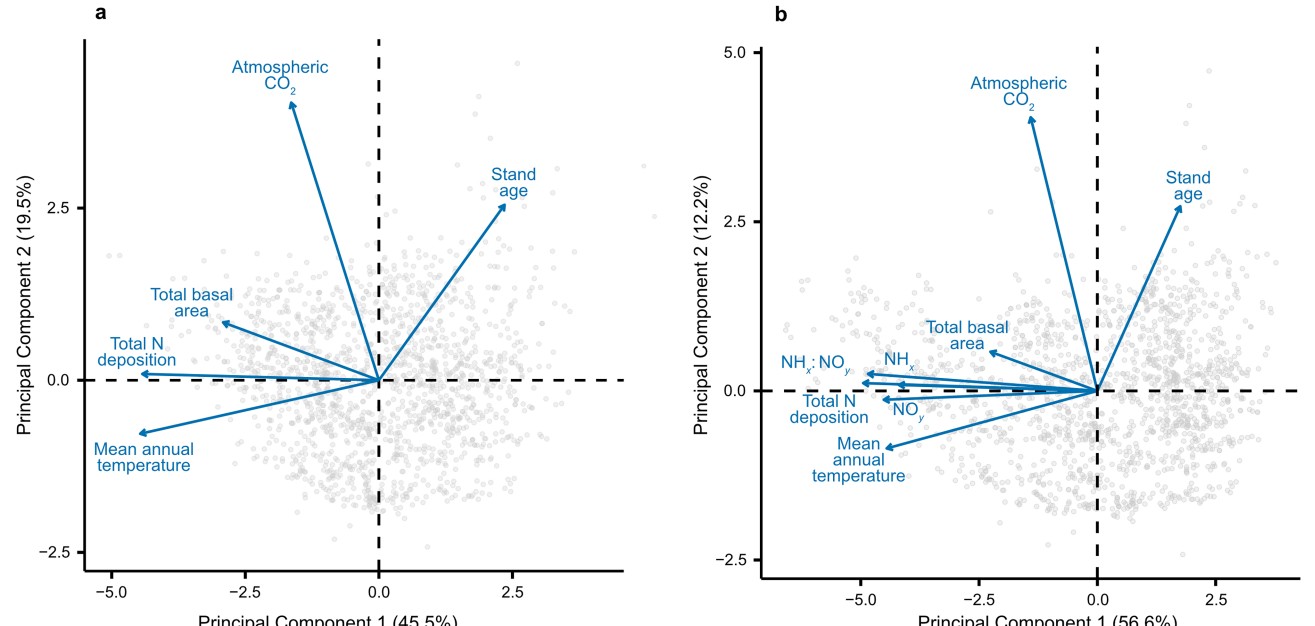

**Extended Data Fig. 1 | Principal Component Analysis (PCA) biplot of scaled variables. a**, Scaled variables: atmospheric carbon dioxide ($CO_2$), mean annual temperature, total basal area, stand age, and total nitrogen (N) deposition; and **b**, all aforementioned variables plus the addition of $NH_x$, $NO_y$, and $NH_x$:$NO_y$.

Individual observations in grey ($n = 1609$) are plotted based on their scores along the first two principal components, which explain 45.5% and 19.5% of the variance, respectively (**a**); and 56.6% and 12.2% of the variance, respectively (**b**). Data sources are cited in the Methods.

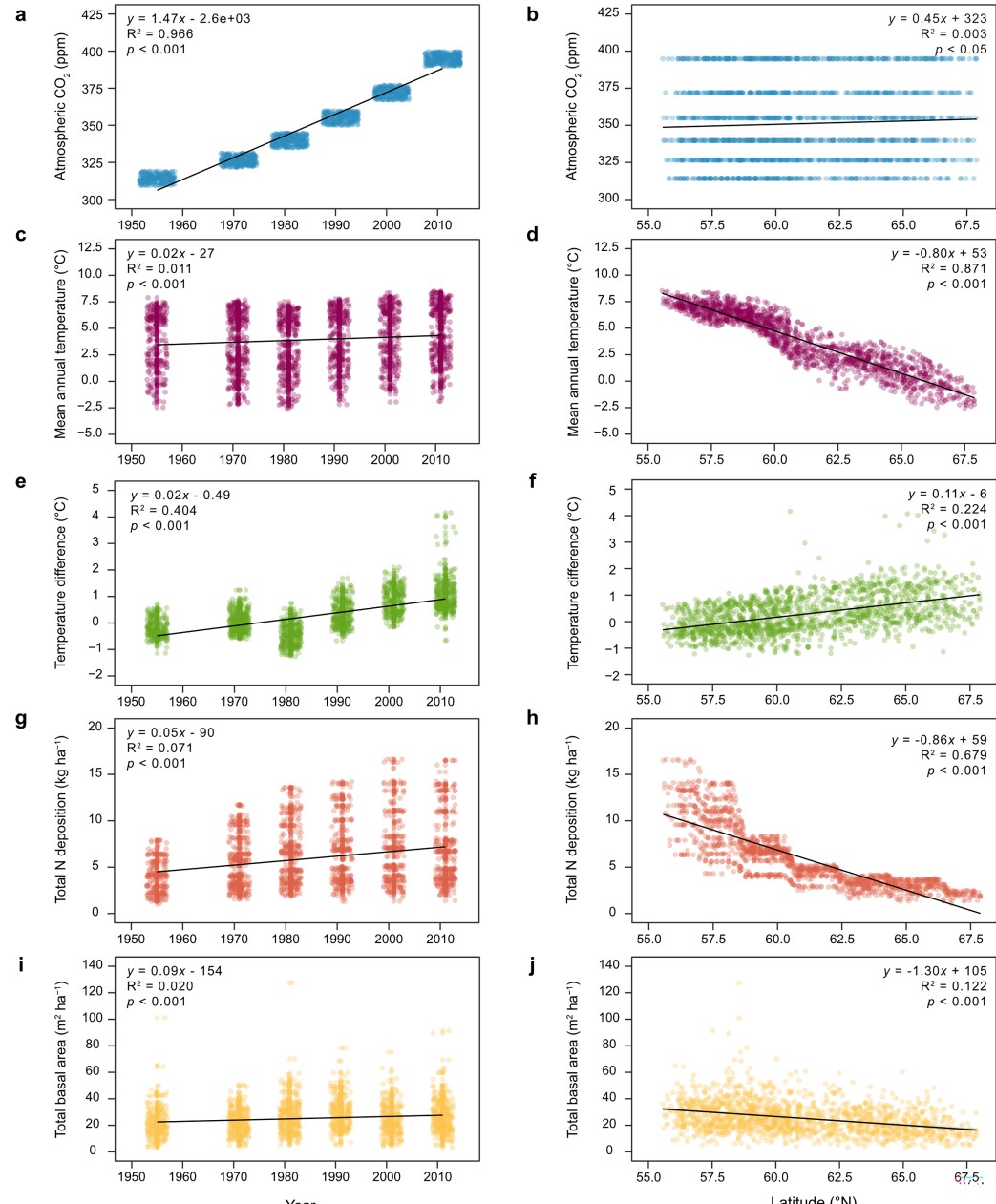

**Extended Data Fig. 2 | Temporal and spatial trends in significant model variables.** Linear regression of atmospheric carbon dioxide ($CO_2$ ppm; **a,b**); mean annual temperature (°C; **c,d**); temperature change (°C; **e,f**); total nitrogen (N) deposition (kg ha$^{-1}$; **g, h**); and total basal area (m$^2$ ha$^{-1}$; **i,j**) plotted over time (average year) and space (latitude), respectively. All relationships are significant ($P < 0.05$). A small horizontal offset was applied to the time plots (**a,c,e,g,i**) to increase the visibility of the data points (see Statistical methods). Data sources are cited in the Methods.

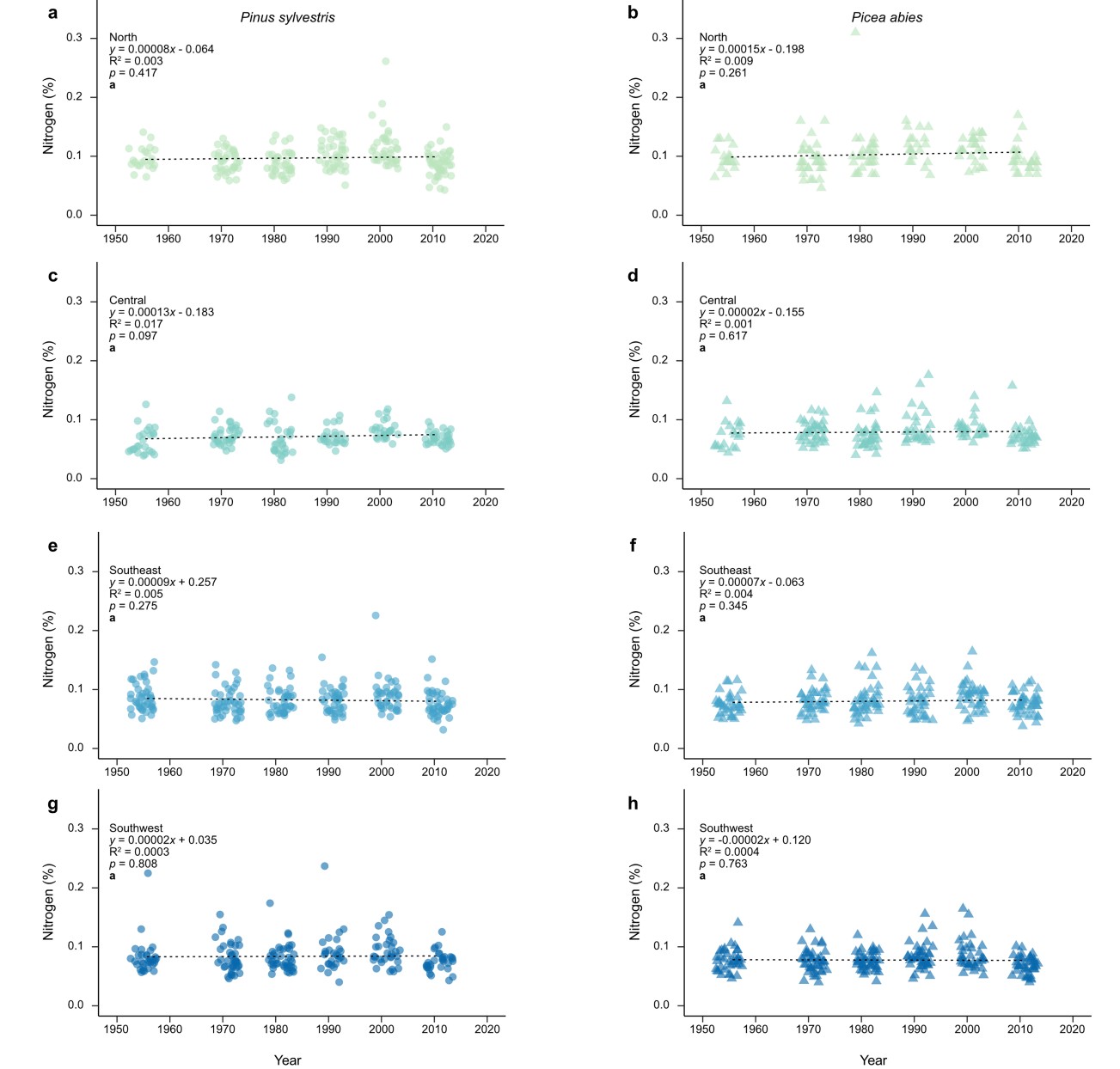

**Extended Data Fig. 3 | Regional and temporal variation in wood nitrogen concentration across Sweden.** Linear regressions of nitrogen (N) % for *P. sylvestris* and *P. abies* by region: north (**a,b**); central (**c,d**); southeast (**e,f**); and southwest (**g,h**) Sweden over the study period, 1950-2017. Slopes indicated with dashed lines were not significant (*P* < 0.05). Data sources are cited in the Methods.

**Extended Data Table 1 | Summarised results of linear mixed-effects models**

**a**

| Variable | SE | df | t-value | *p*-value | VIF | Partial R²m |
|---|---|---|---|---|---|---|
| Atmospheric $CO_2$ | 0.070 | 1403 | -15.828 | **0.0000** | 2.04 | 0.176 |
| Mean annual temperature | 0.093 | 1403 | 2.923 | **0.0035** | 2.86 | 0.006 |
| Total N deposition | 0.094 | 1403 | 2.484 | **0.0131** | 2.99 | 0.004 |
| Total basal area | 0.054 | 1403 | 4.228 | **0.0000** | 1.17 | 0.009 |
| Species | 0.099 | 1403 | 0.130 | 0.8964 | 1.02 | 0.005 |
| $CO_2$:Species | 0.098 | 1403 | 3.456 | **0.0006** | 1.96 | 0.006 |

R²m = 0.214; R²c = 0.241

**b**

| Variable | SE | df | t-value | *p*-value | VIF | Partial R²m |
|---|---|---|---|---|---|---|
| Atmospheric $CO_2$ | 0.070 | 1403 | -15.760 | **0.0000** | 2.04 | 0.174 |
| Mean annual temperature | 0.084 | 1403 | 3.819 | **0.0001** | 2.34 | 0.009 |
| $NH_x$ | 0.084 | 1403 | 2.156 | **0.0312** | 2.38 | 0.004 |
| Total basal area | 0.054 | 1403 | 4.389 | **0.0000** | 1.17 | 0.010 |
| Species | 0.099 | 1403 | 0.119 | 0.9052 | 1.03 | 0.005 |
| $CO_2$:Species | 0.098 | 1403 | 3.442 | **0.0006** | 1.86 | 0.005 |

R²m = 0.213; R²c = 0.240

**c**

| Variable | SE | df | t-value | *p*-value | VIF | Partial R²m |
|---|---|---|---|---|---|---|
| Atmospheric $CO_2$ | 0.069 | 1403 | -15.806 | **0.0000** | 2.00 | 0.174 |
| Mean annual temperature | 0.090 | 1403 | 3.385 | **0.0007** | 2.60 | 0.009 |
| $NO_y$ | 0.089 | 1403 | 2.164 | **0.0307** | 2.71 | 0.003 |
| Total basal area | 0.055 | 1403 | 4.043 | **0.0001** | 1.19 | 0.007 |
| Species | 0.098 | 1403 | 0.165 | 0.8687 | 1.02 | 0.005 |
| $CO_2$:Species | 0.097 | 1403 | 3.518 | **0.0004** | 1.96 | 0.005 |

R²m = 0.212; R²c = 0.245

**d**

| Variable | SE | df | t-value | *p*-value | VIF | Partial R²m |
|---|---|---|---|---|---|---|
| Atmospheric $CO_2$ | 0.069 | 1403 | -15.592 | **0.0000** | 2.00 | 0.171 |
| Mean annual temperature | 0.075 | 1403 | 5.760 | **0.0000** | 1.80 | 0.023 |
| $NH_x$: $NO_y$ ratio | 0.071 | 1403 | 0.380 | 0.7038 | 1.69 | 0.001 |
| Total basal area | 0.054 | 1403 | 4.436 | **0.0000** | 1.17 | 0.010 |
| Species | 0.099 | 1403 | 0.143 | 0.8859 | 1.02 | 0.006 |
| $CO_2$:Species | 0.098 | 1403 | 3.497 | **0.0005** | 1.96 | 0.006 |

R²m = 0.210; R²c = 0.242

Summarised results of linear mixed-effects models with response variable, $\delta^{15}N$, as a function of scaled, fixed main effects of atmospheric carbon dioxide ($CO_2$); mean annual temperature; four nitrogen (N) deposition variables: **a**, Total N deposition; **b**, $NH_x$; **c**, $NO_y$; and **d**, $NH_x$:$NO_y$; total basal area; species; the interaction of atmospheric $CO_2$ and species; and the random effect of grid cell using restricted maximum likelihood (REML). Significant values ($P < 0.05$) in bold. The variance inflation factor (VIF) limit of 5 was used to indicate an acceptable level of collinearity. $R^2$marginal ($R^2$m, fixed effects only) and $R^2$conditional ($R^2$c, fixed and random effects) are reported for each model. The partial $R^2$m values indicate the proportion of variance in the response variable uniquely explained by each respective fixed effect, after accounting for the effects of other variables in the model. Data sources are cited in the Methods.

**Extended Data Table 2 | Summarised results of alternative (2) linear mixed-effects models**

**a**

| Variable | SE | df | t-value | p-value | VIF | Partial R²m |
|---|---|---|---|---|---|---|
| Atmospheric $CO_2$ | 0.082 | 1401 | -13.941 | **0.0000** | 2.88 | 0.096 |
| Mean annual temperature | 0.201 | 1401 | 2.496 | **0.0127** | 13.23 | 0.004 |
| Total N deposition | 0.109 | 1401 | 2.532 | **0.0114** | 3.96 | 0.004 |
| Total basal area | 0.054 | 1401 | 4.145 | **0.0000** | 1.17 | 0.008 |
| Latitude | 0.261 | 1401 | 1.356 | 0.1755 | 21.90 | 0.001 |
| Longitude | 0.091 | 1401 | -1.466 | 0.1428 | 2.72 | 0.001 |
| Species | 0.099 | 1401 | 0.043 | 0.9657 | 1.03 | 0.005 |
| $CO_2$:Species | 0.097 | 1401 | 3.456 | **0.0005** | 1.96 | 0.005 |

R²m = 0.215; R²c = 0.245

**b**

| Variable | SE | df | t-value | p-value | VIF | Partial R²m |
|---|---|---|---|---|---|---|
| Atmospheric $CO_2$ | 0.082 | 1401 | -13.850 | **0.0000** | 2.82 | 0.093 |
| Mean annual temperature | 0.201 | 1401 | 2.505 | **0.0124** | 13.35 | 0.004 |
| $NH_x$ | 0.096 | 1401 | 2.127 | **0.0336** | 3.04 | 0.003 |
| Total basal area | 0.054 | 1401 | 4.324 | **0.0000** | 1.17 | 0.009 |
| Latitude | 0.256 | 1401 | 1.103 | 0.2702 | 21.24 | 0.001 |
| Longitude | 0.091 | 1401 | -1.400 | 0.1617 | 2.72 | 0.001 |
| Species | 0.099 | 1401 | 0.026 | 0.9792 | 1.03 | 0.005 |
| $CO_2$:Species | 0.098 | 1401 | 3.453 | **0.0006** | 1.96 | 0.004 |

R²m = 0.214; R²c = 0.242

**c**

| Variable | SE | df | t-value | p-value | VIF | Partial R²m |
|---|---|---|---|---|---|---|
| Atmospheric $CO_2$ | 0.079 | 1401 | -14.137 | **0.0000** | 2.65 | 0.106 |
| Mean annual temperature | 0.203 | 1401 | 2.308 | **0.0212** | 13.01 | 0.003 |
| $NO_y$ | 0.097 | 1401 | 2.146 | **0.0321** | 3.13 | 0.002 |
| Total basal area | 0.055 | 1401 | 3.959 | **0.0001** | 1.19 | 0.007 |
| Latitude | 0.252 | 1401 | 1.019 | 0.3085 | 19.64 | 0.000 |
| Longitude | 0.093 | 1401 | -1.378 | 0.1685 | 2.70 | 0.001 |
| Species | 0.099 | 1401 | 0.075 | 0.9400 | 1.03 | 0.004 |
| $CO_2$:Species | 0.097 | 1401 | 3.539 | **0.0004** | 1.96 | 0.005 |

R²m = 0.213; R²c = 0.248

**d**

| Variable | SE | df | t-value | p-value | VIF | Partial R²m |
|---|---|---|---|---|---|---|
| Atmospheric $CO_2$ | 0.079 | 1401 | -13.653 | **0.0000** | 2.61 | 0.097 |
| Mean annual temperature | 0.204 | 1401 | 2.288 | **0.0223** | 13.16 | 0.003 |
| $NH_x$: $NO_y$ ratio | 0.077 | 1401 | 0.171 | 0.8642 | 1.95 | 0.000 |
| Total basal area | 0.055 | 1401 | 4.330 | **0.0000** | 1.17 | 0.010 |
| Latitude | 0.250 | 1401 | 0.358 | 0.7204 | 19.35 | 0.000 |
| Longitude | 0.093 | 1401 | -1.213 | 0.2255 | 2.69 | 0.001 |
| Species | 0.099 | 1401 | 0.044 | 0.9649 | 1.02 | 0.006 |
| $CO_2$:Species | 0.098 | 1401 | 3.524 | **0.0004** | 1.96 | 0.006 |

R²m = 0.211; R²c = 0.245

Summarised results of alternative (2) linear mixed-effects models with response variable, $\delta^{15}N$, as a function of scaled, fixed main effects of atmospheric carbon dioxide ($CO_2$), mean annual temperature, total basal area, four nitrogen (N) deposition variables: **a**, Total N deposition; **b**, $NH_x$; **c**, $NO_y$; and **d**, $NH_x$:$NO_y$; latitude, longitude, species, and random effect of grid cell using restricted maximum likelihood (REML). Significant values ($P < 0.05$) in bold. We permitted our variance inflation factor (VIF) to surpass the limit of 5, as we intended this variable to capture spatial variation in the other model factors. $R^2$marginal ($R^2$m, fixed effects only) and $R^2$conditional ($R^2$c, fixed and random effects) are reported for each model. The partial $R^2$m values indicate the proportion of variance in the response variable uniquely explained by each respective fixed effect, after accounting for the effects of other variables in the model. Data sources are cited in the Methods.

**Extended Data Table 3 | Summarised results of alternative (1) linear mixed-effects models**

**a**

| Variable | SE | df | t-value | $p$-value | VIF | Partial $R^2$m |
|---|---|---|---|---|---|---|
| Atmospheric $CO_2$ | 0.093 | 1403 | -14.055 | **0.0000** | 3.67 | 0.108 |
| Temperature change | 0.081 | 1403 | 3.010 | **0.0027** | 2.61 | 0.005 |
| Total N deposition | 0.068 | 1403 | 8.093 | **0.0000** | 1.58 | 0.047 |
| Total basal area | 0.054 | 1403 | 4.943 | **0.0000** | 1.16 | 0.012 |
| Species | 0.099 | 1403 | 0.120 | 0.9041 | 1.02 | 0.006 |
| $CO_2$:Species | 0.098 | 1403 | 3.513 | **0.0005** | 1.96 | 0.006 |

$R^2$m = 0.213; $R^2$c = 0.237

**b**

| Variable | SE | df | t-value | $p$-value | VIF | Partial $R^2$m |
|---|---|---|---|---|---|---|
| Atmospheric $CO_2$ | 0.092 | 1403 | -13.590 | **0.0000** | 3.54 | 0.101 |
| Temperature change | 0.078 | 1403 | 2.127 | **0.0336** | 2.43 | 0.002 |
| $NH_x$ | 0.065 | 1403 | 7.292 | **0.0000** | 1.43 | 0.040 |
| Total basal area | 0.053 | 1403 | 5.560 | **0.0000** | 1.13 | 0.016 |
| Species | 0.099 | 1403 | 0.099 | 0.9215 | 1.02 | 0.005 |
| $CO_2$:Species | 0.098 | 1403 | 3.504 | **0.0005** | 1.96 | 0.006 |

$R^2$m = 0.206; $R^2$c = 0.230

**c**

| Variable | SE | df | t-value | $p$-value | VIF | Partial $R^2$m |
|---|---|---|---|---|---|---|
| Atmospheric $CO_2$ | 0.093 | 1403 | -13.952 | **0.0000** | 3.66 | 0.102 |
| Temperature change | 0.082 | 1403 | 3.150 | **0.0017** | 2.69 | 0.004 |
| $NO_y$ | 0.069 | 1403 | 7.668 | **0.0000** | 1.61 | 0.041 |
| Total basal area | 0.055 | 1403 | 4.523 | **0.0000** | 1.18 | 0.010 |
| Species | 0.098 | 1403 | 0.211 | 0.8328 | 1.02 | 0.006 |
| $CO_2$:Species | 0.097 | 1403 | 3.669 | **0.0003** | 1.96 | 0.006 |

$R^2$m = 0.207; $R^2$c = 0.240

**d**

| Variable | SE | df | t-value | $p$-value | VIF | Partial $R^2$m |
|---|---|---|---|---|---|---|
| Atmospheric $CO_2$ | 0.089 | 1403 | -12.518 | **0.0000** | 3.24 | 0.082 |
| Temperature change | 0.075 | 1403 | 0.472 | 0.6372 | 2.17 | 0.000 |
| $NH_x$:$NO_y$ ratio | 0.059 | 1403 | 4.804 | **0.0000** | 1.17 | 0.021 |
| Total basal area | 0.053 | 1403 | 6.392 | **0.0000** | 1.10 | 0.024 |
| Species | 0.100 | 1403 | 0.119 | 0.9051 | 1.02 | 0.005 |
| $CO_2$:Species | 0.099 | 1403 | 3.595 | **0.0003** | 1.96 | 0.006 |

$R^2$m = 0.187; $R^2$c = 0.218

Summarised results of alternative (1) linear mixed-effects models with response variable, $\delta^{15}N$, as a response to scaled, fixed main effects of atmospheric $CO_2$, temperature change, total basal area, four nitrogen (N) deposition variables: **a**, Total N deposition; **b**, $NH_x$; **c**, $NO_y$; and **d**, $NH_x$:$NO_y$; species, and random effect of grid cell using restricted maximum likelihood (REML). Significant values ($P < 0.05$) in bold. The variance inflation factor (VIF) threshold of 5 was employed to denote an acceptable level of collinearity. $R^2$marginal ($R^2$m, fixed effects only) and $R^2$conditional ($R^2$c, fixed and random effects) are reported for each model. The partial $R^2$m values indicate the proportion of variance in the response variable uniquely explained by each respective fixed effect, after accounting for the effects of other variables in the model. Data sources are cited in the Methods.

## Extended Data Table 4 | Summarised results of linear mixed-effects models

**a**

| Variable | Coefficient | SE | df | t-value | $p$-value | VIF |
|---|---|---|---|---|---|---|
| Absolute forest volume growth | 0.154 | 0.0195 | 1407 | 7.904 | **0.0000** | 2.36 |
| Relative forest volume growth change | -0.163 | 0.0256 | 1407 | -6.351 | **0.0000** | 2.36 |

$R^2$m = 0.044; $R^2$c = 0.075

**b**

| Variable | Coefficient | SE | df | t-value | $p$-value | VIF |
|---|---|---|---|---|---|---|
| Total basal area | 0.039 | 0.039 | 1407 | 6.294 | **0.0000** | 1.73 |
| Relative forest volume growth change | -0.010 | -0.098 | 1407 | -4.497 | **0.0000** | 1.73 |

$R^2$m = 0.027; $R^2$c = 0.080

Summarised results of linear mixed-effects models with response variable, $\delta^{15}N$, as a response to fixed main effects: **a**, absolute volume growth ($m^3$ $ha^{-1}$ $yr^{-1}$), relative forest volume growth change ($m^3$ $ha^{-1}$), and random effect of grid cell; and **b**, total basal area ($m^2$ $ha^{-1}$), relative forest volume growth change ($m^3$ $ha^{-1}$), and random effect of grid cell in relation to response variable, $\delta^{15}N$. Restricted maximum likelihood (REML) was applied to both models. Significant values ($P < 0.05$) in bold, italics. A variance inflation factor (VIF) limit of 5 was used to indicate an acceptable level of collinearity. $R^2$marginal ($R^2$m, fixed effects only) and $R^2$conditional ($R^2$c, fixed and random effects) are reported for each model. Data sources are cited in the Methods.