## [Peer Review File · Nature]

Rising atmospheric CO₂ reduces nitrogen availability in boreal forests

Corresponding Author: Ms Kelley Bassett

Version 0:

Reviewer comments:

Referee #1

(Remarks to the Author)

This manuscript reports on an elegant study that takes advantage of a valuable, national tree core resource collected over decades. The results address the important question of whether rising atmospheric CO₂ will lead to sustained increases in forest growth, or whether decreasing availability of soil nitrogen (N) will ultimately limit potential gains in growth achieved through elevated atmospheric CO₂. Using $\delta^{15}\text{N}$ as a proxy for N availability, the authors show declining $\delta^{15}\text{N}$ values over time in wood from the most recent 10 years of growth in 1609 tree cores from two conifer species collected from the 1950s to the 2010s in four regions of Sweden, spanning 1500 km. The cores were from mesic forests on relatively flat sites and the latitudinal gradient also encompasses a gradient in N deposition. This allowed the authors to separate the effects of CO₂, N form deposition, temperature, stand basal area and species on $\delta^{15}\text{N}$ in stem wood using linear mixed effects models. The models showed that CO₂ was the most significant predictor of $\delta^{15}\text{N}$ values, and had a much stronger effect than N deposition, temperature or basal area.

This work is original and significant. The data set is impressive and the approach has been validated in other studies. The lab in which the isotopic analysis was done has a good reputation. The manuscript is well written, easy to read and the figures are of good quality. The statistics are appropriate and uncertainties are indicated. The abstract and introduction are clear and appropriate, and conclusions are robust. References are appropriate.

This work addresses an important debate over the effects of increasing atmospheric CO₂ on forest nutrition. Some argue that elevated CO₂ will increase photosynthesis, and carbon (C) uptake, leading to greater demand for nutrients, and thus declining nutrient availability over time. For example, the results from this study agree with those of Penuelas et al. (2020), who showed declining concentrations of foliar nutrients in European forests over the last three decades, which they attributed to rising CO₂. Penuelas et al. (2020) argue that future negative feedbacks on C capture can be expected. In contrast, Palmroth et al. (2024) showed sustained sink enhancement in trees subject to elevated CO₂ over 15 years, and argued there was no progressive N limitation in a free-air-enrichment experiment. For the manuscript reviewed, the data was collected from cores spanning a long time period and a geographical range encompassing a gradient in N deposition. This combination provides convincing evidence that N availability has declined in Swedish forests since the 1950s. What is not clear is whether this decline in available N has affected forest growth rates.

A few suggestions:

Line 57 – The soil processes underlying changes in plant and soil $\delta^{15}\text{N}$ values are complex and unfamiliar to many readers. One or two sentences explaining why high N availability favors loss of $\delta^{15}\text{N}$ would be helpful.

Line 74 – As above, a brief explanation of why $\text{NH}_x:\text{NO}_y$ affects $\delta^{15}\text{N}$ would be helpful.

Line 135 – should be -0.763 (negative sign missing)

Line 147 – should be ... changing $\delta^{15}\text{N}$ value of ...

Line 206 – I believe there is little evidence of limited N availability associated with elevated CO₂ leading to reductions in forest growth, to date. Is there evidence of declining forest growth, or is this still hypothetical? Effects of N on growth with elevated CO₂ are complicated by the effects of elevated CO₂ on improved water use efficiency. This confounding effect was not discussed. Perhaps a few sentences of discussion could be added. I wonder if C isotopes were also analyzed from

these wood cores. It would be very interesting to try to separate the effects of N limitation and water use efficiency.

Line 211 – should be ... ectomycorrhizal fungi that mobilize and acquire N ...

Line 221 and throughout – the authors discuss “internal N cycling” but do not define what they mean. To me, internal N cycling is retranslocation within a tree. I understand that ecosystem N cycling is meant, but this should be defined at the outset.

References

Penuelas, J., Fernández-Martínez, M., Vallicrosa, H. et al. Increasing atmospheric CO₂ concentrations correlate with declining nutritional status of European forests. *Commun Biol* 3, 125 (2020). <https://doi.org/10.1038/s42003-020-0839-y>

S. Palmroth, D. Kim, C. A. Maier, D. Medvigy, A. P. Walker, R. Oren. Increased leaf area index and efficiency drive enhanced production under elevated atmospheric [CO₂] in a pine-dominated stand showing no progressive nitrogen limitation. *Glob Change Biol* 30, 2 (2024). <https://doi.org/10.1111/gcb.17190>

Referee #2

(Remarks to the Author)

The manuscript by Bassett et al. investigates the factors driving changes in the nitrogen isotopic composition ($\delta^{15}\text{N}$) in tree stems, identifying atmospheric CO₂ concentrations as the primary driver of the observed $\delta^{15}\text{N}$ decline since the 1960s, while other factors such as N deposition, temperature, and basal area appear to have little or counterbalancing effects. The abstract and the study is well-structured, with a mostly clear experimental design and appropriate statistical analyses (but see my specific response). The writing is also straightforward and easy to follow.

However, I have significant concerns regarding the manuscript's main conclusion that links the $\delta^{15}\text{N}$ decline (caused by CO₂) to increased nitrogen limitation:

1. Interpreting $\delta^{15}\text{N}$ in forest ecosystems is inherently complex, as the processes influencing $\delta^{15}\text{N}$ remain only partially understood. This is particularly evident in the speculative discussion (especially in the "Implications" section) on why atmospheric CO₂ is the primary driver of $\delta^{15}\text{N}$ and nitrogen limitation. The two arguments remain inconclusive about the processes linking $\delta^{15}\text{N}$ and N limitation and instead highlight gaps in our understanding, opening new research questions rather than providing a definitive answer. Furthermore, because this study focuses on trees, I assume that nitrogen limitation refers to a process that restricts nitrogen availability for trees or forest ecosystems. To support this claim, I would expect the authors to present direct evidence linking $\delta^{15}\text{N}$ trends to measurable traits, such as nitrogen concentrations (in different chemical forms) in trees or soils, tree physiology, or growth responses, that would support the conclusion that CO₂ is driving nitrogen limitation in boreal forests.

2. Tree-ring $\delta^{15}\text{N}$ measurements may be influenced by nitrogen mobility, particularly within the sapwood. Previous research (e.g., Tomlinson et al., 2014, DOI: 10.1002/rcm.6897) has shown that nitrogen can be partially remobilized across tree rings, raising concerns about whether $\delta^{15}\text{N}$ values from a given year or period accurately represent the $\delta^{15}\text{N}$ signature of the forest ecosystem at that time. A discussion of this potential limitation is necessary.

3. The manuscript does not sufficiently describe the primary nitrogen sources and isotope fractionation processes that shape tree-ring $\delta^{15}\text{N}$ values. Without this context, it remains unclear to the reader what specific ecological or physiological information $\delta^{15}\text{N}$ in tree rings integrates. Providing a clearer introduction/discussion of these mechanisms would significantly improve the study's transparency and interpretability.

Overall, while the study presents an interesting and well-executed analysis, the interpretation of $\delta^{15}\text{N}$ as a clear indicator of nitrogen limitation requires stronger empirical support and a more nuanced discussion of potential confounding factors

Specific comments:

Line 43: Please clarify what "internal" refers to exactly?

Line 92: Figure S1 presents a PCA, so it is not immediately clear how one can deduce an orthogonal spatio-temporal relationship between atmospheric N_r deposition and eCO₂. Please clarify this statement.

Figure 1B: Since the decrease in wet N deposition in southwest Sweden does not appear to influence $\delta^{15}\text{N}$ patterns differently than at other sites, this could serve as additional evidence that N deposition is not the primary driver of tree-ring $\delta^{15}\text{N}$.

Line 130: Instead of referring to Figure S1, I suggest explicitly listing the co-linear N deposition variables here.

Line 135: Should there be a minus sign for the slope of *P. abies*?

Lines 151–154: Consider adding supporting variables (e.g., t-values from Figure S1) to further illustrate the differences in the relative contributions of CO₂ vs. N deposition traits.

Line 164: The manuscript alternates between CO₂ and eCO₂ when referring to atmospheric CO₂ concentrations. Could the authors clarify the distinction?

Figure S2: I suggest merging Figures S1 and S2 to better emphasize the differences between temperature and CO₂ effects on $\delta^{15}\text{N}$.

Line 374: It is unclear how the number of archived samples explains why some decades are linked to single years and others to three-year periods. It suggests that there is some variability in the length of the tree cores and that the number of individual trees that are used for a specific $\delta^{15}\text{N}$ value of a period/area is variable. Please add some more details.

Statistics: In the case, the selected trees for each period are not independent of each other (i.e. repeated measure), I would suggest implementing the individual tree ID as a random factor in their model. Why do the authors not consider the temporal effect in their model?

Line 395: The connection between the Carbosorp/MgClO₄ traps and sample weighing is not entirely clear. Consider splitting this into two sentences for clarity. Were the wood samples packed in tin capsules?

Line 420: Could the authors provide evidence of ISIMIP3a model uncertainties for their study sites, particularly for the selected N deposition traits? Does the model provide realistic values across all sites in Sweden? Also, does it account for

dry or wet N deposition?

Referee #3

(Remarks to the Author)

This is an impressive dataset, with a well-designed sampling regime. The authors have systematically sampled archived tree cores from across Sweden, over six decades, to look at long-term trends in $\delta^{15}\text{N}$. They are able to show consistent declines over time in two species and in different regions of Sweden. While similar trends have been identified previously, there has been considerable debate in recent literature about the drivers of this trend, and in particular the relative roles of rising CO_2 vs N deposition (Refs 1 – 5). The sampling approach used here provides the opportunity to disentangle these two drivers because patterns of N deposition are quite different in different regions of Sweden, whereas CO_2 increases consistently everywhere. It appears fairly clear from the analysis that N deposition patterns are not a good explanation of the trend, since the same trend is seen throughout Sweden. As a result, this appears to be a well designed study that resolves an outstanding debate in the literature. However, I have a few issues with the novelty and solidity of the conclusions.

It's important to note that some of the data used here appear to have been previously published. Ref 26 tests the methodology for central Sweden. It is referred to in the text as a methods paper, but the text should also have stated that the trend in $\delta^{15}\text{N}$ has already been shown over time for these two species in central Sweden, so the main contribution of this paper is to expand the analysis to all of Sweden, thereby sampling a wider range of N deposition values.

I'm uncomfortable that the link to rising CO_2 is made only via a temporal correlation. Both $\delta^{15}\text{N}$ and CO_2 change monotonically over time. As a result, $\delta^{15}\text{N}$ is correlated with CO_2 – but it would be similarly correlated with any other variable that changed monotonically over time. This cannot be regarded as strong evidence for CO_2 as a driver. There are a number of ways in which the evidence could have been strengthened.

First, the explanation of the mechanisms by which rising CO_2 would reduce $\delta^{15}\text{N}$ was not sufficiently precise. The paper would have been significantly stronger with the support of a conceptual model to explain the proposed mechanism, with some description of what other changes must also be taking place, and an examination of the evidence for those other changes. For example, my understanding of progressive nitrogen limitation is that it is only likely to occur if plant demand for nitrogen has increased because of a stimulation of plant growth. Is there evidence that trees are now growing faster than 50 years ago? That does appear to be the case – they note that “The observed increase in total basal area over time can be attributed to the combined influence of forest management and environmental change factors.” Oddly, however, this is only discussed as a statistical confounder, instead of as a key part of the story.

I was also surprised that $\delta^{15}\text{N}$ was the only wood trait measured – the previous paper (ref 26) also has data on N%, C% and $\delta^{13}\text{C}$, and shows (quite remarkably) an increase in stemwood N% over time, along with a sizeable reduction in C%, from ca 49% to 46%. These trends do not appear to support the idea that CO_2 is driving the reduction in $\delta^{15}\text{N}$, but have not been quantified here. For this paper to have real conceptual novelty – and a solid conclusion that rests on more than a temporal correlation – I would have expected the conceptual model to encompass the likely changes in other wood traits – including these parameters and stem growth rates – with an exploration of what mechanisms are consistent with all observations.

I would also have expected that the study would be informed by large-scale field experiments with CO_2 and warming in Sweden. In particular, this paper explores how stem growth of 40-year-old trees in northern Sweden responds to rising CO_2 with and without additional fertiliser and warming:

Bjarni D. Sigurdsson, Jane L. Medhurst, Göran Wallin, Olafur Eggertsson, Sune Linder, Growth of mature boreal Norway spruce was not affected by elevated $[\text{CO}_2]$ and/or air temperature unless nutrient availability was improved, *Tree Physiology*, Volume 33, Issue 11, November 2013, Pages 1192–1205, <https://doi.org/10.1093/treephys/tpt043>

While this one explores responses of stemwood production to soil warming:

STRÖMGREN, M. and LINDER, S. (2002), Effects of nutrition and soil warming on stemwood production in a boreal Norway spruce stand. *Global Change Biology*, 8: 1194-1204. <https://doi.org/10.1046/j.1365-2486.2002.00546.x>

Some discussion of the role of rising temperatures in altering the N cycle is needed. I note that mean annual temperature is also considered as a predictor in their statistical models, but my suggestion would be to consider spatial and temporal variability in temperature separately. Spatial gradients in temperature are likely to result in long-term changes in soil carbon and nitrogen pools, whereas increases in temperature over the last 50 years can cause shorter-term increases in nitrogen availability via increased decomposition of those pools.

Finally, the conclusions around the implications for the C sink are quite weak. One potential conclusion is that “ecosystem productivity will be increasingly constrained by intensifying N limitation in response to eCO_2 ”. However, if the mechanism is that plants are growing faster because of eCO_2 , and are able to do so with a reduction in N content, an alternative conclusion might be that increasing C sink strength is being achieved despite N limitations – with a fairly different outlook. I'll repeat the importance of using a quantitative conceptual model to explore the mechanisms and potential future trajectories, rather than drawing overly simplistic conclusions. There was a lot of work done in the 1990's and 2000's developing simple C-N ecosystem models that could be applied in this context (e.g. see work by Goran Agren, Ross McMurtrie, Ed Rastetter) so the wheel would not need to be re-invented.

Referee #4

(Remarks to the Author)

Summary: The authors use measurements of the natural abundance of ^{15}N from tree rings carefully selected from across Sweden to show steady decreases in $\delta^{15}\text{N}$ over the last 60 years, regardless of the rate or form of incoming N deposition. Further statistical analyses show that these $\delta^{15}\text{N}$ declines are more strongly linked to increases in atmospheric CO_2 than with any other factor examined (N deposition, rising temperature, stand basal area). The authors attribute these $\delta^{15}\text{N}$ trends to increasing ecosystem N limitation driven by rising atmospheric CO_2 .

Review: I found this manuscript to very convincingly demonstrate that declines in plant $\delta^{15}\text{N}$ over time across Sweden are linked to rising atmospheric CO_2 concentrations rather than to other factors of environmental change, with particularly strong evidence against attribution of these $\delta^{15}\text{N}$ patterns to the rate or form of N deposition. The authors' selection of samples from across Sweden, while also carefully controlling for stand age class, makes this analysis much more robust than the small set of high-profile reviews and data syntheses that it builds on.

The primary explanation proposed as the mechanism driving these ^{15}N trends in this and these other studies is the stimulation of N limitation by elevated CO_2 ($e\text{CO}_2$), building on a framework originally described in the literature as progressive N limitation (PNL, ref. 14 – Luo et al., 2004) and more recently as terrestrial oligotrophication (e.g., refs 1-Craine et al., 2018 Nature EE & 3-Mason et al., 2022 Science). The long-proposed ideas suggest that $e\text{CO}_2$ should drive ecosystem N limitation first by stimulating plant net primary production and accumulating N in plant biomass, and second, by increasing plant litter C:N ratios that should increase soil N accumulation, both with feedbacks that should further enhance N limitation. This manuscript does a nice job of reviewing these general ideas (line 195-202) and should be sure to include references to the earlier PNL framework. It also discusses a potential role for increased ^{15}N fractionation by ectomycorrhizae in contributing to the ^{15}N decline. Overall, this manuscript makes a solid contribution to the literature that improves on these prior assessments; it contains a well-crafted set of measurements and is nicely written.

I think that this manuscript could even go a step further, for greater impact with a small amount of additional text critically evaluating and discussing the potential processes underlying the ^{15}N response to $e\text{CO}_2$ and its attribution to increased N limitation. First, the text can directly assess the first part of the N limitation hypotheses, pertaining to whether $e\text{CO}_2$ -stimulation of plant accumulation of N is really likely to be driving N limitation and the observed decrease in $\delta^{15}\text{N}$. If this mechanism is an important driver, tree-ring $\delta^{15}\text{N}$ should have a fairly strong negative relationship with stand biomass (or basal area, as used here). However, analyses here suggest a weak positive relationship (Fig. S2), which casts doubt on that explanation. Second, the manuscript should also acknowledge and discuss an alternative set of mechanisms by which $e\text{CO}_2$ has been proposed to affect plant N demand, tied to down-regulation of plant N uptake by $e\text{CO}_2$ (e.g., ref. 2-Hiltbrunner et al. 2019, references therein). These mechanisms could perhaps more directly and universally decrease plant N uptake in response to rising CO_2 , regardless of ecosystem N limitation or availability – as seems to be the case here, as Fig. 1c-j and 2b shows little to no effect of N inputs on the tree-ring $\delta^{15}\text{N}$ values.

One important technical question: as detailed below (see comment re. line 134-135), the slopes of relationships between $\delta^{15}\text{N}$ and CO_2 concentration don't seem to make sense or to match Fig. 2a. I'm hoping that the authors can correct or clarify that discrepancy.

Detailed comments:

Line 42. Add a subject after "This ___".

Line 59. This explanation of the relationship between $\delta^{15}\text{N}$ and ecosystem N availability is useful as a broad description of one important set of processes that affect plant $\delta^{15}\text{N}$ (i.e. preferential loss of ^{14}N under conditions of elevated N availability), but it is also somewhat overly simplistic, and neglects the roles of other processes that could be relevant here – especially those pertaining to ^{15}N fractionation on N uptake by mycorrhizae or other N uptake and translocation processes. I appreciate the difficulty of trying to succinctly explain the many factors that can affect plant $\delta^{15}\text{N}$ in the Introduction, but brief acknowledgment is needed here that other factors also affect plant $\delta^{15}\text{N}$.

Line 61-63. Reports of others' findings of decreasing $\delta^{15}\text{N}$ over time would be more informative if presented in more directly comparable units – preferably as rates of decline (per mil) per year or per decade. Doing so would also make these $\delta^{15}\text{N}$ declines more readily comparable with those reported in in this study in line 111-112.

Line 63. Avoid using a dash as both negative and range: "decline of $\delta^{15}\text{N}$ globally (-0.06 to -1.61 per mil..." rather than "(0.06-1.6 per mil"

Line 83-84. State the time period covered by this study somewhere in this section.

Line 96-105. I applaud the authors for their structured sampling approach to control for tree species, location/N deposition history, and tree age class. This approach is a particulate strength of this analysis, and makes it much more robust and convincing than some of the predecessors that it reviews.

Line 114-117. The fact that the slopes of ^{15}N values over time did not differ among regions with very different N deposition rates does indeed appear to support the authors' conclusion that the ^{15}N temporal trends are not driven by temporal patterns

of N deposition.

Line 114-117, 144-150. However, the very similar $\delta^{15}\text{N}$ values and slopes across these regions receiving vastly different amounts of N deposition (Fig. 1c-j) would also seem to also suggest very direct little relationship between $\delta^{15}\text{N}$ and N availability in these forests (confirmed by Fig. 2b). Can the eCO_2 -associated $\delta^{15}\text{N}$ decline be so confidently ascribed to eCO_2 -driven increased N limitation when these relationships with N inputs are so poor?

Line 134-135. Please include the units for the slopes reported here in the text (-0.76 and -1.1) associated with Fig. 2, describing the relationships between tree-ring $\delta^{15}\text{N}$ and CO_2 concentrations. These slope units are presumably ‰ $\delta^{15}\text{N}$ per ppm CO_2 to match the units of the axes in Fig. 2a. However, slopes that steep don't match the $\delta^{15}\text{N}$ values in Fig. 2a or make sense. That is, for the CO_2 range of ~70 ppm shown in Fig. 2a (from roughly 315 to 385 ppm), these slopes should yield implausibly large $\delta^{15}\text{N}$ declines of -53 and -77 per mil $\delta^{15}\text{N}$, while the declines shown are the more reasonable range of < -6 per mil.

Line 159-163, 196-198. The observation reported here of a positive relationship between $\delta^{15}\text{N}$ and basal area (a proxy for biomass) is interesting, but hard to reconcile with the later partial attribution of the $\delta^{15}\text{N}$ trends over time to eCO_2 stimulation N accumulation in plants (lines 196-198). That is, if the mechanism by which eCO_2 is causing $\delta^{15}\text{N}$ to decrease is that increased accumulation of N in plant biomass reduces N availability, then the $\delta^{15}\text{N}$ -biomass relationship should not be positive but strongly negative. The authors might use this information to deepen the discussion of this hypothesis rather than suggesting it as a research part of the research frontier (line 233).

Line 195-202. This section nicely reviews the two proposed mechanisms by which eCO_2 could stimulate N limitation (by increasing N accumulation in plant biomass through stimulated production, and increased C:N ratios of plant tissues driving soil N accumulation) that were clearly laid out several decades ago as part of the progressive N limitation framework (ref. 14, Luo et al., 2004). This attribution should be added.

Line 216-219, 223-226. Although some ectomycorrhizae can mine N from SOM, increased plant reliance on ectomycorrhizae for N uptake (with corresponding $\delta^{15}\text{N}$ fractionation) doesn't necessarily mean that those mycorrhizae are acquiring more N from SOM, only that they are responsible for a larger fraction of plant N.

Line 229-230. In the cited meta-analysis, it's true that soil C "did not increase" in eCO_2 experiments with ectomycorrhizal plants, but soil C stocks didn't decrease, either (no change on average), although a decrease is implied by the suggestion (line 230) that eCO_2 is stimulating decomposition in these forests.

Line 333-337. Somewhere in this section, state the sample size (or size ranges) for the species / decade / region combinations. If $n = 1609$ overall, and there are 2 species, 6 decades, and 4 regions, that would seem to be ~33 samples each on average?

Line 393-395. The 10-year segments of wood would generally need to be ground before weighing 10 mg subsamples. If so, please describe how; if not, please state how the 10 mg representative samples were obtained.

Line 409, 418. Specify the spatial resolutions of the gridded temperature and N deposition datasets.

Line 424, 425. "The observed increase in forest basal area over time can be attributed to..." Observed and attributed by whom? Please add citation/s.

Line 531-532. "Data sources are cited in Supplemental Information." Rephrase. This text is in the SI.

Table S1: Please convert table headers to text from what looks like model commands.

Figure 2. The slope values stated on these figures appear inconsistent with the regression lines plotted. Please see comment above re. line 134-135.

Version 1:

Reviewer comments:

Referee #1

(Remarks to the Author)

The authors have done a very thorough job of responding to the questions and concerns of four reviewers. For my part, I am satisfied with their response to my queries and concerns. I appreciate the thoughtfulness of the authors' responses. I believe this manuscript content is now acceptable.

I have a few very minor editorial suggestions.

L 32 – expected to increase by a factor of 18

L 61 – delete the second "with low $\delta^{15}\text{N}$ values"

L 63 – reference 21 does not need brackets

L 78, 93 – "a study" should be replaced by the author names

L 227 – “including” should be replaced by “and”
L 240 – change to ... , and thus steer ...
L 245 – “In recent decades, forest growth may have levelled off...”. There is no analysis of this trend and it is not always clear, especially in northern regions and for *Picea abies*.
L 271 – reference 61 does not need brackets
L 279 – change to “to the control of $\delta^{15}\text{N}$ values ...”
L 280 - compartments

Referee #2

(Remarks to the Author)

The authors have comprehensively addressed all of my comments from the previous version of the manuscript and I recommend acceptance of the manuscript.

Referee #3

(Remarks to the Author)

The authors have done an excellent job of responding to the reviewers' comments. The revisions have strengthened the paper considerably and I now find this to be a very novel and convincing piece of work. I have no further suggestions for edits to the manuscript.

I've also taken a look at the data shared. This appears to be complete but lacks meta-data. I would strongly recommend adding some meta-data to indicate what the data are (would include an explanation of the contents of each sheet, including explanation of each column with units).

Referee #4

(Remarks to the Author)

I was reviewer #4 for the previously submitted version of this manuscript, so I will not repeat my prior summary of the manuscript, which has changed relatively little from the previous version.

As in my prior review, I am convinced by the first part of the authors' conclusion, that the study elegantly shows that elevated (eCO₂) is the main driver of decreases in wood N isotopic composition ($\delta^{15}\text{N}$) in trees across Sweden over the last ~60 years. Both versions of this manuscript nicely address earlier critiques of a set of high-profile analyses (mainly by a recurring set of authors; manuscript refs. #s 1, 3, 4, 20, 21, 22, 24, 43, 44), that have made similar arguments in interpreting other plant $\delta^{15}\text{N}$ measurements, but did not previously disentangle the role of eCO₂ from that of nitrogen deposition amount or composition as potential drivers of these $\delta^{15}\text{N}$ trends, as the present analyses does very effectively. Resolving that debate and confirming a primary role of eCO₂ in driving plant $\delta^{15}\text{N}$ trends makes a solid contribution to the literature, and I looked forward to citing those results.

That said, I was disappointed by the lack of response to my main content-based requests of the prior manuscript, and I am not yet convinced by the second part of the manuscript's conclusion as summarized in its title, that these eCO₂-driven $\delta^{15}\text{N}$ trends necessarily mean that eCO₂ has intensified N limitation. It could be so, but I don't believe that the manuscript has yet demonstrated intensified N limitation of plant growth. As reviewers are asked to review statistics: I believe that this part of the conclusion also warrants support from statistical analysis rather than the qualitative support included so far.

The manuscript's conclusion is framed in the context of the longstanding hypothesis in ecosystem ecology of progressive N limitation (PNL) that is also embedded in most earth system models (ESMs), and expects that eCO₂ should induce terrestrial N limitation by stimulating accumulation of N in plant biomass and/or soil organic matter. The manuscript's conclusions assume a direct relationship between plant $\delta^{15}\text{N}$ and plant N limitation, and although correlations between plant N concentration and $\delta^{15}\text{N}$ do regularly occur, these relationships are far noisier and more complicated than portrayed in the text or in the authors' Response (e.g., see comments re. lines 71-73 below). The disconnect between $\delta^{15}\text{N}$ and plant N responses is further illustrated by the strong responses of $\delta^{15}\text{N}$ shown over time (Fig. 1) that does not occur in tree-ring %N (Fig. S4), which shows no declines over time in any species or region, a relevant result that the main text does not mention (Fig. S4) and would seem to contradict the manuscript's conclusion that the observed decline in wood $\delta^{15}\text{N}$ is due to increased N limitation.

The authors have now added some tree growth data (Fig. S3) that seemingly could be used (in addition to or instead of basal area) to test the 'enhanced growth' and/or 'growth limitation' part of the PNL hypothesis, but indicate in their Response that adding these data to the overall analysis of $\delta^{15}\text{N}$ drivers would be beyond the scope of the present study. I believe that choice reduces the study's overall impact and the strength of its conclusions. In the manuscript text (line 239-247) and their response, the authors argue that the independent tree growth trend analysis they have added (Fig. S3) support their interpretation of PNL, because these data do indeed show growth increases over time for both species. But in order to support the manuscript's interpretation of PNL-driven N limitation, there should be a primary, direct relationship between the increase in growth (or basal area) and resulting decrease in N availability, here as inferred from $\delta^{15}\text{N}$, and that that relationship should be strong at both the local (plot) and regional scale if it is the main underlying mechanism by which eCO₂ induces N limitation. Without including that direct analysis, the comparison between Fig. 1 and Fig S2i or Fig S3 is qualitative and subjective. It appears that the timing and magnitude of growth stimulation in Fig. S3 varies greatly by region

and tree species, in contrast to the uniformity of the decline in d15N across regions highlighted by the manuscript. It may have been beyond the scope of the author's original intent, yet such analysis would provide the powerful test of the longstanding PNL hypothesis and ESM assumptions that the author's title rests on, and may well make subsequent conclusions of high impact -- whether they support the PNL ideas as presently stated in the title or point to the importance of other mechanisms.

The other prominent mechanism for eCO₂-driven suppression of plant N uptake comes from plant physiology, which routinely shows that eCO₂ allows for greater photosynthetic N use efficiency, allowing the same or even greater photosynthesis in response to eCO₂ accompanied by reduced N uptake (e.g., Stocker et al., 2025, *New Phytologist* 245:49-68 and references therein provide a nice new review), and that appears to be a strong and relatively direct effect of eCO₂ on plant d15N that manifests regardless of plant functional type (ref. 45, BassiriRad et al., 2003), though I believe that it is unclear whether that increased fractionation occurs on uptake or within the plant. Downregulation of N uptake and PNL are not mutually exclusive drivers of plant N responses to eCO₂, but they have fundamentally different implications for the future of the terrestrial C sink and its potential constraint by N limitation, which is expected by PNL, but not from downregulation of plant N uptake. At the very least this alternative explanation merits greater recognition and discussion in the text, as requested by me and several other reviewers.

Detailed comments:

Title: As noted above and in my prior review: so far the manuscript convincingly demonstrates that eCO₂ affects plant d15N, but not that eCO₂ is intensifying N limitation of plant growth.

Line 60-61: typo -- "at low d15N values" is repeated twice in this sentence

Line 66: rephrase "only small isotope discrimination"

Line 69: I suggest adding "can" before "steer plant d15N"

Line 71-73: The relationships among plant d15N, foliar N, and [soil] N availability have been asserted by the refs cited (1 – Craine et al., 2018; 22 – Craine et al., 2009; 23 – Chen et al., 2025; 20 Gerhart & McLaughlin 2014), as well as various others. But these relationships are not nearly as clean and tightly coupled as implied by this text. These variables do frequently emerge as significantly correlated, but often rather weakly. For example, the two figures from refs. 1 & 23 provided by the authors (p. 8 of their Response) illustrate relationships between foliar d15N and foliar N in two data syntheses as very large clouds of data points (Ref. 1 R² = ?; Ref. 23 R² = 0.13). Many factors affect plant d15N values in addition to these particular drivers. Greater recognition of the uncertainty and additional drivers of plant d15N is warranted here and throughout the manuscript.

Line 123-125: This new sentence, asserting that focusing on wood d15N rather than foliar N concentration avoids effects of downregulation doesn't make sense on several levels. Firstly, earlier in the manuscript, the authors state that plant d15N is strongly associated with plant tissue N concentrations, such that wood d15N acts as a reliable index of plant and soil N availability (lines 71-73). Although I don't think those relationships are as tight as stated (above), that is the logical framework provided in the manuscript so far, such that potential decoupling of these variables here requires more explanation and justification. Secondly, if downregulation – which should be described more explicitly in the text in general – was reducing plant N uptake, this process may directly reduce both plant d15N as well as plant N concentration (e.g., ref. 45, the BassiriRad et al., 2003 meta-analysis showing consistent reductions in plant d15N across plant functional types in response to eCO₂ experiments, with no relationship to soil d15N).

Fig. 1-i should use a dashed line rather than solid (or none at all), as this relationship was not statistically significant.

Line 233 (and lines 254, 279): An additional important mechanism linking eCO₂ and d15N is down-regulation of plant N uptake. That process could be acting in addition to or separately from PNL, but has very different implications for the future in terms of N limitation of future plant growth. It deserves explicit recognition and discussion in the main text.

Line 241-245: This added tree growth data is potentially important for making a potential case for a PNL-driven mechanism for the observed decline in wood d15N. I encourage moving these data from the supplement (Fig. S3) to the main text, and as noted previously, I believe that they should be used to directly test, if possible, to what extent the observed d15N trends are associated with changes in wood production in the overall model, as assumed by PNL theory and interpreted by the manuscript.

Line 282. Rephrase "ESM models" (the M = "model")

Line 499. Is the spatial resolution of the temperature data really 5 degrees, as stated? If so, that resolution would appear to be too coarse for use in this application.

Line 582. This text refers to "Supplementary Figure 4, 5", but there is no Fig. S5 in the materials I received for review.

Fig. S2 caption: basal area units are in m² (not m³)

Fig. S2: These panels are interesting but would be more informative if they included information on the slopes and strengths of the plotted relationships.

Version 2:

Reviewer comments:

Referee #4

(Remarks to the Author)

From the start, this manuscript has elegantly and convincingly demonstrated that rising atmospheric CO₂ concentration is the primary driver behind decreasing wood δ¹⁵N trends over nearly seven decades in the boreal forests across Sweden, rather than resulting from shifting amounts or forms of nitrogen deposition as others have proposed. I believe these analyses would be widely accepted as providing evidence that rising CO₂ is increasing N scarcity or “N oligotrophication” in these forests. This insight on its own is an important contribution to the literature, and I again support publication of the analyses.

My primary reservation all along has not been about those analyses, but the declaration that the study demonstrates “N limitation” of boreal forest tree growth, in the title and a handful of key times in the Discussion. It’s the reason why analyses of growth were needed. N limitation is indeed possible and even a likely response, but it is another step beyond what is shown by the tree-ring δ¹⁵N or the new δ¹⁵N v growth analyses (more below). The manuscript makes a solid case for showing indications of decreasing N availability, yet there is a wealth of active research on the extent to which rising CO₂ could allow plants to partly offset that decrease in N availability by, e.g., by fueling more mycorrhizal-mediated N acquisition or by increasing plant nutrient use efficiency. (Here I’ll note that I think the authors have now sufficiently addressed the recently prominent, if – I agree – also hard to test, idea of plant downregulation of leaf N). I believe the distinction of “N limitation” vs “reduced N availability” (or similar term) matters here because it’s the “limitation” aspect of these changes to the N cycle that is so essential for predicting the future of the terrestrial carbon sink and the extent to which it will be constrained by nutrient availability.

This revised version of the manuscript includes an additional set of analyses that provide novel and powerful support for the first part of the longstanding progressive N limitation hypothesis, described well in the manuscript as its main conceptual underpinning to explain its CO₂-δ¹⁵N trends. However, that first part of PNL pertains to growth stimulation by CO₂, and how that should decrease N availability; support for next part of the hypothesis pertaining to N limitation in response to that N scarcity are qualitative and indirect – but tantalizingly close at hand. That is, PNL proposes that (1) increasing atmospheric CO₂ should initially stimulate plant growth and plant demand for nutrients, and that this demand (or increased tissue C:N ratios) should reduce soil nutrient availability; and then (2), this reduced soil nutrient availability should limit further CO₂-stimulation of growth. The new analyses (Figs. 3c & 3d), appear to show that across the whole period of record, there is a discernable trend that locations and times with the largest increases in tree growth have the lightest δ¹⁵N values, providing exciting new support for the PNL hypothesis that increased growth could reduce soil N availability, and has done so in these forests. If I’ve understood them correctly, these new analyses provide a substantial and important advance beyond other high-profile “plant δ¹⁵N trend” papers by showing at least part of why they occur. That said, this new evidence is for the increased growth phase of PNL; not its “limitation” phase.

(In fact, one alternative interpretation of the new Figs 3c & 3d, in reversing their x & y axes, is that isotopically light wood δ¹⁵N correlates with increased rather than flat or declining growth rates as expected for N limitation – presumably because the “stimulation” phase likely dominates most of the period of record of this study.)

Overall, this is an outstanding set of analyses, addressing very important questions pertaining to the future of the terrestrial C sink, and I ultimately support its publication. I applaud the authors for their excellent, well-crafted and insightful analyses, and I extend an appreciation for the additional hard work and intriguing outcome of the δ¹⁵N-growth trend analyses.

I suggest that the authors and editors consider two possible alternatives for publication:

1) Instead of “N limitation” in the title, consider a term such as “N oligotrophication” “decreased N availability” or similar. The authors could also readily speculate that their evidence of reduced N availability should limit additional growth enhancements from further rises in CO₂ as a logical predicted outcome. Yet declaring “Rising atmospheric CO₂ intensifies N limitation in boreal forests” still seems overstated.

2) Consider revising the new “δ¹⁵N v growth change” analyses (Fig. 3c & 3d, Table S4) to consider if or how the δ¹⁵N-growth change relationships differ over time, from the expected early stimulation to later plateau or decline. That might be to break up the analyses by decade or longer, or into two periods, before and after the increased growth trends flatten in Fig. 3a & 3b. If PNL and plant δ¹⁵N act as hypothesized, there should be even stronger and clearer negative relationships between growth changes and wood δ¹⁵N changes in the initial “stimulation” phase, but then those relationships should fall apart or even reverse during a later period of “N limitation.” Linking the δ¹⁵N measurements to both the “stimulation” and “limitation” growth phases of PNL would not only link δ¹⁵N trends to rising CO₂ but demonstrate how they provide long-sought field evidence of CO₂-driven N limitation of growth, and confirm some of its driving plant-soil mechanisms, which would be a spectacular accomplishment and major break-through.

Please clarify a few details in the new Fig. 3c & 3d that will help convey how the analyses were done.

- Isn't there a time component for the "relative forest volume growth change" values on the x-axes? Either as a difference in growth rates from the overall mean (in $\text{m}^3 \text{ha}^{-1} \text{yr}^{-1}$) as the text seems to describe, or as a cumulative $\text{m}^3 \text{ha}^{-1}$ difference over some time period (if so, which period)?
- The $\delta^{15}\text{N}$ values on the y-axes are those measured for 5-year periods corresponding to each "relative forest growth change" value, correct?

REFEREE #1 (REMARKS TO THE AUTHOR):

Comment: This manuscript reports on an elegant study that takes advantage of a valuable, national tree core resource collected over decades. The results address the important question of whether rising atmospheric CO₂ will lead to sustained increases in forest growth, or whether decreasing availability of soil nitrogen (N) will ultimately limit potential gains in growth achieved through elevated atmospheric CO₂. Using $\delta^{15}\text{N}$ as a proxy for N availability, the authors show declining $\delta^{15}\text{N}$ values over time in wood from the most recent 10 years of growth in 1609 tree cores from two conifer species collected from the 1950s to the 2010s in four regions of Sweden, spanning 1500 km. The cores were from mesic forests on relatively flat sites and the latitudinal gradient also encompasses a gradient in N deposition. This allowed the authors to separate the effects of CO₂, N form deposition, temperature, stand basal area and species on $\delta^{15}\text{N}$ in stem wood using linear mixed effects models. The models showed that CO₂ was the most significant predictor of $\delta^{15}\text{N}$ values, and had a much stronger effect than N deposition, temperature or basal area.

This work is original and significant. The data set is impressive and the approach has been validated in other studies. The lab in which the isotopic analysis was done has a good reputation. The manuscript is well written, easy to read and the figures are of good quality. The statistics are appropriate and uncertainties are indicated. The abstract and introduction are clear and appropriate, and conclusions are robust. References are appropriate.

Response: We appreciate the summary and positive assessment of our work.

Comment: This work addresses an important debate over the effects of increasing atmospheric CO₂ on forest nutrition. Some argue that elevated CO₂ will increase photosynthesis, and carbon (C) uptake, leading to greater demand for nutrients, and thus declining nutrient availability over time. For example, the results from this study agree with those of Penuelas et al. (2020), who showed declining concentrations of foliar nutrients in European forests over the last three decades, which they attributed to rising CO₂. Penuelas et al. (2020) argue that future negative feedbacks on C capture can be expected. In contrast, Palmroth et al. (2024) showed sustained sink enhancement in trees subject to elevated CO₂ over 15 years and argued there was no progressive N limitation in a free-air-enrichment experiment. For the manuscript reviewed, the data was collected from cores spanning a long time period and a geographical range encompassing a gradient in N deposition. This combination provides convincing evidence that N availability has declined in Swedish forests since the 1950s. What is not clear is whether this decline in available N has affected forest growth rates.

Response: We agree with the reviewer that our data set provides convincing evidence that nitrogen (N) limitation is intensifying, and that high spatial and temporal resolution is a major strength of our study compared to previous efforts. We thank the reviewer for highlighting these very relevant references. We have now cited the Penuelas et al. (2020) reference at line 83, as well as line 226. We note that Penuelas et al. (2020) had very good replication in central and southern Europe, but presented very little data for boreal forests, which is a novel aspect of our data.

Regarding the Palmroth et al. (2024) reference, it is also a very important and relevant study, but we note that this free-air-enrichment experiment (FACE) experiment was conducted in the temperate region of Eastern North America, rather than in a boreal forest, which differs substantially in climate, baseline N limitation, and productivity. Further, while this study

concluded eCO₂ treatments were not causing progressive nitrogen limitation (PNL) of tree growth within the timeframe of the experiment, other eCO₂ experiments in the same region of the USA have shown that eCO₂-induced PNL eventually constrains forest growth (See Norby et al. 2010, PNAS). Interestingly, these are ectomycorrhizal and endomycorrhizal forests, respectively, and we suspect the different emergence of PNL on growth at the two study sites is due to different capacities of these mycorrhizal types to acquire organic N from soils, when inorganic N supplies are depleted. This was the basis for our discussion around the Terrer et al. (2021) reference in our Implications section, and our discussion on ectomycorrhizal decomposers. In short, rising CO₂ in ectomycorrhizal forests may “prime” ectomycorrhizal decomposer activity, thus tempering nutrient constraints on growth (at least in the short-term), while endomycorrhizal forests that lack mycorrhizal decomposers may be more likely to experience nutrient constraints on growth. We now include the Palmroth et al. (2024) reference in this discussion, at line 271. In this discussion, we also introduce long-term forest growth data derived from the Swedish National Forest Inventory, which we think are relevant in relation to discussions around these two temperate forest FACE experiments.

Regarding boreal forests specifically, eCO₂ experiments are unfortunately very rare, and we are aware of only three such studies. The first study, the SPRUCE experiment, is currently running in a Minnesota peatland; whereas, two previous eCO₂ experiments applied to individual trees were conducted more than 20 years ago in Swedish mesic spruce forests comparable to the forest plots selected for our study (established by a colleague at our same institute, Sune Linder). The first of these Swedish experiments crossed eCO₂ treatments with N fertilizer (n=3), whereas the second experiment crossed eCO₂ with air warming treatments (n=3). Sigurdsson et al. (2013) summarized the growth response of these two experiments and concluded that tree growth responses to eCO₂ were constrained by soil N availability (Sigurdsson et al. 2013). This conclusion is the core mechanism of the PNL hypothesis, i.e. that N limitation constrains the forest growth response to eCO₂. We have now referred to the Sigurdsson et al. (2013) study that summarizes these two experiments, lines 248-250.

Regarding quantifying the effect of progressive N limitation on forest growth, this was actually not a goal of our study, which is why we discussed this aspect in the Implications section, rather than in either of the two Results focused sub-sections. There is broad literature on N limitation and carbon (C) sink strength in boreal forests (e.g. from N fertilizer experiments, Blaško et al. 2022; Gundale et al. 2014; Schulte-Uebbing et al. 2021). Today, a majority of Earth System Models (ESMs) consider nutrient limitation to be an important feedback mechanism that controls/constrains ecosystem C dynamics (e.g. see reviews by Davies-Barnard et al. 2021; Thomas et al. 2015). In the current IPCC assessment, 6 of 11 CMIP6 models that are used to forecast C cycle and climate projections rely upon N cycle feedbacks such as predicted by the PNL framework. These six models on average project 25-30% lower terrestrial C uptake compared to models without N limitation constraints (Section 5.4, Chapter 5, 6th IPCC Assessment Report, 2021). Thus, we feel that the breakthrough of our study is in separating CO₂ from N deposition as the main driver of δ¹⁵N chronologies, which has been the source of substantial recent debate (Craine et al. 2018, 2019; Hiltbrunner et al. 2019; Mason 2022a, 2022b; Olf et al. 2022).

However, since all four reviewers inquired about the corresponding forest growth patterns in Sweden, we have now augmented our Implications section with additional data describing the temporal patterns of Swedish forest growth from the Swedish National Forest Inventory (Supplementary Figure 3). These data show that forest growth has increased significantly from the 1950s in all four regions of Sweden; however, this trend has flattened out during the

last ca. one to two decades of the inventory. The long-term increase in growth is consistent with the explanation that increased forest N demand may be intensifying N limitation. It is beyond the scope of our study to quantify the degree to which PNL is responsible for the recent growth patterns, but we do discuss that this general trend of asymptotic growth is consistent with constrained growth due to nitrogen cycle feedbacks that is predicted by PNL and many earth system models (for good examples, see: Gerber et al., 2010; Kou-Giesbrecht & Arora, 2023; Seiler et al. 2024; Thornton et al. 2007). We now refer to ESM prediction in the Introduction (line 48) and introduce some discussion of ESM predictions in the Implications section (lines 245-248; 278-283).

References:

- Blaško, R. et al. The carbon sequestration response of aboveground biomass and soils to nutrient enrichment in boreal forests depends on baseline site productivity. *Science of the Total Environment* **838**, 156327 (2022).
- Craine, J.M. et al. Isotopic evidence for oligotrophication of terrestrial ecosystems. *Nature Ecology & Evolution* **2**, 1735-1744 (2018).
- Craine, J.M. et al. Reply to "Data do not support large-scale oligotrophication of terrestrial ecosystems." *Nature Ecology and Evolution* **3**, 1287-1288 (2019).
- Canadell, J. G., et al. Global carbon and other biogeochemical cycles and feedbacks (Chapter 5). *Climate Change 2021: The Physical Science Basis. Contribution of Working Group I to the Sixth Assessment Report of the Intergovernmental Panel on Climate Change*, 673–816 (Cambridge University Press 2021).
- Davies-Barnard, T. et al. Nitrogen cycling in CMIP6 land surface models: progress and limitations. *Biogeosciences* **17**, 5129-5148 (2020).
- Gerber, S. et al. Nitrogen cycling and feedbacks in a global dynamic land model. *Global Biogeochemical Cycles* **24**, GB1001 (2010)
- Hiltbrunner, E. et al. Data do not support large-scale oligotrophication of terrestrial ecosystems. *Nature Ecology & Evolution* **3**, 1285–1286 (2019).
- Kou-Giesbrecht, S.K. & Arora, V. K. Compensatory Effects Between CO₂, Nitrogen Deposition, and Nitrogen Fertilization in Terrestrial Biosphere Models without Nitrogen Compromise Projections of the Future Terrestrial Carbon Sink. *Geophysical Research Letters* **50**, e2022GL102618 (2023).
- Mason, R.E. et al. Evidence, causes, and consequences of declining nitrogen availability in terrestrial ecosystems. *Science* **376**, 1-11 (2022a).
- Mason, R.E. et al. Response: Explanations for nitrogen decline. *Science* **376**, 1170 (2022b).
- Olf, H. et al. Explanations for nitrogen decline. *Science* **376**, 1169-1170 (2022).
- Schulte-Uebbing, L.F., G.H. Ros, W. de Vries. Experimental evidence shows minor contribution of nitrogen deposition to global forest carbon sequestration. *Global Change Biology* **28**, 899-917 (2021).
- Seiler, C. et al. The impact of climate forcing biases and the nitrogen cycle on land carbon balance projections. *Journal of Advances in Modeling Earth Systems* **16**, e2023MS003749 (2024).
- Sigurdsson, B.D. et al. Growth of mature boreal Norway spruce was not affected by elevated [CO₂] and/or air temperature unless nutrient availability was improved. *Tree Physiology* **33**, 1192-1205 (2013).
- Thornton, P.E. et al. Influence of carbon-nitrogen cycle coupling on land model response to CO₂ fertilization and climate variability. *Global Biogeochemical Cycles* **21**, GB4018 (2007).
- Thomas, R.Q., Brookshire, E.N.J. & Gerber, S. Nitrogen limitation on land: how can it occur in Earth system models? *Global Change Biology* **21**, 1777-1793 (2015).

A few suggestions:

Comment: Line 57 – The soil processes underlying changes in plant and soil ^{15}N values are complex and unfamiliar to many readers. One or two sentences explaining why high N availability favors loss of ^{15}N would be helpful.

Response: We agree that this is a good suggestion. We have now added more detail in the Introduction (lines 55 – 70), which previously we had not introduced until the Implications section. Also, we have now clarified (lines 66 – 68) that the relationships between plant $\delta^{15}\text{N}$, soil N availability, and foliar N content are well-established from spatial datasets, through meta-analyses. These relationships are not controversial. We clarify in the Introduction that disagreement and controversy over the interpretation of $\delta^{15}\text{N}$ values has focused specifically on time series data (i.e. chronologies), which our study directly addresses.

Comment: Line 74 – As above, a brief explanation of why $\text{NH}_x:\text{NO}_y$ affects $\delta^{15}\text{N}$ would be helpful.

Response: Good suggestion. Here we have inserted a brief explanation that NO_y is derived to a greater degree from fossil organic matter combustion and typically has higher $\delta^{15}\text{N}$ values (lines 87-91). Thus, shifts in the ratio of NH_x to NH_y in deposition can shift the bulk signature of N deposition, providing an alternative explanation for declining $\delta^{15}\text{N}$ values through time. We cite Olf et al. (2022), who proposed this may be a problem in interpreting temporal $\delta^{15}\text{N}$ datasets (i.e. chronologies).

Comment: Line 135 – should be -0.763 (negative sign missing)

Response: Yes, these values were incorrectly reported; they have now been corrected (lines: 160-162).

Comment: Line 147 – should be ... changing $\delta^{15}\text{N}$ value of ...

Response: Change made.

Comment: Line 206 – I believe there is little evidence of limited N availability associated with elevated CO_2 leading to reductions in forest growth, to date. Is there evidence of declining forest growth, or is this still hypothetical? Effects of N on growth with elevated CO_2 are complicated by the effects of elevated CO_2 on improved water use efficiency. This confounding effect was not discussed. Perhaps a few sentences of discussion could be added. I wonder if C isotopes were also analyzed from these wood cores. It would be very interesting to try to separate the effects of N limitation and water use efficiency.

Response: Regarding the first comment, evidence for eCO_2 -induced N limitation of growth, there is no evidence to date that PNL feedbacks are constraining growth in boreal forests, but this feedback is now represented in a majority of ESMs from which global C cycle and climate predictions are based. Our demonstration that rising CO_2 is intensifying N limitation in boreal forests is what we consider our main breakthrough, which directly answers a series of high-profile debates on this topic. Whether intensified N limitation is altering boreal forest growth or C sink strength is beyond the scope of our study, which is why we discussed these topics in the Implications section rather than the results-focused sections. As stated in a

response above, there is mixed evidence for eCO₂-induced PNL constraining growth in temperate forest FACE experiments (see Norby et al. 2010 for supporting evidence). Also, we did not mean to imply here that PNL would lead to a “reduction” in forest growth, but prefer we use the word “constrain”, since CO₂ has a direct positive impact on growth. The indirect effect of negative nitrogen cycling feedbacks (i.e. PNL) on growth would work against the direct positive effect of rising CO₂ on growth and thus is proposed to “constrain” the forest growth response to eCO₂. This “constraining” interpretation is consistent with the original PNL framework (Luo et al. 2004), nitrogen cycle feedbacks represented in ESMs (Davies-Barnard et al. 2021; Thomas et al. 2015) and is also consistent with the growth response to eCO₂ found in two Swedish experiments where it was concluded that N availability constrains the forest growth response to eCO₂ (Sigurdsson et al. 2013). We have added these references to the manuscript.

Regarding the second comment about forest growth patterns in Sweden, we have now compiled growth data from the Swedish National Forest Inventory over the timespan of our $\delta^{15}\text{N}$ chronology. This data shows that growth has generally increased over our study period in all regions of Sweden, but has slowed substantially over the last ca. one to two decades of the inventory (Supplementary Figure 3). The long-term trend of increased growth supports the mechanism that higher plant N demand is a likely factor that has intensified N limitation. Regarding the recent downturn in growth, there are many factors that may be contributing to this (e.g. drought, forest damage from wind, etc), but PNL is certainly a plausible factor that could cause forest growth to follow an asymptotic pattern. It is beyond the scope of our study to quantify the impact of PNL on forest growth or net ecosystem production, but experimental data from eCO₂ experiments (Norby et al. 2010; Sigurdsson et al. 2013), ESM simulations (Davies-Barnard et al. 2021; Thomas et al. 2015), and the most recent IPCC report, all indicate that N limitation and PNL is an important factor that will likely constrain forest growth responses to eCO₂. Here we also note that meta-analysis of eCO₂ experiments has shown that plant $\delta^{15}\text{N}$ values on average decline (BassiriRad et al. 2003). We believe the breakthrough of our study is to complement these experimental or modelling approaches by showing conclusively for the first time that PNL is actually well underway in boreal forests. We have added to and improved text in the Implications section on these points at lines (231-252).

Finally, about water use efficiency, we do indeed have $\delta^{13}\text{C}$ data from the same 10-year increment core segments used for the current manuscript. However, we do not think the link to N limitation is straightforward and believe that such discussion would be too speculative. Of course, the water use efficiency of all plants increases with rising CO₂ concentrations, regardless of whether any physiological stress response occurs. Our unpublished $\delta^{13}\text{C}$ data from the 10-year core segments generally tracks the “Seuss effect” and therefore do not indicate any obvious long-term change in plant physiological stress response that would indicate tree water use efficiency deviates from the general increase in water use efficiency that all plants are experiencing. We note also that boreal forests are not usually considered water limited, except during extreme drought years. Ultimately, we respectfully prefer not to add text in the Implication section on water use efficiency, because of its speculative and tangential nature. We feel that the Implications section already gives the reader many important dimensions to consider.

References:

Davies-Barnard, et al. Nitrogen cycling in CMIP6 land surface models: progress and limitations. *Biogeosciences* **17**, 5129-5148 (2020).

BassiriRad, H. et al. Widespread foliage $\delta^{15}\text{N}$ depletion under elevated CO_2 : inferences for the nitrogen cycle. *Global Change Biology* **9**, 1582-1590 (2003).

Luo, Y. et al. Progressive nitrogen limitation of ecosystem responses to rising atmospheric CO_2 . *BioScience* **54**, 731–739 (2004).

Norby, R.J. et al. CO_2 enhancement of forest productivity constrained by limited nitrogen availability. *PNAS* **107**, 19368-19373 (2010).

Thomas, R.Q., Brookshire, E.N.J. & Gerber, S. Nitrogen limitation on land: how can it occur in Earth system models? *Global Change Biology* **21**, 1777-1793 (2015).

Sigurdsson, B.D. et al. Growth of mature boreal Norway spruce was not affected by elevated $[\text{CO}_2]$ and/or air temperature unless nutrient availability was improved. *Tree Physiology* **33**: 1192-1205 (2013).

Comment: Line 211 – should be ... ectomycorrhizal fungi that mobilize and acquire N ...

Response: Change made.

Comment: Line 221 and throughout – the authors discuss “internal N cycling” but do not define what they mean. To me, internal N cycling is retranslocation within a tree. I understand that ecosystem N cycling is meant, but this should be defined at the outset.

Response: We indeed refer to the more traditional biogeochemical definition of the internal N cycle here (Stevenson & Cole 1999), which refers to the turnover of N via detrital inputs and decomposition. We have reworded this to “...if CO_2 increases plant C:N that stimulates microbial N immobilization during detrital decomposition, thus also reducing soil N availability” at line 45.

References:

Stevenson, F. J. & Cole, M. A. *Cycles of Soils: Carbon, Nitrogen, Phosphorus, Sulfur, Micronutrients*. 2nd edn, 448 pp. (Wiley, 1999).

Comment: References

Penuelas, J. et al. Increasing atmospheric CO_2 concentrations correlate with declining nutritional status of European forests. *Commun Biol* **3**, 125 (2020).

Palmroth, S. et al. Increased leaf area index and efficiency drive enhanced production under elevated atmospheric $[\text{CO}_2]$ in a pine-dominated stand showing no progressive nitrogen limitation. *Glob Change Biol* **30**, 2 (2024).

Response: We have now included the Penuelas reference, referred to at lines 83 and 226, and the Palmroth et al. reference at line 271.

REFEREE #2 (REMARKS TO THE AUTHOR):

Comment: The manuscript by Bassett et al. investigates the factors driving changes in the nitrogen isotopic composition ($\delta^{15}\text{N}$) in tree stems, identifying atmospheric CO_2 concentrations as the primary driver of the observed $\delta^{15}\text{N}$ decline since the 1960s, while other factors such as N deposition, temperature, and basal area appear to have little or counterbalancing effects. The abstract and the study is well-structured, with a mostly clear experimental design and appropriate statistical analyses (but see my specific response). The writing is also straightforward and easy to follow.

Response: We thank the reviewer for the general positive assessment of the manuscript. Below we address the specific comments about consequences of our core data for the C cycle, which were discussed in our “Implications” section.

Comment: However, I have significant concerns regarding the manuscript’s main conclusion that links the $\delta^{15}\text{N}$ decline (caused by CO_2) to increased nitrogen limitation:

1. Interpreting $\delta^{15}\text{N}$ in forest ecosystems is inherently complex, as the processes influencing $\delta^{15}\text{N}$ remain only partially understood. This is particularly evident in the speculative discussion (especially in the "Implications" section) on why atmospheric CO_2 is the primary driver of $\delta^{15}\text{N}$ and nitrogen limitation. The two arguments remain inconclusive about the processes linking $\delta^{15}\text{N}$ and N limitation and instead highlight gaps in our understanding, opening new research questions rather than providing a definitive answer. Furthermore, because this study focuses on trees, I assume that nitrogen limitation refers to a process that restricts nitrogen availability for trees or forest ecosystems. To support this claim, I would expect the authors to present direct evidence linking $\delta^{15}\text{N}$ trends to measurable traits, such as nitrogen concentrations (in different chemical forms) in trees or soils, tree physiology, or growth responses, that would support the conclusion that CO_2 is driving nitrogen limitation in boreal forests.

Response: We thank the reviewer for this comment, and for the opportunity to address it. Indeed, our study relies on a pre-existing understanding of $\delta^{15}\text{N}$ as a proxy for N limitation, and our study did not attempt to prove this, as we feel there is already a robust literature indicating that plant $\delta^{15}\text{N}$ is well correlated with foliar N and ecosystem N availability, which has been established globally (See figures included below), and across plant functional groups that differ in growth rates, or mycorrhizal association types associated with different habitat fertilities. A few well-cited reviews and meta-analyses on these well accepted patterns include Craine et al. 2009, 2018; Chen et al. 2025; Gerhart & McLauchlan, 2014. These spatial relationships are not controversial and provide robust support that plant $\delta^{15}\text{N}$ is a reliable proxy for N availability to plants. It is also not controversial that the two main source of N fractionation that influence plant $\delta^{15}\text{N}$ values are: a) N transformations associated with detrital decomposition and associated N loss pathways (nitrification, denitrification, and leaching) that control the $\delta^{15}\text{N}$ value of N taken up by plants; and b) fractionation during transfer of N from soil to plants via mycorrhizal fungi (Craine et al. 2015; Hobbie & Högberg 2012). What *has been* controversial is the interpretation of $\delta^{15}\text{N}$ values in a temporal context, which is why we framed our study in reference to these ongoing debates about the interpretation of $\delta^{15}\text{N}$ chronologies, specifically (Craine et al. 2018, 2019; Hiltbrunner et al. 2019; Mason 2022a, 2022b; Olf et al. 2022). Our results are a major advancement in this debate because we clearly separate the roles of rising CO_2 , changing N deposition, temperature, and forest biomass on $\delta^{15}\text{N}$ patterns. Regarding the final point about providing “direct evidence” linking rising CO_2 to measurable traits, we argue that direct evidence from FACE experiments linking $e\text{CO}_2$ to $\delta^{15}\text{N}$ already exists (see BassiriRad et al. 2003). The novelty of our data is that it provides the best spatio-temporal dataset describing N limitation to date (see comments by other reviewers on this), which allow clear separation of debated drivers of temporal $\delta^{15}\text{N}$ patterns. The advantage of the archive-based chronology we present is that it provides a long-term high-quality temporal dataset, that no other research approach can provide. For example, FACE-type experiments can provide experimental control of CO_2 , but they are inherently short term and poorly replicated due to their extreme cost. We are aware of only three $e\text{CO}_2$ -enrichment experiments in boreal conifer forests to date, the most recent SPRUCE experiment in Minnesota, which is established in a very low-productivity

peatland that is inherently different than upland boreal forests that are the focus of our current study. In addition to this experiment, two earlier experiments were carried out in Sweden 1998-2000, and 2001-2004, where individual trees were subjected to eCO₂ x N fertilization (n=3), or eCO₂ x warming (n=3), respectively (Sigurdsson et al. 2013). While these two Swedish experiments are highly relevant for our dataset, they were both very short-term and small scale (i.e. treatments applied to individual trees with only a few replicates). Growth responses from these two Swedish experiments showed that eCO₂ promotes growth, but only when N limitation is simultaneously alleviated (Sigurdsson et al. 2013). While the result of this experiment is very complimentary to our study (now cited in our Implications section), the duration and scale of these experiments is just too small to provide a “direct” evaluation of PNL. We therefore maintain that our data provides the most direct and conclusive evidence that PNL is occurring in boreal forests to date, where our interpretation is supported by a wide range of other data sets, including meta-analyses from FACE experiments.

[Redacted figures]

References:

- BassiriRad, H. et al. Widespread foliage $\delta^{15}\text{N}$ depletion under elevated CO₂: inferences for the nitrogen cycle. *Global Change Biol* **9**, 1582–1590 (2003).
- Chen, Q. et al. Global mycorrhizal status drives leaf $\delta^{15}\text{N}$ patterns. *J Ecol* **113**, 1150-1163 (2025).
- Craine, J.M. et al. Ecological interpretations of nitrogen isotope ratios of terrestrial plants and soils. *Plant Soil* **396**, 1–26 (2015).

- Craine, J.M. et al. Isotopic evidence for oligotrophication of terrestrial ecosystems. *Nat Ecol Evol* **2**, 1735-1744 (2018).
- Craine, J. et al. Reply to "Data do not support large-scale oligotrophication of terrestrial ecosystems." *Nature Ecology and Evolution* **3**, 1287-1288 (2019).
- Gerhart, L.M. & McLauchlan, K.M. Reconstructing terrestrial nutrient cycling using stable nitrogen isotopes in wood. *Biogeochemistry* **120**, 1-21 (2014).
- Hiltbrunner, E., Körner, C., Meier, R., Braun, S. & Kahmen, A. Data do not support large-scale oligotrophication of terrestrial ecosystems. *Nat Ecol Evol* **3**, 1285–1286 (2019).
- Hobbie, E.A. & Högberg, P. Nitrogen isotopes link mycorrhizal fungi and plants to nitrogen dynamics. *New Phyt* **196**, 367–382 (2012).
- Mason, R.E. et al. Evidence, causes, and consequences of declining nitrogen availability in terrestrial ecosystems. *Science* **376**, 1-11 (2022a).
- Mason, R.E. et al. Response: Explanations for nitrogen decline. *Science* **376**, 1170 (2022b).
- Olf, H. et al. Explanations for nitrogen decline. *Science* **376**, 1169-1170 (2022).
- Sigurdsson, B.D., Medhurst, J., Wallin, G., Eggertsson, O. & Linder, S. 2013. Growth of mature boreal Norway spruce was not affected by elevated [CO₂] and/or air temperature unless nutrient availability was improved. *Tree Physiology* **33**, 1192-1205.

Comment: 2. Tree-ring $\delta^{15}\text{N}$ measurements may be influenced by nitrogen mobility, particularly within the sapwood. Previous research (e.g., Tomlinson et al., 2014, DOI: 10.1002/rcm.6897) has shown that nitrogen can be partially remobilized across tree rings, raising concerns about whether $\delta^{15}\text{N}$ values from a given year or period accurately represent the $\delta^{15}\text{N}$ signature of the forest ecosystem at that time. A discussion of this potential limitation is necessary.

Response: We are highly aware of this issue, and the fact that our methodology explicitly controlled for this issue is a major strength of our study, which we already highlighted at lines 112-117. In brief, our use of archived samples (as opposed to constructing chronologies from samples only collected in the present), allowed us to hold tree age constant, and thus always analyze the $\delta^{15}\text{N}$ value of the outermost 10-year increment of trees in the same age window for every decade. We clearly show the advantage of our approach of controlling for the issue of translocation in a preceding manuscript (Bassett et al. 2023), which we show provides much greater sensitivity to identifying historical trends. We have included the Tomlinson reference in the manuscript as a good reference for this problem, and Bassett et al. (2023) as a reference to our methodological solution to this problem.

References:

- Bassett, K.R., Östlund, L., Gundale, M.J., Fridman, J. & Jämtgård, S. Forest inventory tree core archive reveals changes in boreal wood traits over seven decades. *Sci Total Environ* **900**, 165795 (2023).

Comment: 3. The manuscript does not sufficiently describe the primary nitrogen sources and isotope fractionation processes that shape tree-ring $\delta^{15}\text{N}$ values. Without this context, it remains unclear to the reader what specific ecological or physiological information $\delta^{15}\text{N}$ in tree rings integrates. Providing a clearer introduction/discussion of these mechanisms would significantly improve the study's transparency and interpretability.

Response: There are two dominant fractionation processes that influence plant $\delta^{15}\text{N}$ values: a) transformations and losses related to the soil N cycling process (e.g. nitrification, denitrification, NO_3^- leaching); and b) uptake by mycorrhizal fungi (see reviews by Craine et

al. 2015; Gerhart & McLauchlan 2014; Hobbie & Högberg 2012), which we now cite in the Introduction. These are non-mutually exclusive mechanisms which can operate simultaneously in response to “opening” or “tightening” of the N cycle. Both increased ecosystem N retention and higher mycorrhizal uptake are symptomatic of tightening N cycles, and steer $\delta^{15}\text{N}$ more negative. In the previous draft, most of these details were presented in the Implications section, and we very briefly describe controlling processes in the Introduction. This was obviously a mistake, since multiple reviewers commented on this. To address this comment, we have now shifted more of this background to the Introduction section, with supporting references. Our modification to the Introduction is found at lines 55- 70, where more details and foundational review/meta-analysis references have been added.

References:

- Craine, J.M. et al. Ecological interpretations of nitrogen isotope ratios of terrestrial plants and soils. *Plant and Soil* **396**, 1-26 (2015).
- Gerhart, L.M. & McLauchlan, K.K. Reconstructing terrestrial nutrient cycling using stable nitrogen isotopes in wood. *Biogeochemistry* **120**, 1-21 (2014).
- Hobbie, E.A. & Högberg, P. Nitrogen isotopes link mycorrhizal fungi and plants to nitrogen dynamics. *New Phytologist* **196**, 367-382 (2012).

Comment: Overall, while the study presents an interesting and well-executed analysis, the interpretation of $\delta^{15}\text{N}$ as a clear indicator of nitrogen limitation requires stronger empirical support and a more nuanced discussion of potential confounding factors.

Response: See our response above. In short, there is a robust literature supporting that $\delta^{15}\text{N}$ is an indicator of N limitation and retentiveness across spatial gradients, to which we now provide several references (Craine et al. 2009, 2018; Chen et al. 2025, Templer et al. 2012). Recent debates on the interpretation of $\delta^{15}\text{N}$ data have focused on temporal $\delta^{15}\text{N}$ patterns (chronologies), not the general utility of $\delta^{15}\text{N}$ to serve as an indicator of N limitation. We hope we have made this clearer in the introduction now (see lines 55-70). The main breakthrough of our study is that it greatly clarifies the debated contributions of eCO₂ versus N deposition towards temporal patterns. Regarding nuance, we respectfully feel that our analysis makes a very clear case that rising CO₂, not atmospheric N deposition, is the main driver of the declining $\delta^{15}\text{N}$ chronologies we report in our study. We have included many environmental and stand level factors that could conceivably influence $\delta^{15}\text{N}$ chronologies, using a very unique study system and methodological tool. The fact that rising CO₂ clearly emerges as the dominant predictor, even after considering additional model variants suggested by Reviewer 3, we think our main conclusion is very robust. We have now provided new supporting data to help guide our speculation on the implications our results may have for the C cycle. This includes long-term data on forest growth, and references to published eCO₂ experiments and ESM modelling studies, all of which highlight the importance of eCO₂-induced PNL on the ecosystem C dynamics. Here we are very careful in our language (i.e. nuance) in that we do not claim the intensified N limitation explains a certain quantity of C cycle change. We carefully argue that intensified N limitation appears to be underway and may help us understand changes in the development of forest growth, which can be evaluated through new approaches.

References:

- Chen, Q. et al. Global mycorrhizal status drives leaf $\delta^{15}\text{N}$ patterns. *J Ecol* **113**, 1150-1163 (2025).

Craine, J.M. et al. Isotopic evidence for oligotrophication of terrestrial ecosystems. *Nat Ecol Evol* **2**, 1735-1744 (2018).

Craine, J.M. et al. Global patterns of foliar nitrogen isotopes and their relationships with climate, mycorrhizal fungi, foliar nutrient concentrations, and nitrogen availability. *New Phyt* **183**, 980-992 (2009).

Templer, P.H. et al. Sinks for nitrogen inputs in terrestrial ecosystems: a meta-analysis of ^{15}N tracer studies. *Ecology* **93**, 1816-1829 (2012).

Specific comments:

Comment: Line 43: Please clarify what "internal" refers to exactly?

Response: Thank you for pointing this out, which Reviewer 1 also mentioned. We have now reworded this so it now just describes soil N turnover via decomposition of detrital inputs.

Comment: Line 92: Figure S1 presents a PCA, so it is not immediately clear how one can deduce an orthogonal spatio-temporal relationship between atmospheric N_r deposition and eCO_2 . Please clarify this statement.

Response: Orthogonal simply refers to two factors being crossed, or at a right angle to one another, which implies they are independent. Specifically, here we mean that CO_2 has risen uniformly across Sweden (because CO_2 is atmospherically well mixed), whereas N deposition is much higher in the south versus the north. Because of this, they come at right angles in the PCA (Supplementary Figure 1), indicating they are not co-linear, which is ideal for distinguishing their relative importance within the linear mixed modelling framework we used. This fact is a major advantage of our study system, and it allowed us to robustly and confidently conclude that eCO_2 had much stronger explanatory power than N deposition. We have changed the word orthogonal to independent to avoid any confusion (now line 109). We note that many lower latitude regions have a high degree of co-linearity between N deposition and rising CO_2 , and the lack of co-linearity between these variables in our study area is one of several major advantages of performing this analysis on the Swedish forest landscape.

Comment: Figure 1B: Since the decrease in wet N deposition in southwest Sweden does not appear to influence $\delta^{15}\text{N}$ patterns differently than at other sites, this could serve as additional evidence that N deposition is not the primary driver of tree-ring $\delta^{15}\text{N}$.

Response: Our model exactly weighs this pattern of N deposition data against rising CO_2 concentrations (as well as temperature and basal area). Indeed, the fact that the same exact $\delta^{15}\text{N}$ patterns emerge in the northernmost region of Sweden, where N deposition is very low and stable, is exactly the reason why CO_2 emerges as a much strongest predictor in our models. This comment suggests the reviewer is in agreement with the interpretation of our model output, and the core interpretation presented in the first draft of the manuscript. To make this point even more clear, we have now modified the sentence at line 138, in reference to this figure "...indicating that N deposition was not strongly influential on $\delta^{15}\text{N}$ ".

Comment: Line 130: Instead of referring to Figure S1, I suggest explicitly listing the co-linear N deposition variables here.

Response: Done

Comment: Line 135: Should there be a minus sign for the slope of *P. abies*?

Response: Thank you for catching this mistake. Change made.

Comment: Lines 151–154: Consider adding supporting variables (e.g., t-values from Figure S1) to further illustrate the differences in the relative contributions of CO₂ vs. N deposition traits.

Response: More relevant than t-values here, are the partial R²m values, which we have now inserted into the parentheses (now line 180).

Comment: Line 164: The manuscript alternates between CO₂ and eCO₂ when referring to atmospheric CO₂ concentrations. Could the authors clarify the distinction?

Response: Thanks for pointing out the lack of clarity here. We now use CO₂ in all occurrences, except when we refer to an experimental increase in CO₂ (e.g. FACE studies), in which case we refer to eCO₂, and define this at the first usage (lines 13-14).

Comment: Figure S2: I suggest merging Figures S1 and S2 to better emphasize the differences between temperature and CO₂ effects on δ¹⁵N.

Response: Here we wonder if the reviewer is instead referring to Figure 2 and S2, which are both outputs from our linear mixed effects model. Figure S1 shows a PCA analysis used to assess co-linearity or independence of the different predictor variables, and thus we do not see it as a good fit to combine this figure with S2. However, assuming the reviewer meant figure 2 and S2, we have now combined these, so all Linear Mixed Model results are presented in a single location (Figure 2).

Comment: Line 374: It is unclear how the number of archived samples explains why some decades are linked to single years and others to three-year periods. It suggests that there is some variability in the length of the tree cores and that the number of individual trees that are used for a specific δ¹⁵N value of a period/area is variable. Please add some more details. Statistics: In the case, the selected trees for each period are not independent of each other (i.e. repeated measure), I would suggest implementing the individual tree ID as a random factor in their model. Why do the authors not consider the temporal effect in their model?

Response: Thanks for pointing out this missing detail. The reason for this is that from the 1980s onward, the sampling intensity of tree core collection and archiving was reduced by the National Forest Inventory, and so we needed to increase the time window from which to randomly select cores, which we did by adding a year before and after the original target year. Regardless of the sampling year, it was always a 10-year segment we analyzed. Thus, this had no consequence for our analysis, other than it assured we could carry out a good randomization process in which specific cores selected for the analysis were completely independent. Hopefully, with this added detail, it is clear that the samples are independent and not repeated measures. We have added this detail to the Methods section (now at lines 454-457).

Comment: Line 395: The connection between the Carbosorp/MgClO₄ traps and sample weighing is not entirely clear. Consider splitting this into two sentences for clarity. Were the wood samples packed in tin capsules?

Response: We have now split this into two sentences. The samples were indeed packed in tin capsules, following standard procedures.

Comment: Line 420: Could the authors provide evidence of ISIMIP3a model uncertainties for their study sites, particularly for the selected N deposition traits? Does the model provide realistic values across all sites in Sweden? Also, does it account for dry or wet N deposition?

Response: The ISIMIP3a data set does not report any model uncertainties for specific point locations. They only report the relevant scale that their data is interpolated to ($0.5^\circ \times 0.5^\circ$ grid or ca. 50 x 50 km), which was done using the nearest neighbor method. Their data included dry and wet deposition. These details have been added to the Methods section at lines 507-512. There are multiple sources of global or regional gridded N deposition available, and they can differ slightly in their estimates for specific point locations, because they are based on different models. However, all datasets generally capture very well the strong gradient in N deposition from south to north across Sweden, which is central to our analysis. For example, an alternative estimation of N deposition patterns can be seen in Pihl-Karlsson et al. (2024), which clearly shows the same north to south N deposition gradient described by the ISIMIP3a data (our Figure 1a,b). Their dataset shows slightly lower N deposition rates in the far north, and slightly higher rates in the far south, but the overall gradient matches the ISIMIP3a dataset extremely well. We selected ISIMIP3a data for the study for several reasons, including that it is a mainstream dataset derived from gridded NH_x and NO_y deposition data which is simulated by the NCAR Chemistry-Climate Model Initiative (CCMI), it has been used to inform analysis within the 6th IPCC report, and has also been included in model intercomparison projects (Lamarque et al. 2013). ISIMIP3a also spans the duration of our $\delta^{15}\text{N}$ chronology, whereas some other commonly used N deposition datasets span shorter durations. Here we note that we compared extracted ISIMIP3a data with the Multi-scale Atmospheric Transport and Chemistry model (MATCH; $0.5^\circ \times 0.5^\circ$, 50 km x 50 km resolution) developed by the Swedish Meteorological and Hydrological Institute) database and found it to be highly correlated (Pearson's correlation coefficient, $r = 0.83$).

[Redacted image]

References:

Karlsson, G.P. et al. Atmospheric deposition and soil water chemistry in Swedish forests since 1985 – Effects of reduced emission of sulphur and nitrogen. *Science of the Total Environment* **913**, 169734 (2024).

Lamarque et al. Multi-model mean nitrogen and sulfur deposition from the Atmospheric Chemistry and Climate Model Intercomparison Project (ACCMIP): evaluation of historical and projected future changes. *Atmos. Chem. Phys.* **13**, 7997–8018 (2013).

REFeree #3 (REMARKS TO THE AUTHOR):

Comment: This is an impressive dataset, with a well-designed sampling regime. The authors have systematically sampled archived tree cores from across Sweden, over six decades, to look at long-term trends in $\delta^{15}\text{N}$. They are able to show consistent declines over time in two species and in different regions of Sweden. While similar trends have been identified previously, there has been considerable debate in recent literature about the drivers of this trend, and in particular the relative roles of rising CO_2 vs N deposition (Refs 1 – 5). The sampling approach used here provides the opportunity to disentangle these two drivers because patterns of N deposition are quite different in different regions of Sweden, whereas CO_2 increases consistently everywhere. It appears fairly clear from the analysis that N deposition patterns are not a good explanation of the trend, since the same trend is seen throughout Sweden. As a result, this appears to be a well designed study that resolves an outstanding debate in the literature. However, I have a few issues with the novelty and solidity of the conclusions.

Response: We thank the reviewer for the positive assessment of our core data and conclusion, and we highlight that separating these debated drivers is what we consider the main focus and breakthrough of the study. The study was designed and organized to directly address this debate, and this is the first study to provide a broad and robust spatio-temporal dataset of plant $\delta^{15}\text{N}$ values that provides an answer to current debates in this field.

Comment: It's important to note that some of the data used here appear to have been previously published. Ref 26 tests the methodology for central Sweden. It is referred to in the text as a methods paper, but the text should also have stated that the trend in $\delta^{15}\text{N}$ has already been shown over time for these two species in central Sweden, so the main contribution of this paper is to expand the analysis to all of Sweden, thereby sampling a wider range of N deposition values.

Response: The reviewer is correct that a portion of the $\delta^{15}\text{N}$ data we present, from two counties in north central Sweden (Jämtland and Västernorrland, 2 of 21 counties in Sweden), were previously analyzed and described in a preceding methods paper. This data accounts for 17% of the data in the current paper. In the previous manuscript, we compare different ways of constructing $\delta^{15}\text{N}$ chronologies (from single cores sampled in the present, or multiple cores from the archive where tree age and the issue of N translocation is held constant). That study helped us identify the strength of the methodology we employed for the current study, and we carried a small portion of the data from that study forward to this study to complete national coverage of sampling. That previous study had a completely different objective. The current manuscript provides much more data that covers the entirety of Sweden's forest area, and more importantly spans the wide N deposition gradient from southern to northern Sweden. Thus, the full dataset featured in the current study uniquely allows us to disentangle the debated drivers of $\delta^{15}\text{N}$ chronologies, specifically N deposition versus rising atmospheric CO_2 . We now mention in the Methods section the overlap of data for these two counties with our previous study (lines 460-461).

Comment: I'm uncomfortable that the link to rising CO₂ is made only via a temporal correlation. Both d¹⁵N and CO₂ change monotonically over time. As a result, d¹⁵N is correlated with CO₂ – but it would be similarly correlated with any other variable that changed monotonically over time. This cannot be regarded as strong evidence for CO₂ as a driver. There are a number of ways in which the evidence could have been strengthened.

Response: The explanatory factors we chose for our model development were all factors that have been discussed and debated as influential on nutrient limitation and/or δ¹⁵N temporal chronologies or have been discussed in context of Earth System Models as drivers of nitrogen cycle feedbacks. So, these explanatory variables were selected strategically, based on robust literature suggesting their potential importance (see for example Craine et al. 2015; Vitousek et al. 2024). We believe the correct way to construct a model and test hypotheses is with this type of *a priori* knowledge about relevant predictors, rather than compiling a wide basket of explanatory variables, which could potentially result in spurious correlations. Could some hidden factor that we did not consider in our model explain our δ¹⁵N response? We think this is very unlikely since we have incorporated into our model all major factors that have been in the scientific discussion. We are unable to identify any other factor that would conceivably operate so uniformly across the 1500 km latitudinal gradient of Sweden, where forest properties and climate varies so much. Finally, we would like to highlight a study by BassiriRad et al. (2003) that describes a widespread decline in foliar δ¹⁵N in elevated CO₂ experiments. We now reference this study in the Implications section at line 225, which provides good experimental support to our approach.

References:

- BassiriRad, H. et al. Widespread foliage δ¹⁵N depletion under elevated CO₂: inferences for the nitrogen cycle. *Global Change Biol* **9**, 1582–1590 (2003).
- Craine, J.M. et al. Ecological interpretations of nitrogen isotope ratios of terrestrial plants and soils. *Plant Soil* **396**, 1–26 (2015).
- Vitousek, P.M., Cen, X. & Groffman, P.M. Has nitrogen availability decreased over much of the land surface in the past century? A model-based analysis. *Biogeochemistry* **167**, 793-806 (2024).

Comment: First, the explanation of the mechanisms by which rising CO₂ would reduce d¹⁵N was not sufficiently precise. The paper would have been significantly stronger with the support of a conceptual model to explain the proposed mechanism, with some description of what other changes must also be taking place, and an examination of the evidence for those other changes. For example, my understanding of progressive nitrogen limitation is that it is only likely to occur if plant demand for nitrogen has increased because of a stimulation of plant growth. Is there evidence that trees are now growing faster than 50 years ago? That does appear to be the case – they note that “The observed increase in total basal area over time can be attributed to the combined influence of forest management and environmental change factors.” Oddly, however, this is only discussed as a statistical confounder, instead of as a key part of the story.

Response: We have sharpened our introduction to better indicate the main mechanisms by which eCO₂ could result in PNL (lines 39-45), and why declining δ¹⁵N chronologies are diagnostic of PNL (lines 55-70). Regarding δ¹⁵N chronologies, the main mechanisms by which CO₂ induced PNL would steer δ¹⁵N chronologies negative is due to higher plant demand and/or reduced soil N supply due to microbial biomass immobilization, and/or greater tree reliance on ectomycorrhizal fungi. These are non-mutually exclusive responses that

would occur under a “tightening” of the N cycle (i.e. intensified N limitation). Further, we have included in the introduction a statement about how the development of PNL should influence the forest C balances through time. Here we emphasize that increasing forest growth and changing stoichiometry in response to rising CO₂ should intensify N demand (i.e. resulting in less fractionation and lower $\delta^{15}\text{N}$ values), and resulting PNL would eventually constrain growth, as predicted by many ESMs (see Davies-Barnard et al. 2021; Thomas et al. 2015 for example reviews).

Further, we have now added forest growth data to the manuscript for pine and spruce forests in Sweden. This consists of stand level annual volume increment data for forests with the same characteristics as those for which we sampled $\delta^{15}\text{N}$. These data generally show that growth has increased in Sweden over the long term, but that this long-term trend has paused for the last ca. 1 - 2 decades. We are not comfortable stating that recent deceleration of growth in Sweden is purely the result of PNL, but we do propose that it is plausible that PNL may contribute to this, in addition to a variety of other change factors that typically dominate the discussion (drought stress, disease, disturbance, etc.).

References:

- Davies-Barnard, T. et al. Nitrogen cycling in CMIP6 land surface models: progress and limitations. *Biogeosciences* **17**, 5129-5148 (2020).
- Thomas, R.Q., Brookshire, E.N.J. & Gerber, S. Nitrogen limitation on land: how can it occur in Earth system models? *Global Change Biology* **21**, 1777-1793 (2015).
- Terrer, C. et al. A trade-off between plant and soil carbon storage under elevated CO₂. *Nature* **591**, 599-603 (2021).

Comment: I was also surprised that $\delta^{15}\text{N}$ was the only wood trait measured – the previous paper (ref 26) also has data on N%, C% and $\delta^{13}\text{C}$, and shows (quite remarkably) an increase in stemwood N% over time, along with a sizeable reduction in C%, from ca 49% to 46%. These trends do not appear to support the idea that CO₂ is driving the reduction in $\delta^{15}\text{N}$, but have not been quantified here. For this paper to have real conceptual novelty – and a solid conclusion that rests on more than a temporal correlation – I would have expected the conceptual model to encompass the likely changes in other wood traits – including these parameters and stem growth rates – with an exploration of what mechanisms are consistent with all observations.

Response: Thank you for allowing us to address this comment. Indeed, in the previous study the reviewer refers to (focused on two counties in north central Sweden), we unexpectedly found that wood C content declined, and %N increased from the 1950s to the 2010s. However, this pattern does not hold up in the complete dataset covering all of Sweden. We have now included data on the corresponding temporal trend in wood %N content to the manuscript, which shows no change through time (Supplementary Figure 4). We did not present wood %N content in the previous draft, because it is generally not considered diagnostic of tree “N status” (Doucet et al. 2011; Gerhart & McLaughlin 2014; Poulson et al. 1995). Wood and foliar N contents, and wood %N and plant $\delta^{15}\text{N}$ values are often poorly correlated because the %N of wood is extremely low (<0.01% in our samples), and thus small changes in the more abundant elements in wood (e.g. C and O) can lead to substantial variation in wood percent N (Gerhart & McLaughlin 2014). Regarding wood %C, it can change by a few percentage points (thereby affecting the %N content), which is mainly due to changes in the ratio of earlywood to latewood, or wood lignin to cellulose content (Lamloom & Savidge 2003; Arzac et al. 2019; Buttò et al. 2021). We discussed this exact point in our

previous study the reviewer refers to (Bassett et al. 2024). Thus, there is a good precedent for focusing on wood $\delta^{15}\text{N}$ instead of $\%N$, assuming the issues of ^{15}N translation in wood can be overcome, such as we were able to do in our study.

References:

- Bassett, K.R., Östlund, L., Gundale, M.J., Fridman, J. & Jämtgård, S. Forest inventory tree core archive reveals changes in boreal wood traits over seven decades. *Sci Total Environ* **900**, 165795 (2023).
- Doucet, A., Savard, M.M., Bégin, C. & Smirnoff, A. Is wood pre-treatment essential for tree-ring nitrogen concentration and isotope analysis? *Rapid Commun. Mass Spectrom* **25**, 469–475 (2011).
- Gerhart, L.M. & McLauchlan, K.K. Reconstructing terrestrial nutrient cycling using stable nitrogen isotopes in wood. *Biogeochemistry* **120**, 1-21 (2014).
- Poulson, S.R., Chamberlain, C.P., & Friedland, A.J. Nitrogen isotope variation of tree rings as a potential indicator of environmental change. *Chem. Geol.* **125**, 307–315 (1995).
- Lamlom, S.H. & Savidge, R.A. A reassessment of carbon content in wood: variation within and between 41 North American species. *Biomass Bioenergy* **25**, 381–388 (2003).
- Arzac, A. et al. Increasing radial and latewood growth rates of *Larix cajanderi* Mayr. and *Pinus sylvestris* L. in the continuous permafrost zone in Central Yakutia (Russia). *Ann. For. Sci.* **76**, 1–15 (2019).
- Buttò, V. et al. Region-wide temporal gradients of carbon allocation allow for shoot growth and latewood formation in boreal black spruce. *Glob. Ecol. Biogeogr.* **30**, 1657–1670 (2021).

Comment: I would also have expected that the study would be informed by large-scale field experiments with CO₂ and warming in Sweden. In particular, this paper explores how stem growth of 40-year-old trees in northern Sweden responds to rising CO₂ with and without additional fertiliser and warming:

References:

- Sigurdsson, B.D. et al. Growth of mature boreal Norway spruce was not affected by elevated [CO₂] and/or air temperature unless nutrient availability was improved, *Tree Physiology* **33** 1192–1205 (2013).

While this one explores responses of stemwood production to soil warming:

References:

- Strömgren, M. & Linder, S. Effects of nutrition and soil warming on stemwood production in a boreal Norway spruce stand. *Global Change Biology* **8**, 1194-1204 (2002).

Response: We agree that these manuscripts are very useful, especially Sigurdsson et al. 2013, where eCO₂ treatments were applied to Swedish forests of a similar age range as the trees we focused on. The second manuscript (Strömgren & Linder 2002) does not relate to eCO₂ treatments, so is not quite as relevant. The first manuscript shows that growth response to CO₂ is constrained by N availability, which is consistent with PNL, and in line with our discussion in the Implications section. We have now referred to this study explicitly in the Implications section (lines 248-252).

Comment: Some discussion of the role of rising temperatures in altering the N cycle is needed. I note that mean annual temperature is also considered as a predictor in their

statistical models, but my suggestion would be to consider spatial and temporal variability in temperature separately. Spatial gradients in temperature are likely to result in long-term changes in soil carbon and nitrogen pools, whereas increases in temperature over the last 50 years can cause shorter-term increases in nitrogen availability via increased decomposition of those pools.

Response: The potential role of rising temperatures in shaping the N cycle is an important consideration in our work. As a result, we have included additional analysis to resolve this possibility. Regarding spatial versus temporal variation of predictor variables, the reviewer is correct that our model did not partition these aspects of the data. We included “grid cell” as a random factor in our model, which effectively groups samples together at the grid cell level, but this does not accomplish what the reviewer is asking. Thus, to further illuminate the spatial versus temporal variability of the predictor variables, we have added Supplementary Figure 2, where we have now plotted each predictor variable separately against latitude or time. Further, in order to hold the spatial dimension constant in the linear mixed model, we have now repeated each model with latitude and longitude included as fixed effects. This modification did not change the model conclusions (i.e. CO₂ is still by far the best predictor of the $\delta^{15}\text{N}$ chronologies; Supplementary Table 2). We have now highlighted this at lines 156-159.

Regarding the second point, indeed increased soil N mineralization in response to rising temperatures is a plausible pathway that could influence $\delta^{15}\text{N}$ values. As noted in our original draft, this mechanism would steer $\delta^{15}\text{N}$ values more positive through time, which is opposite the observed pattern in wood $\delta^{15}\text{N}$, leading to our original conclusion that temperature had a weak positive relationship with $\delta^{15}\text{N}$ values. To more explicitly describe “temperature change” instead of absolute temperature in our models, we have now created a new temperature variable, where all temperature data is subtracted from the 1961 value of any given grid cell (described in the Methods section at line 501). When we repeat the model with this new temperature variable, we still find that CO₂ is the dominant predictor of $\delta^{15}\text{N}$ values, and temperature change has a significant positive, but weak relationship with $\delta^{15}\text{N}$ (partial R²m values ≤ 0.005). We reference this new analysis at lines 152-155. Given that all analyses point to a small role of temperature in shaping $\delta^{15}\text{N}$ values, we feel confident in maintaining an emphasis on rising CO₂ in the Implications section.

Comment: Finally, the conclusions around the implications for the C sink are quite weak. One potential conclusion is that “ecosystem productivity will be increasingly constrained by intensifying N limitation in response to eCO₂”. However, if the mechanism is that plants are growing faster because of eCO₂, and are able to do so with a reduction in N content, an alternative conclusion might be that increasing C sink strength is being achieved despite N limitations – with a fairly different outlook. I’ll repeat the importance of using a quantitative conceptual model to explore the mechanisms and potential future trajectories, rather than drawing overly simplistic conclusions. There was a lot of work done in the 1990’s and 2000’s developing simple C-N ecosystem models that could be applied in this context (e.g. see work by Goran Agren, Ross McMurtrie, Ed Rastetter) so the wheel would not need to be re-invented.

Response: We respectfully disagree that a conceptual model is the most engaging way to frame our study. Work by the mentioned authors is indeed highly relevant, but in the 2-3 decades since, C-N coupled models have entered the mainstream, and are now represented in a majority of Earth System Models (ESMs). Now more than half of the models used in the

IPCC inter-model comparison include C-N feedback coupling (Section 5.4, Chapter 5, 6th IPCC Assessment Report, 2021). These models represent C-N coupling in different ways, but in general show that C-N coupling feedbacks dampen or constrain predicted NPP or NEP responses to rising CO₂. If the reviewer is referring to a quantitative ¹⁵N fractionation model, such as Vitousek et al. (2024), we feel that such quantitative models rest on many assumptions and are limited by accurate parametrization. We therefore feel the best approach for interpreting our δ¹⁵N data is by relying on existing δ¹⁵N syntheses that show high N availability generally increases tissue δ¹⁵N, and high CO₂ decreases tissue δ¹⁵N values.

Thus, we respectfully feel that the most engaging way to frame our study for a wide readership is to highlight recent debates regarding interpretation of δ¹⁵N chronologies. Our introduction (which we have now improved) first frames different mechanisms and directions that N limitation could be changing in boreal forests, focused on the recent debate about N deposition- versus CO₂-induced PNL. Regarding rising CO₂ and PNL, we have now highlighted in the introduction the main mechanisms by which rising CO₂ is expected to stimulate N limitation, first by promoting forest growth and altering plant stoichiometry, and second by initiating a feedback via reduced detrital turnover, where N supply to plants declines as a result of N mobilization into organic matter and microbial biomass (lines 42 - 49). These core mechanisms are what most coupled C-N ESMs have focused on in the past decades, and we have now referred to ESMs in general as part of our framing about the obvious carbon cycle implications of the work (line 48), and again in the Implications section (lines 245-247; 278-283). Finally, we would like to highlight that the core breakthrough of our study is not to predict how CO₂-induced PNL will impact ecosystem carbon stocks, as this was only introduced as an implication of our finding. The current collection of ESMs is much better positioned to make these predictions than we are. The core breakthrough of our study is to provide the first evidence that PNL is actually underway in boreal forests, and to resolve recent debate regarding the drivers of reported δ¹⁵N chronologies.

However, to better guide our discussion on the C cycle implications of our data. we have now added new data from the Swedish National Forest Inventory describing forest volume production for mesic pine and spruce forests in Sweden, divided into each sub-region we consider. These data show a long-term trend of increased growth, but deviation from this trend over the past one to two decades. We think these new data provide solid support that increased forest N demand is a viable mechanism that is enhancing N limitation over the long run and further suggest we have recently entered a period where growth appears to be increasingly constrained, which is also consistent with PNL predictions. We are not in a position to isolate and quantify the impact of PNL on C cycling, as this is very complicated, and is a task best taken on by ESM Modelers.

References:

- BassiriRad, H. et al. Widespread foliage δ¹⁵N depletion under elevated CO₂: inferences for the nitrogen cycle. *Global Change Biol* **9**, 1582–1590 (2003).
- Canadell, J. G. et al. *Global carbon and other biogeochemical cycles and feedbacks* (Chapter 5). *Climate Change 2021: The Physical Science Basis. Contribution of Working Group I to the Sixth Assessment Report of the Intergovernmental Panel on Climate Change*, 673–816. (Cambridge University Press 2021).
- Craine, J.M. et al. Isotopic evidence for oligotrophication of terrestrial ecosystems. *Nature Ecology & Evolution* **2**, 1735-1744 (2018).

Vitousek, P.M., Cen, X. & Groffman, P.M. Has nitrogen availability decreased over much of the land surface in the past century? A model-based analysis. *Biogeochemistry* **167**, 793-806 (2024).

REFEREE #4 (REMARKS TO THE AUTHOR):

Comment: Summary: The authors use measurements of the natural abundance of ^{15}N from tree rings carefully selected from across Sweden to show steady decreases in $\delta^{15}\text{N}$ over the last 60 years, regardless of the rate or form of incoming N deposition. Further statistical analyses show that these $\delta^{15}\text{N}$ declines are more strongly linked to increases in atmospheric CO_2 than with any other factor examined (N deposition, rising temperature, stand basal area). The authors attribute these $\delta^{15}\text{N}$ trends to increasing ecosystem N limitation driven by rising atmospheric CO_2 .

Response: We thank the reviewer for the accurate assessment of the study, and for nicely summarizing the main breakthrough of our work. The study and this conclusion directly address recent debates focused on the trajectory of terrestrial N limitation, and $\delta^{15}\text{N}$ chronologies specifically.

Comment: Review: I found this manuscript to very convincingly demonstrate that declines in plant $\delta^{15}\text{N}$ over time across Sweden are linked to rising atmospheric CO_2 concentrations rather than to other factors of environmental change, with particularly strong evidence against attribution of these $\delta^{15}\text{N}$ patterns to the rate or form of N deposition. The authors' selection of samples from across Sweden, while also carefully controlling for stand age class, makes this analysis much more robust than the small set of high-profile reviews and data syntheses that it builds on.

Response: We thank the reviewer for the overall assessment of the manuscript, and for recognizing the robustness of our methods and main conclusion.

Comment: The primary explanation proposed as the mechanism driving these ^{15}N trends in this and these other studies is the stimulation of N limitation by elevated CO_2 (eCO_2), building on a framework originally described in the literature as progressive N limitation (PNL, ref. 14 – Luo et al., 2004) and more recently as terrestrial oligotrophication (e.g., refs 1-Craine et al., 2018 Nature EE & 3-Mason et al., 2022 Science). The long-proposed ideas suggest that eCO_2 should drive ecosystem N limitation first by stimulating plant net primary production and accumulating N in plant biomass, and second, by increasing plant litter C:N ratios that should increase soil N accumulation, both with feedbacks that should further enhance N limitation. This manuscript does a nice job of reviewing these general ideas (line 195-202) and should be sure to include references to the earlier PNL framework. It also discusses a potential role for increased ^{15}N fractionation by ectomycorrhizae in contributing to the ^{15}N decline. Overall, this manuscript makes a solid contribution to the literature that improves on these prior assessments; it contains a well-crafted set of measurements and is nicely written.

Response: We thank the reviewer for the positive assessment of the manuscript and have added references to the earlier PNL framework by Luo et al. (2004) at lines 42, 214, 236, and 252.

Comment: I think that this manuscript could even go a step further, for greater impact with a small amount of additional text critically evaluating and discussing the potential processes underlying the $\delta^{15}\text{N}$ response to $e\text{CO}_2$ and its attribution to increased N limitation. First, the text can directly assess the first part of the N limitation hypotheses, pertaining to whether $e\text{CO}_2$ -stimulation of plant accumulation of N is really likely to be driving N limitation and the observed decrease in $\delta^{15}\text{N}$. If this mechanism is an important driver, tree-ring $\delta^{15}\text{N}$ should have a fairly strong negative relationship with stand biomass (or basal area, as used here). However, analyses here suggest a weak positive relationship (Fig. S2), which casts doubt on that explanation. Second, the manuscript should also acknowledge and discuss an alternative set of mechanisms by which $e\text{CO}_2$ has been proposed to affect plant N demand, tied to down-regulation of plant N uptake by $e\text{CO}_2$ (e.g., ref. 2-Hiltbrunner et al. 2019, references therein). These mechanisms could perhaps more directly and universally decrease plant N uptake in response to rising CO_2 , regardless of ecosystem N limitation or availability – as seems to be the case here, as Fig. 1c-j and 2b shows little to no effect of N inputs on the tree-ring $\delta^{15}\text{N}$ values.

Response: Excellent suggestions. Regarding the first point, our explanatory variables vary both across space and time (except for CO_2 , which only varies through time due to its high degree of atmospheric mixing), and thus both the spatial and temporal dimensions of the data can contribute to the relationships depicted in Figure 2, while holding other factors in the model constant. To better illuminate the spatial and temporal variability of these predictors, we have now added a new Supplementary Figure 2 based on a comment from Reviewer 3. A few things need to be taken into account when interpreting the “less important” predictors that were retained in our model. Firstly, despite the fact that our forward and backward selection approach with AIC selection criteria retained temperature, N deposition variables, basal area, species and species* CO_2 in the model (meaning the cost of retaining these predictors to the model was worth the small improvement in model fit), it is important to note that their explanatory power was extremely low (see partial R^2 values in Supplementary Table 1, 2, & 3).

Regarding the positive relationship between basal area and $\delta^{15}\text{N}$ depicted in Figure 2, we also have to keep in mind that both the spatial and temporal dimension of the data influence these relationships, as well as some co-linearity between basal area and other predictors (since the figure depicts relationships when other model factors are held constant). In general, basal area trends upward through time (Supplementary Figure 2), and $\delta^{15}\text{N}$ downward through time (Figure 1), indicating a negative temporal relationship at any given latitude. However, in our new Supplementary Figure 2, you can also see that the spatial relationship of basal area with latitude in Sweden generally follows a similar but less steep relationship as mean annual temperature, and likewise the temporal pattern of basal area exhibits a similar but less steep pattern as rising CO_2 and temperature. Also, basal area is the only predictor variable derived specifically from survey plots, as opposed to broader climate datasets that vary among 50 x 50 km grid cells. So, we think that the ability of basal area to account for some within grid cell variation in stand productivity is the main reason basal area was retained in our model, whereas temperature, and especially CO_2 , do a better job predicting the temporal patterns, even though basal area is positively related to both CO_2 and temperature changes through time (Supplementary Figure 2). For these reasons, we think it is not wise to put too much weight into the interpretation of the relationship between basal area and $\delta^{15}\text{N}$ depicted in Figure 2. We have now changed the results focused text at lines 188-194 as follows:

“Basal area was the only predictor variable derived from individual sample plots and thus accounted for plot-to-plot variation in stand productivity within grid cells, while also representing forest biomass variation with latitude and time (Supplementary Figure 2i,j). The weak positive relationship may reflect that higher fertility sampling plots within a given grid cell have higher basal area, after accounting for some degree of broad spatial or temporal co-linearity with temperature and CO₂, which were environmental factors with greater explanatory power in our model.”

Despite the low importance and positive relationship of basal area and $\delta^{15}\text{N}$ in our model, we think a better way to describe forest N demand is to report the pattern of forest growth through time (which is inherently related to basal area, but more explicitly describes growth). We have included a new figure (Supplementary Figure 3) that now shows the annual volume growth per ha of mesic pine and spruce forests in Sweden based on a wider range of forest inventory plots. These data show that growth of mesic pine and spruce forests similar to those from which we measured $\delta^{15}\text{N}$, has indeed increased through time in all parts of Sweden. This would result in higher forest N demand, consistent with PNL. We have now introduced this new data into the Implications section (lines 241-245), and more explicitly discussed increased N demand as a viable mechanism to explain $\delta^{15}\text{N}$ chronologies. We also highlight that this long-term trend of increased growth has flattened out during the last one to two decades. We also describe in the Implications section that this pause in growth is consistent with predictions by ESMs that increasing N limitation will eventually constrain growth, but then cautiously add that several other factors may also contribute to this recent growth deviations.

Regarding the second point, the reason we did not highlight the “photosynthetic down-regulation” idea proposed by Hiltbrunner et al. (2019), is because this mechanism mainly would only explain a decrease in foliar %N. Photosynthetic down-regulation does not provide any direct fractionation mechanisms itself, and within plant fractionation is considered to be very minor compared to fractionation related to soil/detrital N turnover processes and mycorrhizal transfer to plants (Enta et al. 2020; Gerhart & McLauchlan, 2014; Hobbie & Högberg, 2012; Kolb & Evans, 2002). Photosynthetic downregulation could indirectly influence plant $\delta^{15}\text{N}$ by decreasing plant N demand, however, this would steer $\delta^{15}\text{N}$ chronologies in the opposite direction than we observe, and thus does not provide a viable mechanism to explain the declining $\delta^{15}\text{N}$ trends we report. We have now mentioned this additional mechanism introduced by Hiltbrunner et al. (2019), and why plant $\delta^{15}\text{N}$ values are somewhat isolated from this issue (lines 122-124).

References:

- Enta, A. et al. Nitrogen resorption and fractionation during leaf senescence in typical tree species in Japan. *Journal of Forest Research* **31**, 2053-2062 (2020).
- Gerhart, L.M. & McLauchlan, K.K. Reconstructing terrestrial nutrient cycling using stable nitrogen isotopes in wood. *Biogeochemistry* **120**, 1-21 (2014).
- Hiltbrunner, E. et al. Data do not support large-scale oligotrophication of terrestrial ecosystems. *Nature Ecology & Evolution* **3**, 1285–1286 (2019).
- Hobbie, E.A. & Högberg, P. Tansley Review: Nitrogen isotopes link mycorrhizal fungi and plants to nitrogen dynamics. *New Phytologist* **196**, 367-382 (2012).
- Kolb, K.J. & Evans, R.D. Implications of leaf nitrogen recycling on the nitrogen isotope composition of deciduous plant tissues. *New Phytologist* **156**, 57-64 (2002)

Comment: One important technical question: as detailed below (see comment re. line 134-135), the slopes of relationships between d15N and CO2 concentration don't seem to make sense or to match Fig. 2a. I'm hoping that the authors can correct or clarify that discrepancy.

Response: It appears we have missed a negative sign for one of the slopes, and have now fixed this, so this is now consistent. Also, as another reviewer pointed out that we reported incorrect slope values here, and they are now fixed. Thanks for pointing out this error.

Detailed comments:

Comment: Line 42. Add a subject after "This ___".

Response: Done

Comment: Line 59. This explanation of the relationship between d15N and ecosystem N availability is useful as a broad description of one important set of processes that affect plant d15N (i.e. preferential loss of 14N under conditions of elevated N availability), but it is also somewhat overly simplistic, and neglects the roles of other processes that could be relevant here – especially those pertaining to 15N fractionation on N uptake by mycorrhizae or other N uptake and translocation processes. I appreciate the difficulty of trying to succinctly explain the many factors that can affect plant d15N in the Introduction, but brief acknowledgment is needed here that other factors also affect plant d15N.

Response: It is clear from this comment, and similar comments from other reviewers that we needed a more complete introduction of the dominant $\delta^{15}\text{N}$ fractionation processes in the Introduction; whereas, in the first draft this deeper description did not come until the Implications section. We have therefore shifted some of this background up into the introduction (Line 55 – 70), and we hope this is now clear.

Comment: Line 61-63. Reports of others' findings of decreasing d15N over time would be more informative if presented in more directly comparable units – preferably as rates of decline (per mil) per year or per decade. Doing so would also make these d15N declines more readily comparable with those reported in in this study in line 111-112.

Response: We have now standardized this to per mil per year for each reference here, so it is easier to compare across studies.

Comment: Line 63. Avoid using a dash as both negative and range: "decline of d15N globally (-0.06 to -1.61 per mil..." rather than "(0.06-1.6 per mil"

Response: Done!

Comment: Line 83-84. State the time period covered by this study somewhere in this section.

Response: Done!

Comment: Line 96-105. I applaud the authors for their structured sampling approach to control for tree species, location/N deposition history, and tree age class. This approach is a particular strength of this analysis, and makes it much more robust and convincing than some of the predecessors that it reviews.

Response: Thank you. We agree and feel that as a result this uniquely allows us to separate the debated influence of CO₂ versus N deposition on $\delta^{15}\text{N}$ chronologies, which we believe is the main breakthrough of our study.

Comment: Line 114-117. The fact that the slopes of ^{15}N values over time did not differ among regions with very different N deposition rates does indeed appear to support the authors' conclusion that the ^{15}N temporal trends are not driven by temporal patterns of N deposition.

Response: We agree, and it is the independent (orthogonal) relationship between atmospheric N deposition and CO₂, arising from the strong north-south gradient in N deposition, that is a major strength of our study system and data set. This uniquely allowed us to separate the debated drivers of $\delta^{15}\text{N}$ chronologies, which again, we believe is the main breakthrough of our study.

Comment: Line 114-117, 144-150. However, the very similar ^{15}N values and slopes across these regions receiving vastly different amounts of N deposition (Fig. 1c-j) would also seem to also suggest very direct little relationship between $\delta^{15}\text{N}$ and N availability in these forests (confirmed by Fig. 2b). Can the eCO₂-associated $\delta^{15}\text{N}$ decline be so confidently ascribed to eCO₂-driven increased N limitation when these relationships with N inputs are so poor?

Response: There certainly is abundant literature suggesting that N_r deposition increases soil N availability in other regions (e.g. in temperate regions near population centers), especially when input rates are very high. In Sweden, experimental additions of high rates of N fertilizers also clearly increase soil N availability. However, there has been numerous publications in Sweden investigating the impacts of simulated chronic N deposition on forests. Studies in Swedish forests show that simulated chronic N deposition comparable to N deposition rates in Swedish (typically $\leq 12 \text{ kg N ha}^{-1} \text{ y}^{-1}$) usually do not result in significant impacts on soil N availability, and trees access a very small portion of this added N (see Gundale et al. 2011, 2014). Significant ecosystem property or process responses to chronic N addition are usually found only at much higher levels of chronic N input (e.g. $\geq 50 \text{ kg N ha}^{-1}$; Blaško et al. 2022; Maaroufi et al. 2015, 2016, 2017, 2019; Forsmark et al. 2020a, 2020b, 2021, 2024; Hyvönen et al. 2007). When low level N inputs are applied to Swedish forests, most of the N is quickly incorporated into soil organic matter, with a very low proportion acquired by trees (Gundale et al. 2014). Thus, experimental evidence suggests the link between N deposition and enhanced soil N availability is indeed very weak in boreal forests. Further, it is important to note that N deposition rates in the northern half of Sweden are very low, and even here $\delta^{15}\text{N}$ chronologies decline at the same rate as southern Sweden. So, this pattern clearly indicates that rising CO₂ is the dominant driver of $\delta^{15}\text{N}$ chronologies.

References:

- Blaško, R. et al. The carbon sequestration response of aboveground biomass and soils to nutrient enrichment in boreal forests depends on baseline site productivity. *Science of the Total Environment* **838**, 156327 (2022).
- Gundale, M.J., From, F., Back-Holmen, L. & Nordin, A. Nitrogen deposition in boreal forests has a minor impact on the global carbon cycle. *Global Change Biology* **20**, 276-286 (2014).
- Gundale, M.J., DeLuca, T.H. & Nordin, A. Bryophytes attenuate anthropogenic nitrogen inputs in boreal forests. *Global Change Biology* **17**, 2743-2753 (2011).
- Hyvönen, R. et al. Impact of long-term nitrogen addition on carbon stocks in trees and soils in northern Europe. *Biogeochemistry* **89**, 121-137 (2007).

- Maaroufi, N. et al. Anthropogenic nitrogen deposition enhances carbon sequestration in boreal soils. *Global Change Biology* **21**, 3169-3180 (2015).
- Maaroufi, N., Nordin, A., Palmqvist, K. & Gundale, M.J. Chronic nitrogen deposition has a minor effect on the quantity and quality of aboveground litter in a boreal forest. *PLoS ONE* **11**, e0162086 (2016).
- Maaroufi, N., Nordin, A., Palmqvist, K. & Gundale, M.J. Nitrogen enrichment impacts on boreal litter decomposition are driven by changes in soil microbiota rather than litter quality. *Scientific Reports* **7**, 4083 (2017).
- Maaroufi, N. et al. Anthropogenic nitrogen enrichment enhances soil carbon accumulation by impacting saprotrophs rather than ectomycorrhizal fungal activity. *Global Change Biology* **25**, 2900-2914 (2019).
- Forsmark, B., Nordin, A., Maaroufi, N., Lundmark, T. & Gundale, M.J. Low and High nitrogen deposition rates in northern coniferous forests have different impacts on aboveground litter production, soil respiration, and soil carbon stocks. *Ecosystems* **23**, 1423–1436 (2020).
- Forsmark, B., Wallander, H., Nordin, A. & Gundale, M.J. Long-term nitrogen enrichment does not increase microbial phosphorus mobilization in a northern coniferous forest. *Functional Ecology* **35**, 277-287 (2020).
- Forsmark, B., Nordin, A., Resnstock, N.P., Wallander, H. & Gundale, M.J. Anthropogenic nitrogen enrichment increased the efficiency of belowground biomass production in a boreal forest. *Soil Biology Biochemistry* **155**, 108154 (2021).
- Forsmark, B. et al. Shifts in microbial community composition and metabolism correspond with rapid soil carbon accumulation in response to 20 years of simulated nitrogen deposition. *Science of the Total Environment* **918**, 170741 (2024).

Comment: Line 134-135. Please include the units for the slopes reported here in the text (-0.76 and -1.1) associated with Fig. 2, describing the relationships between tree-ring $\delta^{15}\text{N}$ and CO_2 concentrations. These slope units are presumably ‰ $\delta^{15}\text{N}$ per ppm CO_2 to match the units of the axes in Fig. 2a. However, slopes that steep don't match the $\delta^{15}\text{N}$ values in Fig. 2a or make sense. That is, for the CO_2 range of ~70 ppm shown in Fig. 2a (from roughly 315 to 385 ppm), these slopes should yield implausibly large $\delta^{15}\text{N}$ declines of -53 and -77 per mil $\delta^{15}\text{N}$, while the declines shown are the more reasonable range of < -6 per mil.

Response: The reviewer is correct about the units, and they are now added (now lines 161-162). Regarding the actual slope values, we thank the reviewer for pointing out this error. We have corrected this.

Comment: Line 159-163, 196-198. The observation reported here of a positive relationship between $\delta^{15}\text{N}$ and basal area (a proxy for biomass) is interesting, but hard to reconcile with the later partial attribution of the $\delta^{15}\text{N}$ trends over time to eCO_2 stimulation N accumulation in plants (lines 196-198). That is, if the mechanism by which eCO_2 is causing $\delta^{15}\text{N}$ to decrease is that increased accumulation of N in plant biomass reduces N availability, then the $\delta^{15}\text{N}$ -biomass relationship should not be positive but strongly negative. The authors might use this information to deepen the discussion of this hypothesis rather than suggesting it as a research part of the research frontier (line 233).

Response: We addressed this comment above. In short, basal area co-varies across space with the large temperature gradient in Sweden and varies through time with both temperature and CO_2 . We think the reason our selection approach (forward/backward selection using AIC criterion) included basal area is because it is the only included predictor that had unique plot

specific information (as opposed to grid-cell specific information), and thus was likely able to account for site-to-site variation in baseline fertility, that the broader scale environmental predictors estimated at the 50 x 50 km pixel level were not able to capture. The variable had a very small influence on our model (indicated by the very low partial R^2_m values), with a relatively uncertain slope. We thus regret that we may have put more interpretation into this response than was warranted. We have thus modified our interpretation of this variable to a more cautious tone in lines 182-197. Instead, we have now introduced new data from the Swedish National Forest Inventory showing annual growth patterns through time of mesic pine and spruce forests across Sweden. While this data is not available at the plot level, and cannot be incorporated into our model, they are more relevant for understanding forest N demand than basal area, since it is specifically a metric of growth. The data show long-term upward trends (discussed in the implications section at lines 241-248), with a deviation of this long-term trend occurring in the past one to two decades. As described above, we believe these data illuminate that increased forest N demand is a very likely driver of the intensified N limitation we describe, and we suggest that intensified N limitation is a feasible factor that contributes to a long-term asymptotic growth trend (lines 241-248).

Comment: Line 195-202. This section nicely reviews the two proposed mechanisms by which eCO₂ could stimulate N limitation (by increasing N accumulation in plant biomass through stimulated production, and increased C:N ratios of plant tissues driving soil N accumulation) that were clearly laid out several decades ago as part of the progressive N limitation framework (ref. 14, Luo et al., 2004). This attribution should be added.

Response: We have added this important reference. Additionally, we moved some of this background further up into the Introduction, which was requested by other reviewers.

Comment: Line 216-219, 223-226. Although some ectomycorrhizae can mine N from SOM, increased plant reliance on ectomycorrhizae for N uptake (with corresponding $\delta^{15}N$ fractionation) doesn't necessarily mean that those mycorrhizae are acquiring more N from SOM, only that they are responsible for a larger fraction of plant N.

Response: We fully agree. We have modified the sentence at lines 261-264 that declining $\delta^{15}N$ could indicate that a larger portion of tree N is being acquired via ectomycorrhizal fungi. About soil organic N mining, evidence is building that fungal communities shift towards taxa with peroxidase enzyme capacity when N limitation becomes more severe, and where the C cost of peroxidase based organic matter mining becomes worth the expense. Two highly relevant manuscripts on this are cited at lines 267 (Gundale et al. 2024; Lindahl et al. 2021), which provide examples of these ectomycorrhizal taxa responding to N limitation gradients in a variety of Swedish forests. We also agree that tapping into a more costly N pool probably results in less total N acquisition (because it is a more expensive, and a less available N pool), and likely also represents an allocation of carbon away from tree growth to the belowground system. These are really important dimensions, but we hesitate to expand the discussion to include these nuances, because other reviews suggested we should reduce the amount of speculation, and because of the expectation we keep the manuscript concise and accessible to a broad readership.

References:

Gundale, M.J. et al. The biological controls of soil carbon accumulation following wildfire and harvest in boreal forests: a review. *Global Change Biology* **30**, e17276 (2024).

Lindahl, B.D. et al. A group of ectomycorrhizal fungi restricts organic matter accumulation in boreal forest. *Ecol Lett* **24**, 1341-1351 (2021).

Comment: Line 229-230. In the cited meta-analysis, it's true that soil C "did not increase" in eCO₂ experiments with ectomycorrhizal plants, but soil C stocks didn't decrease, either (no change on average), although a decrease is implied by the suggestion (line 230) that eCO₂ is stimulating decomposition in these forests.

Response: Because eCO₂ stimulated NPP, higher C inputs to soil may be expected to promote soil C accumulation. So, what we mean here is that decomposition appears to keep pace with the higher C inputs derived from the higher NPP. We have clarified the language here (now at lines 267-269). Further, to better support the discussion here, we have now added supplementary data on forest growth in Sweden (1951-2020; Supplementary Figure 3).

Comment: Line 333-337. Somewhere in this section, state the sample size (or size ranges) for the species / decade / region combinations. If n = 1609 overall, and there are 2 species, 6 decades, and 4 regions, that would seem to be ~33 samples each on average?

Response: We have now stated this in the Methods section at line 541, where we discuss the regressions of the four regions.

Comment: Line 393-395. The 10-year segments of wood would generally need to be ground before weighing 10 mg subsamples. If so, please describe how; if not, please state how the 10 mg representative samples were obtained.

Response: The external lab we used to produce the $\delta^{15}\text{N}$ data (University of Maryland, Central Appalachians Stable Isotope Facility (CASIF)), uses a technique to shave thin radial slices from the 10-year increment cores, rather than sample grinding. These slices are then chopped up with a razor, weighed, and analyzed on the GC-MS. The lab communicated to us that this approach reduces the possibility of cross sample contamination, while still enabling complete sample combustion on the instrument. These details are added at line 477.

Comment: Line 409, 418. Specify the spatial resolutions of the gridded temperature and N deposition datasets.

Response: Done! For each dataset, the resolution was 0.5 x 0.5° (ca. 50 x 50 km) raster cells.

Comment: Line 424, 425. "The observed increase in forest basal area over time can be attributed to..." Observed and attributed by whom? Please add citation/s.

Response: We have added a citation here.

References:

Elfving, B. & Tegnhammar, L. Trends of tree growth in Swedish forests 1953-1992: An analysis based on sample trees from the national forest inventory. *Scandinavian Journal of Forest Research* **11**, 26-37 (1994).

Comment: Line 531-532. "Data sources are cited in Supplemental Information." Rephrase. This text is in the SI.

Response: Thank you for catching this. We meant the Methods section.

Comment: Table S1: Please convert table headers to text from what looks like model commands.

Response: We now have moved the statistical model formulation to a footnote in all Tables and use plain text in the header.

Comment: Figure 2. The slope values stated on these figures appear inconsistent with the regression lines plotted. Please see comment above re. line 134-135.

Response: This was indeed a mistake, and we have fixed the slopes.

Referees' comments:

Referee #1 (Remarks to the Author):

Comment: The authors have done a very thorough job of responding to the questions and concerns of four reviewers. For my part, I am satisfied with their response to my queries and concerns. I appreciate the thoughtfulness of the authors' responses. I believe this manuscript content is now acceptable. I have a few very minor editorial suggestions.

Response: We thank the reviewer for their positive assessment.

Comment: L 32 – expected to increase by a factor of 18

Response: We thank the reviewer for detecting this omission; it has been corrected.

Comment: L 61 – delete the second “with low delta15 N values”

Response: We thank the reviewer for detecting this repetition in the sentence; it has been corrected.

Comment: L 63 – reference 21 does not need brackets

Response: We thank the reviewer for this suggestion; we have now adjusted this to ^{ref. 21} to avoid confusing the reader that 21 was part of the chemical formula. This change now appears at L62.

Comment: L 78, 93 – “a study” should be replaced by the author names

Response: We thank the reviewer for this style suggestion; we have replaced this with the author's names.

Comment: L 227 – “including” should be replaced by “and”

Response: We thank the reviewer for detecting this error; it has been corrected.

Comment: L 240 – change to ... , and thus steer ...

Response: We thank the reviewer for detecting this omission; it has been corrected.

Comment: L 245 – “In recent decades, forest growth may have levelled off...”. There is no analysis of this trend and it is not always clear, especially in northern regions and for *Picea abies*.

Response: We thank the reviewer for this comment; we have now rescaled the axes to make the trends clearer; this figure is now included in the main text (Figure 3). Regarding analysis, we have applied simple linear regressions to these figures, which are reported in the upper left-hand corner of each sub-panel. This analysis is meant to determine whether growth has significantly increased through time or not. It shows that for both species in all regions, growth has significantly increased through time. Regarding “levelling off”, this is now more visible with the rescaling, where it can be seen that seven of the eight growth curves have decelerated in the past several decades (Figure 3a,b).

Comment: L 271 – reference 61 does not need brackets

Response: We thank the reviewer for this suggestion; we have now adjusted this to ^{ref. ##} to avoid confusing the reader that the number was part of the chemical formula. The reference number has also changed and this now applies in two sentences at L301, 305.

Comment: L 279 – change to “to the control of delta15 N values ...”

Response: We have modified this statement slightly differently, as follows: “Determining the relative importance of these two mechanisms as controls on $\delta^{15}\text{N}$ values in across different boreal forest

contexts...” This change now appears at L308.

Comment: L 280 – compartments

Response: We thank the reviewer for detecting this error; it has been corrected.

Referee #2 (Remarks to the Author):

Comment: The authors have comprehensively addressed all of my comments from the previous version of the manuscript and I recommend acceptance of the manuscript.

Response: We thank the reviewer for their previous insightful comments and thank them for their positive recommendation.

Referee #3 (Remarks to the Author):

Comment: The authors have done an excellent job of responding to the reviewers' comments. The revisions have strengthened the paper considerably and I now find this to be a very novel and convincing piece of work. I have no further suggestions for edits to the manuscript.

I've also taken a look at the data shared. This appears to be complete but lacks meta-data. I would strongly recommend adding some meta-data to indicate what the data are (would include an explanation of the contents of each sheet, including explanation of each column with units).

Response: We thank the reviewer for this suggestion; we have now added an additional worksheet to the file with meta-data describing the variables.

Referee #4 (Remarks to the Author):

Comment: I was reviewer #4 for the previously submitted version of this manuscript, so I will not repeat my prior summary of the manuscript, which has changed relatively little from the previous version. As in my prior review, I am convinced by the first part of the authors' conclusion, that the study elegantly shows that elevated (eCO₂) is the main driver of decreases in wood N isotopic composition (δ¹⁵N) in trees across Sweden over the last ~60 years. Both versions of this manuscript nicely address earlier critiques of a set of high-profile analyses (mainly by a recurring set of authors; manuscript refs. #s 1, 3, 4, 20, 21, 22, 24, 43, 44), that have made similar arguments in interpreting other plant δ¹⁵N measurements, but did not previously disentangle the role of eCO₂ from that of nitrogen deposition amount or composition as potential drivers of these δ¹⁵N trends, as the present analyses does very effectively. Resolving that debate and confirming a primary role of eCO₂ in driving plant δ¹⁵N trends makes a solid contribution to the literature, and I looked forward to citing those results.

Response: We thank the reviewer for this assessment and agree that the major breakthrough of our study is to clearly identify that CO₂ is the strongest predictor of wood δ¹⁵N values. We appear to be in agreement regarding the most central part of the manuscript but disagree on whether changes in δ¹⁵N values reflect changes in N limitation. To more directly address the reviewers concern, we have followed their suggestion and now provide statistical evidence for a direct relationship between temporal change in forest growth and δ¹⁵N values. We have now included a new supplementary table with the analysis results (Table S4), and include two new sub-panel figures, which we have now combined with the long-term forest growth data as a new Figure 3 in the main manuscript.

Comment: That said, I was disappointed by the lack of response to my main content-based requests of

the prior manuscript, and I am not yet convinced by the second part of the manuscript's conclusion as summarized in its title, that these eCO₂-driven d¹⁵N trends necessarily mean that eCO₂ has intensified N limitation. It could be so, but I don't believe that the manuscript has yet demonstrated intensified N limitation of plant growth. As reviewers are asked to review statistics: I believe that this part of the conclusion also warrants support from statistical analysis rather than the qualitative support included so far.

Response: We are happy to accommodate the reviewer here and provide more statistical evidence for a direct relationship between temporal forest growth change and wood δ¹⁵N values. This is a challenging relationship to model, because forest basal area and forest growth exhibit much more spatial (i.e. latitude) than temporal variation. In order to better isolate the temporal aspect of the forest growth data, we have now calculated a new relative forest volume growth variable (which we refer to as "relative forest volume growth change"), where forest growth estimates from each plot are calculated relative to the grid cell forest growth average of all decades for each tree species. We then constructed two additional linear mixed effects models to evaluate the relationship between this relative forest volume growth change variable and wood δ¹⁵N values, where we also include either absolute plot basal area or absolute forest volume growth as fixed factors, and grid cell as a random factor. These models show highly significant negative relationships between relative forest volume growth change and wood δ¹⁵N values, in line with the reviewer's suggestion. Simply put, this means that when absolute forest basal area or volume growth are held constant, the change in growth through time exhibits a significant negative relationship with wood δ¹⁵N. We have now added a new supplementary table (Table S4) and figure panels (in a new Figure 3) depicting these relationships and introduce new text in the Implications section of the manuscript on this relationship (L255). We also mention here that this relationship contrasts with the suggestion that leaf-level photosynthetic downregulation is a dominant process that controls forest-level N demand under rising CO₂. Indeed, photosynthetic downregulation of N at the leaf level may be occurring simultaneously to increasing N demand at the canopy level due to higher growth, biomass accumulation, and leaf area. We agree that these are not mutually exclusive processes. The strong negative relationship we now show between relative forest volume growth change and δ¹⁵N values suggests that increasing N demand at the stand or hectare level is an influential process contributing to PNL. Regarding the figures, we have combined them as sub-panels with the long-term forest growth data and moved this new multi-panel figure into the main document, which is now Figure 3. We thank the review for this suggested analysis and agree that it strengthens the manuscript.

Comment: The manuscript's conclusion is framed in the context of the longstanding hypothesis in ecosystem ecology of progressive N limitation (PNL) that is also embedded in most earth system models (ESMs), and expects that eCO₂ should induce terrestrial N limitation by stimulating accumulation of N in plant biomass and/or soil organic matter. The manuscript's conclusions assume a direct relationship between plant d¹⁵N and plant N limitation, and although correlations between plant N concentration and d¹⁵N do regularly occur, these relationships are far noisier and more complicated than portrayed in the text or in the authors' Response (e.g., see comments re. lines 71-73 below). The disconnect between d¹⁵N and plant N responses is further illustrated by the strong responses of d¹⁵N shown over time (Fig. 1) that does not occur in tree-ring %N (Fig. S4), which shows no declines over time in any species or region, a relevant result that the main text does not mention (Fig. S4) and would seem to contradict the manuscript's conclusion that the observed decline in wood d¹⁵N is due to increased N limitation.

Response: We agree that meta-analysis relationships we reported in our previous response letter between δ¹⁵N and foliar %N are noisy, but these analyses are also highly replicated and exhibit highly significant relationships. Our mention of these relationships in the manuscript was specifically: "The

spatial relationship between plant $\delta^{15}\text{N}$ values, foliar N content, and N availability are now well established and accepted^{1,22,23}". This statement is based on the high statistical certainty of the relationships in these meta-analyses, as indicated by their highly significant p-values ($p < 0.001$ in both Chen et al. (2025), and Craine et al. (2018)). Thus, we maintain that this statement is very well supported by existing data. We were highly aware of the "noisiness" of $\delta^{15}\text{N}$ data when initiating our study, which was our main rationale for targeting a large sample size, which combats this.

Regarding the comment on the disconnect between $\delta^{15}\text{N}$ and wood %N in our data, we did address this in our previous response letter. In the field of dendroecology, wood %N is not generally considered a good trait to indicate a plant's N limitation status, because its concentration is very low in wood (averaging ca. 0.08% in our samples), such that small changes in more abundant wood elements (e.g. C or O) can drive substantial variation in wood %N values. We would also like to highlight a recent study by Thurner et al. (2025) which showed very clearly that N contents of leaves are much more responsive to variation in climate factors than the N contents of wood, generalized across many northern forest tree species and locations. This analysis supports our point that wood %N is not an ideal trait to focus in order to address the objectives of our study.

In contrast to wood %N values, $\delta^{15}\text{N}$ is a ratio of a ratio ($^{15}\text{N}/^{14}\text{N}$ of plant tissue relative to the atmosphere), and thus provides an internal standard for assessment of N cycle change. It varies little across plant organs (Enta et al. 2020; Gerhart and McLauchlan, 2014; Hobbie & Högberg, 2012; Kolb & Evans, 2002), and its interpretation is well supported by the meta-analysis mentioned above, as well as numerous review manuscripts (Craine et al. 2009, 2018; Chen et al. 2025; Gerhart & McLauchlan, 2014). In the previous draft we added the following statement at line 494: "Our analysis focuses on $\delta^{15}\text{N}$ rather than %N, given that %N of wood is not considered a good indicator of plant N limitation^{21,64}. We have now moved this statement further up into the narrative to L102, and amended the statement as follows: "We focused on $\delta^{15}\text{N}$ rather than %N, given that %N of wood is not considered a good indicator of plant N limitation due to its very low concentration and subsequent sensitivity to concentration variability of more abundant wood elements (e.g. C and O) (Gerhart & McLauchlan, 2014; Doucet et al. 2011; Bukata & Kyser 2005; Poulson, Chamberlain & Friedland 1995; Thurner et al. (2025))." This statement now includes additional relevant references in the text including Bukata & Kyser (2005), Poulson, Chamberlain & Friedland (1995) as well as the recent Thurner et al. (2025) study mentioned above.

Bukata, A.R. & Kyser, T.K. Response of the nitrogen isotopic composition of tree-rings following tree-clearing and land-use change. *Environ Sci Technol* **39**, 7777–7783 (2005).

Chen et al. Global mycorrhizal status drives leaf $\delta^{15}\text{N}$ patterns. *Journal of Ecology* **5**, 1150-1163 (2025).

Craine, J.M. et al. Isotopic evidence for oligotrophication of terrestrial ecosystems. *Nature Ecology & Evolution* **2**, 1735-1744 (2018).

Doucet, A., Savard, M.M., Bégin, C., Smirnoff, A. Is wood pre-treatment essential for tree-ring nitrogen concentration and isotope analysis? *Rapid Commun. Mass Spec.* **25**, 469–475 (2011).

Enta, A. et al. Nitrogen resorption and fractionation during leaf senescence in typical tree species in Japan. *Journal of Forest Research* **31**, 2053-2062 (2020).

Gerhart, L.M. & McLauchlan, K.M. Reconstructing terrestrial nutrient cycling using stable nitrogen isotopes in wood. *Biogeochemistry* **120**, 1-21 (2014).

Hobbie, E.A. & Högberg, P. Tansley Review: Nitrogen isotopes link mycorrhizal fungi and plants to nitrogen dynamics. *New Phytologist* **196**, 367-382 (2012)

Kolb, K.J. & R.D. Evans. Implications of leaf nitrogen recycling on the nitrogen isotope composition of deciduous plant tissues. *New Phytologist* **156**, 57-64 (2002).

Poulson, S.R., Chamberlain, C.P. & Friedland, A.J. Nitrogen isotope variation of tree rings as a potential indicator of environmental change. *Chem Geol* **12**, 307–315 (1995).

Thurner, M., K. Yu, S. Manzoni., A. Prokushkin, M.A. Thurner., Z. Wang., and T. Hickler. Nitrogen concentrations in boreal and temperate tree tissues vary with tree age/size, growth rate, and climate. *Biogeosciences* **22**,1475-1493 (2025).

Comment: The authors have now added some tree growth data (Fig. S3) that seemingly could be used (in addition to or instead of basal area) to test the ‘enhanced growth’ and/or ‘growth limitation’ part of the PNL hypothesis, but indicate in their Response that adding these data to the overall analysis of $\delta^{15}\text{N}$ drivers would be beyond the scope of the present study. I believe that choice reduces the study’s overall impact and the strength of its conclusions. In the manuscript text (line 239-247) and their response, the authors argue that the independent tree growth trend analysis they have added (Fig. S3) supports their interpretation of PNL, because these data do indeed show growth increases over time for both species. But in order to support the manuscript’s interpretation of PNL-driven N limitation, there should be a primary, direct relationship between the increase in growth (or basal area) and resulting decrease in N availability, here as inferred from $\delta^{15}\text{N}$, and that that relationship should be strong at both the local (plot) and regional scale if it is the main underlying mechanism by which eCO_2 induces N limitation. Without including that direct analysis, the comparison between Fig. 1 and Fig S2i or Fig S3 is qualitative and subjective. It appears that the timing and magnitude of growth stimulation in Fig. S3 varies greatly by region and tree species, in contrast to the uniformity of the decline in $\delta^{15}\text{N}$ across regions highlighted by the manuscript. It may have been beyond the scope of the author’s original intent, yet such analysis would provide the powerful test of the longstanding PNL hypothesis and ESM assumptions that the author’s title rests on, and may well make subsequent conclusions of high impact -- whether they support the PNL ideas as presently stated in the title or point to the importance of other mechanisms.

Response: We agree that such a direct analysis is useful, as it helps us weigh the potential importance of enhanced plant demand as an influential PNL mechanism. To repeat our response above, we have now calculated a new relative forest volume growth change variable, where forest growth data from each plot are calculated relative to the grid cell forest growth average, for each tree species. This new variable describes forest biomass growth change. We then constructed two additional linear mixed effects models to evaluate the relationship between this standardized forest growth variable and wood $\delta^{15}\text{N}$ values, where we also include either absolute plot basal area or absolute volume growth as fixed factors, and grid cell as a random factor. This analysis approach effectively holds constant the positive latitudinal relationship between forest volume growth or basal area and $\delta^{15}\text{N}$ (i.e. warmer southern environments exhibit both higher $\delta^{15}\text{N}$ values and higher growth and basal area) and isolates the temporal dimension of the data. These models show significant negative relationships between relative forest volume growth change and wood $\delta^{15}\text{N}$ values. In plain language, this means that for a given level of absolute forest basal area or annual volume growth, the change in growth through time exhibits a significant negative relationship with wood $\delta^{15}\text{N}$. We have now added a new Supplementary Table (Table S4) and figure sub-panels depicting these relationships (new Figure 3) and introduced new text in the Implications section of the manuscript.

Comment: The other prominent mechanism for eCO₂-driven suppression of plant N uptake comes from plant physiology, which routinely shows that eCO₂ allows for greater photosynthetic N use efficiency, allowing the same or even greater photosynthesis in response to eCO₂ accompanied by reduced N uptake (e.g., Stocker et al., 2025, *New Phytologist* 245:49-68 and references therein provide a nice new review), and that appears to be a strong and relatively direct effect of eCO₂ on plant δ¹⁵N that manifests regardless of plant functional type (ref. 45, BassiriRad et al., 2003), though I believe that it is unclear whether that increased fractionation occurs on uptake or within the plant. Downregulation of N uptake and PNL are not mutually exclusive drivers of plant N responses to eCO₂, but they have fundamentally different implications for the future of the terrestrial C sink and its potential constraint by N limitation, which is expected by PNL, but not from downregulation of plant N uptake. At the very least this alternative explanation merits greater recognition and discussion in the text, as requested by me and several other reviewers.

Response: We do not dispute that eCO₂ can enhance photosynthetic nitrogen use efficiency and might even result in photosynthetic downregulation of foliar N as a result of this. However, none of the suggested references, including an additional recent relevant PNAS publication by Bassiouni et al. (2025), provide a mechanism by which internal leaf downregulation can steer δ¹⁵N chronologies negative. Neither the Stocker et al. (2025) nor Bassiouni et al. (2025) mention internal plant isotope fractionation in response to rising CO₂. BassiriRad et al. (2003) suggested that internal plant nitrate reductase might be a mechanism that can explain δ¹⁵N of plants in response to eCO₂ (along with less speculative ecosystem-level PNL-based mechanisms). However, this mechanism is not plausible for boreal forests because nitrate production (i.e. nitrification), soil nitrate accumulation, and plant nitrate uptake are general minor fluxes in boreal forests (Sponseller et al. 2016). We also highlight that the BassiriRad reference only mentions bulk soil δ¹⁵N, which cannot be considered a good indicator of plant available N over the short timescale of a FACE experiment.

The physiological downregulation perspective on foliar N raised by the reviewer is well articulated by a recent manuscript by Bassiouni et al. (2025) published in PNAS. The study showed that leaf level N contents are declining in Europe through time and suggested (primarily on a theoretical basis, without direct empirical proof) that this is due to physiological downregulation of Rubisco in leaves, which is occurring because plants are able to use Rubisco more efficiently under high CO₂. The main evidence Bassiouni et al. (2025) provide for this hypothesis is that observed temporal foliar N concentrations conform to predictions from physiological optimum allocation theory. Here we mention that these patterns also conform to the PNL framework, which was not addressed in the Bassiouni et al. (2025) manuscript, because their theoretical framework assumes no depletion of soil N supply. Bassiouni et al. (2025) further argue that physiological foliar N downregulation by plants suggests plants are not actually experiencing progressive nitrogen limitation but are rather achieving the same level of photosynthesis with less N, as part of an optimization strategy. As the reviewer notes, this idea and PNL are not mutually exclusive. We therefore do not disagree that forests might be downregulating Rubisco and foliar N at the leaf level in response to rising CO₂ and look forward to future direct proof demonstrating this.

However, we highlight that the new analyses we have added showing a strong negative relationship between relative forest volume growth change and wood δ¹⁵N suggests that increasing biomass N demand at the hectare level offsets or outweighs any leaf level N downregulation that may be occurring. We also highlight that photosynthetic down-regulation does not provide a clear explanation for the core data in our manuscript, i.e. the negative δ¹⁵N chronology. Firstly, photosynthetic downregulation would reduce plant demand and uptake, causing the pool of available soil N to

accumulate (i.e. reduced demand relative to supply). This would cause more available soil N to enter loss pathways (e.g. nitrification, denitrification, and leaching), and cause plants to depend less on mycorrhizae. Both of these responses would steer $\delta^{15}\text{N}$ chronologies in a positive direction through time, instead of the negative direction that we observe. Secondly, photosynthetic downregulation cannot explain the negative relationship between relative forest volume growth change and $\delta^{15}\text{N}$, which we now include as a new figure (Figure 3), based on the reviewer's suggestion. This negative relationship suggests that a net increase in forest N demand is driving wood $\delta^{15}\text{N}$ values more negative, which is inconsistent with the photosynthetic down-regulation explanation. Even if some degree of photosynthetic downregulation is occurring at the individual leaf level, this appears to be counterbalanced by the increase in forest N demand at the canopy, plot or hectare level, where CO_2 has been shown to increase the leaf area index, standing biomass, and growth. The Stocker et al. (2025) manuscript mentioned above very well supports that eCO_2 causes an increase in vegetation growth and biomass (see their Figure 3c). Finally, and in relation to the previous point, photosynthetic downregulation of leaf-level N concentrations cannot explain why aquatic exports from boreal forest ecosystems are decreasing through time, whereas the $\delta^{15}\text{N}$ evidence we provide for PNL provides exactly the evidence needed to explain these trends. Here we highlight three references from the Swedish landscape that show concurrent aquatic oligotrophication linked to increasing vegetation growth (Lucas et al. 2016; Nilsson et al. 2024, Goedkoop et al. 2025), two of which specifically show N exports are declining in synchrony with increasing vegetation growth (Lucas et al. 2016, Goedkoop et al. 2025). Thus, we maintain that our original $\delta^{15}\text{N}$ data, combined with the newly added relationships between forest growth and $\delta^{15}\text{N}$ provide robust evidence for ongoing PNL in boreal forests, which is a highly novel finding. We thank the reviewer for pushing us to add the new analysis of forest growth change and wood $\delta^{15}\text{N}$ values, which we agree strengthens the manuscript.

Bassiouni, M., Smith, N.G., Reu, J.C., Peñuelas, J. & Keenan, T.F. Observed declines in leaf nitrogen explained by photosynthetic acclimation to CO_2 , *Proc. Natl. Acad. Sci. U.S.A.* **122**, e2501958122, (2025).

Goedkoop, W., Adler, S., Huser, B., Gardfjell, H. & Lau, D.C.P. Climate Change-Induced Landscape Alterations Increase Nutrient Sequestration and Cause Severe Oligotrophication of Subarctic Lakes. *Global Change Biology* **31**, e70314 (2025).

Nilsson, J.L., Camiolo, S., Huser, B., Agstam-Norlin, O. & Futter, M. Widespread and persistent oligotrophication of northern rivers. *Science of the Total Environment* **955**, 177261 (2024).

Lucas, R., Sponseller, R. Gundale, M.J., Stendahl, J., Fridman, J., Högberg, P. & Laudon, H. Long-term declines in stream and river inorganic nitrogen (N) export correspond to forest change. *Ecological Applications* **26**, 545-556 (2016)

Sponseller, R.A., Gundale, M.J., Futter, M., Ring, E., Nordin, A., Näsholm, T. & Laudon, H. Nitrogen dynamics in managed boreal forests: recent advances and future research directions. *Ambio* **45**, 175-187 (2016).

Detailed comments:

Comment: Title: As noted above and in my prior review: so far the manuscript convincingly demonstrates that eCO_2 affects plant $\delta^{15}\text{N}$, but not that eCO_2 is intensifying N limitation of plant growth.

Response: We feel that the addition of the new analysis on the relationship between relative forest growth volume change and $\delta^{15}\text{N}$, as requested by the reviewer, clearly justifies the original title. We thank the reviewer for asking us to include this analysis, and hope the reviewer is now satisfied.

Comment: Line 60-61: typo -- “at low d15N values” is repeated twice in this sentence

Response: We thank the reviewer for detecting this repeat in the sentence; it has been corrected.

Comment: Line 66: rephrase “only small isotope discrimination”

Response: We thank the reviewer for this comment; we have now rephrased this sentence.

Comment: Line 69: I suggest adding “can” before “steer plant d15N”

Response: We thank the reviewer for this comment; we have now rephrased this sentence.

Comment: Line 71-73: The relationships among plant d15N, foliar N, and [soil] N availability have been asserted by the refs cited (1 – Craine et al., 2018; 22 – Craine et al., 2009; 23 – Chen et al., 2025; 20 Gerhart & McLaughlin 2014), as well as various others. But these relationships are not nearly as clean and tightly coupled as implied by this text. These variables do frequently emerge as significantly correlated, but often rather weakly. For example, the two figures from refs. 1 & 23 provided by the authors (p. 8 of their Response) illustrate relationships between foliar d15N and foliar N in two data syntheses as very large clouds of data points (Ref. 1 $R^2 = ?$; Ref. 23 $R^2 = 0.13$). Many factors affect plant d15N values in addition to these particular drivers. Greater recognition of the uncertainty and additional drivers of plant d15N is warranted here and throughout the manuscript.

Response: Our exact wording here was “The spatial relationship between plant $\delta^{15}\text{N}$ values, foliar N content, and N availability are now well established and accepted”. We acknowledge that the relationships in the cited references are noisy (indicated by low R^2 values), but we do not remark on their strength and instead are referring to the high certainty of the relationships, indicated by their p-values. Both Chen et al. (2025) and Craine et al. (2018) report $p < 0.001$, indicating a high degree of certainty that foliar $\delta^{15}\text{N}$ and foliar N exhibit a positive relationship. Thus, we respectfully do not feel that our statement is controversial in any way and is very well supported by these meta-analyses.

Comment: Line 123-125: This new sentence, asserting that focusing on wood d15N rather than foliar N concentration avoids effects of downregulation doesn’t make sense on several levels. Firstly, earlier in the manuscript, the authors state that plant d15N is strongly associated with plant tissue N concentrations, such that wood d15N acts as a reliable index of plant and soil N availability (lines 71-73). Although I don’t think those relationships are as tight as stated (above), that is the logical framework provided in the manuscript so far, such that potential decoupling of these variables here requires more explanation and justification. Secondly, if downregulation – which should be described more explicitly in the text in general – was reducing plant N uptake, this process may directly reduce both plant d15N as well as plant N concentration (e.g., ref. 45, the BassiriRad et al., 2003 meta-analysis showing consistent reductions in plant d15N across plant functional types in response to eCO₂ experiments, with no relationship to soil d15N).

Response: Our exact statement at lines 123-125 were “Finally, our focus on wood $\delta^{15}\text{N}$ avoids concerns raised that foliar %N chronologies may not be a good indicator of PNL, due to possible photosynthetic downregulation in response to rising CO₂⁽²⁾”. Here we are addressing what we respectfully believe is a weakness in the argument that photosynthetic down-regulation is a mechanism that can explain negative plant $\delta^{15}\text{N}$ chronologies. If photosynthetic down-regulation is a factor reducing foliar N concentrations, and therefore plant N demand, it does not provide a direct mechanism explaining $^{15}\text{N}/^{14}\text{N}$ fractionation. It may indirectly affect plant isotope ratios by reducing

plant N demand, which would result in less mycorrhizal uptake, and greater ecosystem N losses. However, if this response pathway occurred, it would steer $\delta^{15}\text{N}$ chronologies in the opposite direction than we observe. Moreover, a decline in leaf-level N concentrations does not necessarily equate to reduced total plant N demand, as $e\text{CO}_2$ often increases total leaf area, NPP, and biomass (See Stocker et al. 2024, Figure 3c for meta-analyses). We do not see any logical way in which photosynthetic down-regulation can *directly* alter the N content or isotope ratio of wood, as this mechanism is specific to leaves. We therefore respectfully believe this statement is accurate and necessary. We have modified the statement to make this point even more clear. We have now changed this sentence (L126-129) to: “Finally, our focus on wood $\delta^{15}\text{N}$ avoids concerns raised regarding the interpretation of foliar %N chronologies as an indicator of PNL, which have alternatively been suggested to result from photosynthetic down-regulation instead of PNL, since this alternative explanation would not directly affect wood nitrogen isotope ratios².”

Comment: Fig. 1-i should use a dashed line rather than solid (or none at all), as this relationship was not statistically significant.

Response: We thank the reviewer for this comment; we have changed non-significant relationships from a solid to a dashed line in both the suggested Figure 1i as well as where it is applicable elsewhere (Figure S3a-h).

Comment: Line 233 (and lines 254, 279): An additional important mechanism linking $e\text{CO}_2$ and $\delta^{15}\text{N}$ is down-regulation of plant N uptake. That process could be acting in addition to or separately from PNL, but has very different implications for the future in terms of N limitation of future plant growth. It deserves explicit recognition and discussion in the main text.

Response: As we clearly addressed in the previous revision, this mechanism cannot explain a negative $\delta^{15}\text{N}$ chronology. Reduced N demand by the forest would result in an accumulation of available soil N, which would then be subjected to greater fractionating losses, as well as less fractionation during transfer via ectomycorrhiza. Both of these changes would steer $\delta^{15}\text{N}$ chronologies positive, rather than negative. Further, this mechanism could not explain the reported decreases in aquatic N exports in Swedish forest landscapes, or the negative relationship between forest growth change and $\delta^{15}\text{N}$. So, in short, this mechanism is inconsistent with the data. Also, as noted above, even if photosynthetic downregulation is occurring (for which we look forward to more solid proof), total forest N demand can still be increasing due to the long-term increase in forest biomass and higher growth on a per hectare basis, which acts as a stronger N demand and sink. Given the many comments by the reviewer on photosynthetic downregulation of N, we have now more explicitly referred to this hypothesis (L 128,259) and explain why our data do not fit this explanation (L258-267).

Comment: Line 241-245: This added tree growth data is potentially important for making a potential case for a PNL-driven mechanism for the observed decline in wood $\delta^{15}\text{N}$. I encourage moving these data from the supplement (Fig. S3) to the main text, and as noted previously, I believe that they should be used to directly test, if possible, to what extent the observed $\delta^{15}\text{N}$ trends are associated with changes in wood production in the overall model, as assumed by PNL theory and interpreted by the manuscript.

Response: As stated above, we have now analyzed the relationship between forest growth and wood $\delta^{15}\text{N}$ data. This was challenging, because forest growth varies substantially across the 1500km north south climate gradient in Sweden. To bring the temporal dimension of this data forward more clearly, we first created a new “relative forest volume growth change” variable, where the growth data in each grid cell is relativized to the mean growth value for each species in each grid cell. We then predicted $\delta^{15}\text{N}$ values using this variable, as well as absolute forest volume growth or absolute basal area (in two

separate models), and grid cell as a random factor. Accounting for the large spatial variation across Sweden through this approach revealed that temporal changes in forest volume growth are negatively related to wood $\delta^{15}\text{N}$ values. We have now created figures for these two models and combined them with the forest growth data that was previously in the supplement. Together, these sub-panels form a new Figure 3 in the main manuscript.

Comment: Line 282. Rephrase “ESM models” (the M = “model”)

Response: We thank the reviewer for detecting this error; it has been corrected.

Comment: Line 499. Is the spatial resolution of the temperature data really 5 degrees, as stated? If so, that resolution would appear to be too coarse for use in this application.

Response: We thank the reviewer for catching this mistake. We incorrectly referred to the database CRUTEM, whereas the actual dataset was CRU TS v4.07, which is hosted by the same research unit (which was the source of our confusion). The CRU TS database we used for analysis has a resolution of $0.5^\circ \times 0.5^\circ$. We have corrected this and also updated the reference to Harris et al. (2020).

Harris, I., Osborn, T. J., Jones, P., & Lister, D. Version 4 of the CRU TS Monthly High-Resolution Gridded Multivariate Climate Dataset. *Sci Data*, 7 (2020).

Comment: Line 582. This text refers to “Supplementary Figure 4, 5”, but there is no Fig. S5 in the materials I received for review.

Response: We thank the reviewer for detecting this error; the error has now been corrected to properly refer to the Supplementary Figures; as the reviewer correctly pointed out, there is no Suppl. Figure 5. Furthermore, we have now moved one supplementary figure into the main text, so this required further renumbering of the Supplementary Figures.

Comment: Fig. S2 caption: basal area units are in m^2 (not m^3)

Response: We thank the reviewer for detecting this error; it has been corrected in the caption.

Comment: Fig. S2: These panels are interesting but would be more informative if they included information on the slopes and strengths of the plotted relationships.

Response: We thank the reviewer for this good suggestion; we have made the included additional information.

Referees' comments:

Referee #4 (Remarks to the Author):

Comment: From the start, this manuscript has elegantly and convincingly demonstrated that rising atmospheric CO₂ concentration is the primary driver behind decreasing wood δ¹⁵N trends over nearly seven decades in the boreal forests across Sweden, rather than resulting from shifting amounts or forms of nitrogen deposition as others have proposed. I believe these analyses would be widely accepted as providing evidence that rising CO₂ is increasing N scarcity or “N oligotrophication” in these forests. This insight on its own is an important contribution to the literature, and I again support publication of the analyses.

My primary reservation all along has not been about those analyses, but the declaration that the study demonstrates “N limitation” of boreal forest tree growth, in the title and a handful of key times in the Discussion. It's the reason why analyses of growth were needed. N limitation is indeed possible and even a likely response, but it is another step beyond what is shown by the tree-ring δ¹⁵N or the new δ¹⁵N v growth analyses (more below). The manuscript makes a solid case for showing indications of decreasing N availability, yet there is a wealth of active research on the extent to which rising CO₂ could allow plants to partly offset that decrease in N availability by, e.g., by fueling more mycorrhizal-mediated N acquisition or by increasing plant nutrient use efficiency. (Here I'll note that I think the authors have now sufficiently addressed the recently prominent, if – I agree – also hard to test, idea of plant downregulation of leaf N). I believe the distinction of “N limitation” vs “reduced N availability” (or similar term) matters here because it's the “limitation” aspect of these changes to the N cycle that is so essential for predicting the future of the terrestrial carbon sink and the extent to which it will be constrained by nutrient availability.

This revised version of the manuscript includes an additional set of analyses that provide novel and powerful support for the first part of the longstanding progressive N limitation hypothesis, described well in the manuscript as its main conceptual underpinning to explain its CO₂-δ¹⁵N trends. However, that first part of PNL pertains to growth stimulation by CO₂, and how that should decrease N availability; support for next part of the hypothesis pertaining to N limitation in response to that N scarcity are qualitative and indirect – but tantalizingly close at hand. That is, PNL proposes that (1) increasing atmospheric CO₂ should initially stimulate plant growth and plant demand for nutrients, and that this demand (or increased tissue C:N ratios) should reduce soil nutrient availability; and then (2), this reduced soil nutrient availability should limit further CO₂-stimulation of growth. The new analyses (Figs. 3c & 3d), appear to show that across the whole period of record, there is a discernable trend that locations and times with the largest increases in tree growth have the lightest δ¹⁵N values, providing exciting new support for the PNL hypothesis that increased growth could reduce soil N availability, and has done so in these forests. If I've understood them correctly, these new analyses provide a substantial and important advance beyond other high-profile “plant δ¹⁵N trend” papers by showing at least part of why they occur. That said, this new evidence is for the increased growth phase of PNL; not its “limitation” phase. (In fact, one alternative interpretation of the new Figs 3c & 3d, in reversing their x & y axes, is that isotopically light wood δ¹⁵N correlates with increased rather than flat or declining growth rates as expected for N limitation – presumably because the “stimulation” phase likely dominates most of the period of record of this study.)

Overall, this is an outstanding set of analyses, addressing very important questions pertaining to the future of the terrestrial C sink, and I ultimately support its publication. I applaud the authors for their excellent, well-crafted and insightful analyses, and I extend an appreciation for the additional hard work and intriguing outcome of the δ¹⁵N-growth trend analyses.

Response: We sincerely appreciate the reviewer's insightful comments on our manuscript and thank them for their positive recommendation.

Comment: I suggest that the authors and editors consider two possible alternatives for publication:

- 1) Instead of “N limitation” in the title, consider a term such as “N oligotrophication” “decreased N availability” or similar. The authors could also readily speculate that their evidence of reduced N availability should limit additional growth enhancements from further rises in CO₂ as a logical predicted outcome. Yet declaring “Rising atmospheric CO₂ intensifies N limitation in boreal forests” still seems overstated.
- 2) Consider revising the new “d15N v growth change” analyses (Fig. 3c & 3d, Table S4) to consider if or how the d15N-growth change relationships differ over time, from the expected early stimulation to later plateau or decline. That might be to break up the analyses by decade or longer, or into two periods, before and after the increased growth trends flatten in Fig. 3a & 3b. If PNL and plant d15N act as hypothesized, there should be even stronger and clearer negative relationships between growth changes and wood d15N changes in the initial “stimulation” phase, but then those relationships should fall apart or even reverse during a later period of “N limitation.” Linking the 15N measurements to both the “stimulation” and “limitation” growth phases of PNL would not only link d15N trends to rising CO₂ but demonstrate how they provide long-sought field evidence of CO₂-driven N limitation of growth, and confirm some of its driving plant-soil mechanisms, which would be a spectacular accomplishment and major break-through.

Response: We thank the reviewer for offering alternative suggestions to clarify our manuscript; in response, we have decided to modify the language of the title from “*Rising atmospheric CO₂ intensifies nitrogen limitation in boreal forests*” to “*Rising atmospheric CO₂ reduces nitrogen availability in boreal forests*”. Furthermore, we have adjusted the language throughout the manuscript as follows:

L165-168 from “However, the remaining three Nr deposition variables (Total N deposition, NH_x, NO_y) showed weak positive relationships with δ¹⁵N, which either could indicate that higher Nr deposition reduces N limitation or that changing δ¹⁵N values of Nr deposition directly alters the δ¹⁵N values of trees” to “*However, the remaining three Nr deposition variables (Total N deposition, NH_x, NO_y) showed weak positive relationships with δ¹⁵N, which either could indicate that higher Nr deposition **increases N availability** or that changing δ¹⁵N values of Nr deposition directly alters the δ¹⁵N values of trees*”;

L192-194 from “Our results have multiple implications for understanding how boreal forest N limitation is responding to environmental change factors, in particular rising CO₂ and anthropogenic Nr deposition” to “*Our results have multiple implications for **understanding the response of boreal forest N availability** to environmental change factors, particularly rising atmospheric CO₂ and anthropogenic Nr deposition*”;

L199-202 from “As our sampling area spanned a large gradient in latitude, forest biomass, mean annual temperature, and Nr deposition, these findings are broadly relevant for understanding the trajectory of N limitation in northern forests where declining δ¹⁵N chronologies have previously been identified” to “*As our sampling area spanned a large gradient in latitude, forest biomass, mean annual temperature, and Nr deposition, these findings are broadly relevant for understanding the **trajectory of N availability** in northern forests where declining δ¹⁵N chronologies have previously been identified*”;

L226-229 from “In support that increased forest growth may be contributing to N limitation, data from the Swedish National Forest Inventory (NFI) shows that growth of mesic pine and spruce forests, similar to those from which we established δ¹⁵N chronologies, show a long-term growth increase since

the 1950s in all regions (Fig. 3a,b)” to “*In support that increased forest growth may contribute to **reducing N availability**, data from the Swedish National Forest Inventory (NFI) shows that growth of mesic pine and spruce forests, similar to those from which we established $\delta^{15}\text{N}$ chronologies, show a long-term growth increase since the 1950s in all regions (Fig. 3a,b)*”;

L246-248 from “These findings support our interpretation that intensified N limitation, as indicated by our $\delta^{15}\text{N}$ chronologies, may reduce the sensitivity of boreal forest growth to future increases in CO_2 ” to “*These findings support our interpretation that **reduced N availability**, as indicated by our $\delta^{15}\text{N}$ chronologies, may reduce the sensitivity of boreal forest growth to future increases in CO_2* ”; and **L 282-284** from “It further suggests that increasing N limitation in response to rising CO_2 will be a more dominant driver of boreal forest C exchange in the future, compared to N enrichment from atmospheric N_r deposition” to “*It further suggests that **declining N availability** in response to rising CO_2 will be a more dominant driver of boreal forest C exchange in the future, compared to N enrichment from atmospheric N_r deposition*”.

Comment: Please clarify a few details in the new Fig. 3c & 3d that will help convey how the analyses were done.

- Isn't there a time component for the “relative forest volume growth change” values on the x-axes? Either as a difference in growth rates from the overall mean (in $\text{m}^3 \text{ha}^{-1} \text{yr}^{-1}$) as the text seems to describe, or as a cumulative $\text{m}^3 \text{ha}^{-1}$ difference over some time period (if so, which period)?
- The $\delta^{15}\text{N}$ values on the y-axes are those measured for 5-year periods corresponding to each “relative forest growth change” value, correct?

Response: Regarding the first point, we thank the reviewer for bringing this omission to our attention. The error has been corrected in the figure and now reflects time: relative forest volume growth change ($\text{m}^3 \text{ha}^{-1} \text{yr}^{-1}$). Concerning the second point, our $\delta^{15}\text{N}$ values were measured from 10-year wood segments, whereas the forest inventory growth data in these analyses covers only a five-year increment. The wood core samples from which we obtained $\delta^{15}\text{N}$ values were obtained from temporary sample plots, as the Swedish National Forest Inventory exclusively cores trees within these plots. These cores were used to estimate plot growth for a five-year growth increment, which we subsequently used for $\delta^{15}\text{N}$ analysis. For $\delta^{15}\text{N}$, we focused on 10-year increments, which were necessary to obtain sufficient wood mass for analysis, given the very low nitrogen content of wood. Consequently, out of necessity, there is a slight mismatch in the temporal duration of these datasets; however, we do not consider this problematic in our analysis, as both the growth trends and $\delta^{15}\text{N}$ values we report develop gradually over time, spanning multiple decades. This clarification is provided in the Methods section, lines 629-630.